# CS-Bench: A Comprehensive Benchmark for Large Language Models towards Computer Science Mastery

**Xiaoshuai Song**[*], **Muxi Diao**[*], **Guanting Dong**[†], **Zhengyang Wang**, **Yujia Fu**,
**Runqi Qiao**, **Zhexu Wang**, **Dayuan Fu**, **Huangxuan Wu**, **Bin Liang**, **Weihao Zeng**,
**Yejie Wang**, **Zhuoma GongQue**, **Jianing Yu**, **Qiuna Tan**, **Weiran Xu**[†]
Beijing University of Posts and Telecommunications, Beijing, China
{songxiaoshuai, dmx, xuweiran}@bupt.edu.cn

## ABSTRACT

Large language models (LLMs) have demonstrated significant potential in advancing various fields of research and society. However, the current community of LLMs overly focuses on benchmarks for analyzing specific foundational skills (e.g. mathematics and code generation), neglecting an all-round evaluation of the computer science field. To bridge this gap, we introduce CS-Bench, the first multilingual (English, Chinese, French, German) benchmark dedicated to evaluating the performance of LLMs in computer science. CS-Bench comprises approximately 10K meticulously curated test samples, covering 26 subfields across 4 key areas of computer science, encompassing various task forms and divisions of knowledge and reasoning. Utilizing CS-Bench, we conduct a comprehensive evaluation of over 30 mainstream LLMs, revealing the relationship between CS performance and model scales. We also quantitatively analyze the reasons for failures in existing LLMs and highlight directions for improvements, including knowledge supplementation and CS-specific reasoning. Further cross-capability experiments show a high correlation between LLMs' capabilities in computer science and their abilities in mathematics and coding. Moreover, expert LLMs specialized in mathematics and coding also demonstrate strong performances in several CS subfields. Looking ahead, we envision CS-Bench serving as a cornerstone for LLM applications in the CS field and paving new avenues in assessing LLMs' diverse reasoning capabilities. Our project homepage is available at https://csbench.github.io/.

## 1 INTRODUCTION

Large language models (LLMs), exemplified by ChatGPT (OpenAI, 2022), have emerged as a significant milestone in artificial intelligence (AI), demonstrating potential far beyond natural language processing (NLP) (Zhao et al., 2023; Chang et al., 2024). These models are increasingly impacting diverse fields including education, industry, and scientific research (Guha et al., 2023; Guo et al., 2023; Xiao et al., 2023; Huang et al., 2023a; Zhou et al., 2023a; Zhao et al., 2024; Zhang et al., 2024a). As computer science (CS) continues to be the cornerstone of modern information technology, from the advent of electronic computers to today's AI advancements (Denning, 2000; Campbell-Kelly et al., 2023), a key challenge lies in enabling LLMs to better utilize CS knowledge to serve humanity in the evolving intelligent era (Donadel et al., 2024; Murtuza, 2024; Marques et al., 2024).

Understanding the performance of LLMs in computer science is fundamental to the research and application of LLMs within the field. Despite studies like MMLU and C-Eval (Hendrycks et al., 2021a; Huang et al., 2023b; Liu et al., 2023a; Li et al., 2024; Gu et al., 2024) covering a wide range of fields including CS, their broad scope implies that CS is merely a component within the multiple categories of science and engineering, overlooking the importance of thoroughly evaluating the CS field. Moreover, such evaluation result can further guide the development of LLMs, offering

---

[*]Equal Contribution
[†]Corresponding Authors, and Weiran Xu is the primary one.

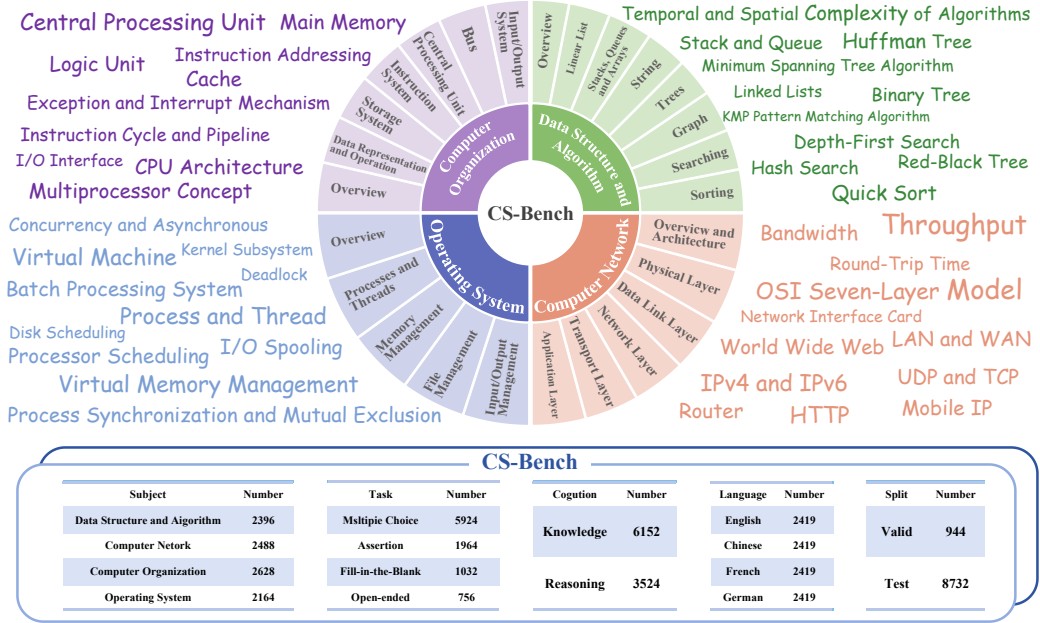

Figure 1: Overview diagram and statistics of CS-Bench.

practical insights to advance the corresponding capabilities. Recently, a series of studies have devoted on actively assessing and analyzing the capabilities of LLMs in mathematics, coding, and logical reasoning (Frieder et al., 2023; Collins et al., 2023; Wu et al., 2023; Yuan et al., 2023b; Dong et al., 2024; Liu et al., 2024; Zhang et al., 2024b; Lin et al., 2024; Liu et al., 2023b; Saparov et al., 2023; Xu et al., 2023; Wu et al., 2024). Unfortunately, efforts on LLMs in cross-capability evaluation is quite scarce. Considering the intersection of computer science with coding, mathematics, and reasoning abilities, we have grounds to believe that cross-capability research and analysis in CS can effectively propel the comprehensive development of the LLM community. Here, we are particularly interested in two research questions for evaluating LLMs' proficiency in computer science field:

***RQ1***: *How do LLMs perform in the field of computer science and what are the challenges and potential directions for improvement?*

***RQ2***: *What are the relationship between the abilities of LLMs in computer science, mathematics, and code programming?*

As the bedrock for exploration, we introduce CS-Bench, the first benchmark dedicated to evaluating the performance of LLMs in computer science. CS-Bench features high-quality, diverse task forms, varying capacities, and quadrilingual evaluation. Firstly, CS-Bench comprises approximately 10K carefully curated test items spanning 26 sections across 4 key CS domains. Unlike conventional benchmarks consisting solely of multiple-choice (MC) questions (Hendrycks et al., 2021a; Huang et al., 2023b; Li et al., 2024), CS-Bench includes 4 tasks: multiple-choice, assertion, fill-in-the-blank (FITB), and open-ended, to better simulate real-world scenarios and assess the robustness of LLMs to different task formats. In addition to knowledge-type questions assessing LLMs' mastery of CS knowledge, reasoning-type questions further evaluate LLMs' ability to apply CS knowledge for reasoning. Lastly, by supporting evaluation in English, Chinese, French, and German, CS-Bench allows assessment of LLMs' ability to tackle CS challenges in various language contexts.

In response to RQ1, we evaluate over 30 mainstream LLMs on CS-Bench. Our main findings are: (1) CS-Bench effectively differentiates the capabilities of LLMs in the CS field while also posing challenges to the best-performing GPT-4o/OpenAI-o1. (2) LLMs exhibit a consistent logarithmic growth pattern in scale and a linear growth pattern in scores on the CS-Bench. By establishing the scale-score fitting function, smaller models can be used to predict and guide the development of larger-scale models. (3) Further error type analysis indicates that the primary reason for the limited performance of LLMs is the lack of domain knowledge, and the CS-specific reasoning is difficult to achieve merely by enhancing general reasoning abilities, necessitating targeted reinforcement.

In response to RQ2, we perform a detailed analysis of the relationship of General LLMs' ability in three domains: mathematics, coding, and computer science, as well as the performance of code- and math-specific expert LLMs on CS-Bench. We observe consistent trends in the overall performance of the general LLMs across CS-Bench and scores in benchmarks related to mathematics and coding, indicating a strong correlation between LLM's computer science proficiency and its mathematical and programming abilities. Furthermore, despite a decline in general capabilities, some expert LLMs still exhibit improvements in certain areas of CS, such as data structures and algorithms, with more pronounced knowledge and reasoning capabilities evident in supplementary smaller-scale models.

To summarize, our contributions are as follows:

- We introduce CS-Bench, the first benchmark dedicated to evaluate the performance of LLMs in the field of computer science. CS-Bench supports four languages, covers four key areas with 26 subfields, and includes a diverse range of task formats.

- Utilizing CS Bench, we conduct a comprehensive evaluation of mainstream LLMs, revealing the relationship between CS performance and model scales. We also quantitatively analyze the reasons for failures in existing LLMs and highlight directions for improvement.

- We conduct exploratory experiments on LLMs' cross-ability and find a strong relationship between their CS proficiency and mathematical and programming abilities. Moreover, the expertise in mathematics and programming of expert LLMs can improve performance in specific CS subfields.

## 2    CS-BENCH

### 2.1    DESIGN PRINCIPLE

The objective of CS-Bench is to robustly assess the knowledge and reasoning capabilities of LLMs in different linguistic contexts within the field of computer science. To this end, our benchmark adheres to the following guidelines: (1) **Coverage of key domains:** it covers key areas of CS with finer subfields for specificity. (2) **Diverse task forms:** questions vary in format to simulate diverse real-world user queries. (3) **CS-specific reasoning:** it evaluates CS logical and arithmetic reasoning in addition to CS knowledge. (4) **Multilinguality support:** it supports assesses LLMs' performance in different language environments. Based on these criteria, CS-Bench focuses on quadrilingual evaluation in English, Chinese, French, and German, covering four domains: Data Structure and Algorithm (DSA), Computer Organization (CO), Computer Network (CN), and Operating System (OS). Twenty-six fine-grained subfields, diverse task forms, and divisions of knowledge and reasoning are further developed to enrich the dimensions of assessment and simulate real-world scenarios.

### 2.2    DATA COLLECTION

**Data Sources.** Diverse data sources are key to achieving the sample diversity of CS-Bench. Our raw data originates from three sources: (1) Computer science-related questions obtained from publicly available online channels, such as professional exams and practice tests. (2)Knowledge-type questions obtained through the initial manual extraction and subsequent adaptation of blog articles from various computer-related websites. (3) Construction of teaching materials and examination papers authorized by the authors' institutions. [1] The latter two cate-

Table 1: Comparison of perplexity (PPL) across evaluation datasets. The PPL of English and Chinese datasets is calculated on Llama2-7B and Qwen1.5-7B. "MC" denotes multiple-choice, and "ALL" denotes all tasks.

| English Dataset | PPL | Chinese Dataset | PPL |
|---|---|---|---|
| TruthfulQA (MC) | 7.73 | C-Eval | 11.47 |
| MMLU | 9.54 | CMMLU | 13.62 |
| CS-Bench (MC) | 11.86 | CS-Bench (MC) | 13.31 |
| CS-Bench (ALL) | 13.3 | CS-Bench (ALL) | 16.95 |

gories constitute the vast majority (over 70%) of the data, and these data are not directly exposed on the internet, effectively reducing the likelihood of LLMs encountering these questions during pre-training. We compare the perplexity (Jelinek et al., 1977) of models on CS-Bench English and Chinese datasets with several other prominent evaluation datasets in Table 1. In both English and Chinese, the perplexity of CS-Bench is comparable to or even higher than that of other datasets, further indicating the high quality of CS-Bench samples and the rarity of data leakage instances.

---

[1]More collection and processing procedures can be found in Appendix C.

**Data Processing.**   The data processing relies on a team composed of five members, each holding a bachelor's degree in computer science and receiving appropriate compensation. Initially, we parse questions and answers for each sample from the data sources either automatically or manually. Subsequently, we manually label questions with knowledge-type or reasoning-type tags depending on whether in-depth reasoning and calculation are required. For reasoning-type questions, we attempt to collect explanations from the data sources whenever possible; otherwise, we handle them through cross-annotation and verification among team members. We first construct Chinese data, then translate it into other languages using GPT-4, supplemented by manual checks, to create t multilingual data. Finally, we conduct thorough manual checks on the entire dataset to ensure quality. As this benchmark pertains to objective knowledge and reasoning in the field of computer science, the annotation content is not influenced by regional or cultural differences among annotators.

**Statistics.**   CS-Bench is an evaluation benchmark supporting quadrilingual assessment, encompassing a total of 26 subfields across 4 domains, with a cumulative total of 9676 samples. These samples encompass various task formats including multiple-choice, assertion, fill-in-the-blank, and open-ended questions. Besides, CS-Bench assesses both knowledge-type and higher-order reasoning-type questions, with each reasoning question accompanied by an explanation. To validate the effectiveness of models, we randomly sample 10% of the data for validation, using the remaining 90% for testing. The statistics of CS-Bench are shown in Figure 1, with detailed exposition provided in Appendix C.

## 3   EXPERIMENT

### 3.1   EXPERIMENTAL SETUP

**Evaluation Protocols.**   Due to the diverse task formats in CS-Bench, we first design question templates for each task type.  For comprehension tasks (MC and Assertion), we use regex to match LLM's predictions and then calculate their accuracy against the ground-truth answers. For generation tasks (FITB and Open-ended), due to the diversity of ground-truth answers, we score LLM's predictions by GPT-4 using standard answers in CS-Bench as references. In detail, FITB questions are scored as either 0 or 1, while the score range for Open-ended questions is 1-10, which is then linearly mapped to a range of 0.1 to 1. Finally, scores are weighted based on the quantity of each type to derive the ultimate overall score. It is worth emphasizing that while using GPT-4 for scoring generation tasks may introduce a certain threshold for evaluation, its primary purpose is to simulate diverse task formats in real-world scenarios. Therefore, we encourage isolating comprehension tasks from CS-Bench to facilitate automatic evaluation with no need for GPT-4. We provide the details of the evaluation setup in Appendix D, where we also verify the validity of GPT-4 scoring through its consistency with manually scored results, and the effect of different scoring models.

**Models.**   For open-source models, we selected Gemma-2B/7B (Team et al., 2024), Llama2-7B/13B/70B (Touvron et al., 2023), Llama3-8B/70B (Meta, 2024), Llama3.1-405B (Dubey et al., 2024), ChatGLM3-6B (THUDM, 2023), Baichuan2 (v2.0)-7B/13B (Yang et al., 2023), InternLM2-7B/20B (Cai et al., 2024) , Qwen1.5-4B/7B/14B/72B/110B (Alibaba, 2024), Mistral-7B (v0.2) (Jiang et al., 2023), Mixtral-8×7B (v0.1) (Jiang et al., 2024), Mistral-Large-123B (MistralAI, 2024), and DeepSeekLLM-7B/67B (Bi et al., 2024). For closed-source commercial models, we utilized PaLM-2 (palm-2-chat-bison) (Anil et al., 2023), Claude-2.1 (Anthropic, 2023), Claude-3 (opus) (Anthropic, 2024a), Claude-3.5 (Sonnet) (Anthropic, 2024b), as well as GPT-3.5, GPT-4 (0125 version) (Achiam et al., 2023), GPT-4o (OpenAI, 2024a) and OpenAI-o1-mini/preview (0912 version) (OpenAI, 2024b)[2]. To ensure the instruction-following abilities, we employ the official chat or instruction-tuned versions for all models. Details on these models are provided in Appendix D.4.

### 3.2   MAIN RESULTS

Table 2 presents the overall results of all foundation models directly answering questions under the zero-shot setting [3]. In summary, the overall scores of models range from 39.86% to 72.29%, demonstrating CS-Bench's effectiveness in distinguishing between the abilities of various models

---

[2]Due to the unique reasoning form of OpenAI-o1, we exclude it from Section 3.2 analyze it in Section 3.3.

[3]Due to space constraints, the results and analyses in other languages are provided in Appendix E.3.

Table 2: Zero-shot scores (%) of LLMs across domains on CS-Bench (EN), where "Klg" denotes knowledge-type, "Rng" denotes reasoning-type, and "Avg" denotes Average. The random scores are weighted as follows: 25% for MC, 50% for Assertion, 0% for FITB, and 10% for Open-ended. The highest scores for open-source and closed-source LLMs is marked in green and blue respectively.

| Model | Data Struc & Algo | | | Computer Organization | | | Computer Network | | | Operating System | | | Overall | | |
|---|---|---|---|---|---|---|---|---|---|---|---|---|---|---|---|
| | Klg | Rng | Avg | Klg | Rng | Avg | Klg | Rng | Avg | Klg | Rng | Avg | Klg | Rng | Avg |
| Random | 28.04 | 24.63 | 26.65 | 26.57 | 25.24 | 26.13 | 26.34 | 22.49 | 24.98 | 29.06 | 24.23 | 27.27 | 27.4 | 24.12 | 26.2 |
| *Open-source LLM (Scale<10B)* | | | | | | | | | | | | | | | |
| Gemma-2B | 56.76 | 23.44 | 43.20 | 47.69 | 30.18 | 41.92 | 45.22 | 26.38 | 38.59 | 37.79 | 31.32 | 35.39 | 46.89 | 27.59 | 39.86 |
| Qwen1.5-4B | 58.76 | 36.56 | 49.72 | 52.31 | 33.88 | 46.23 | 52.70 | 33.97 | 46.11 | 40.03 | 38.52 | 39.47 | 51.18 | 35.70 | 45.54 |
| ChatGLM3-6B | 51.10 | 34.08 | 44.17 | 48.11 | 32.73 | 43.04 | 51.15 | 32.66 | 44.64 | 43.57 | 37.03 | 41.14 | 48.63 | 34.07 | 43.33 |
| Llama2-7B | 51.51 | 32.61 | 43.82 | 48.89 | 31.82 | 43.26 | 46.72 | 30.75 | 41.10 | 41.04 | 26.26 | 35.55 | 47.15 | 30.48 | 41.08 |
| DeepseekLLM-7B | 56.42 | 28.94 | 45.23 | 52.09 | 32.48 | 45.62 | 52.43 | 31.41 | 45.03 | 41.66 | 31.98 | 38.06 | 50.87 | 31.11 | 43.67 |
| Baichuan2-7B | 53.11 | 34.95 | 45.72 | 45.10 | 38.67 | 42.98 | 51.26 | 34.27 | 45.28 | 43.47 | 33.63 | 39.82 | 48.29 | 35.33 | 43.57 |
| Gemma-7B | 59.53 | 35.18 | 49.62 | 49.97 | 33.27 | 44.46 | 60.87 | 37.09 | 52.50 | 48.67 | 34.23 | 43.31 | 54.90 | 35.02 | 47.66 |
| Qwen1.5-7B | 59.90 | 35.28 | 49.88 | 55.21 | 42.73 | 51.09 | 61.56 | 43.02 | 55.04 | 52.01 | 39.78 | 47.47 | 57.34 | 40.08 | 51.05 |
| InternLm2-7B | 59.57 | 40.92 | 51.98 | 58.83 | 37.94 | 51.94 | 62.65 | 40.60 | 54.89 | 50.94 | 39.29 | 46.61 | 58.31 | 39.77 | 51.56 |
| Mistral-7B | 63.24 | 34.86 | 51.69 | 57.52 | 38.67 | 51.30 | 61.48 | 44.92 | 55.65 | 51.66 | 43.79 | 48.73 | 58.63 | 40.44 | 52.01 |
| Llama3-8B | 66.25 | 37.29 | 54.46 | 55.38 | 40.67 | 50.53 | 62.21 | 53.02 | 58.98 | 55.26 | 49.34 | 53.06 | 59.75 | 44.97 | 54.37 |
| *Open-source LLM (Scale>10B)* | | | | | | | | | | | | | | | |
| Llama2-13B | 51.74 | 35.00 | 44.93 | 51.81 | 36.18 | 46.66 | 53.03 | 37.99 | 47.74 | 48.12 | 32.36 | 42.27 | 51.31 | 35.46 | 45.54 |
| Baichuan-13B | 54.82 | 33.39 | 46.10 | 50.50 | 39.52 | 46.88 | 55.87 | 42.21 | 51.06 | 48.44 | 34.73 | 43.35 | 52.53 | 37.44 | 47.03 |
| Qwen1.5-14B | 64.95 | 46.74 | 57.54 | 60.06 | 45.58 | 55.28 | 68.66 | 52.91 | 63.12 | 56.56 | 51.48 | 54.67 | 62.79 | 49.18 | 57.83 |
| InternLm2-20B | 66.72 | 38.21 | 55.11 | 58.38 | 39.82 | 52.26 | 64.13 | 50.35 | 59.28 | 53.51 | 46.43 | 50.88 | 60.81 | 43.66 | 54.56 |
| Qwen1.5-32B | 69.70 | 51.19 | 62.17 | 67.63 | 52.91 | 62.78 | 69.23 | 58.74 | 65.54 | 60.06 | 56.21 | 58.63 | 66.87 | 54.72 | 62.45 |
| Mistral-8×7B | 70.94 | 40.50 | 58.55 | 66.88 | 42.06 | 58.70 | 67.49 | 52.86 | 62.34 | 57.56 | 51.65 | 55.37 | 65.91 | 46.66 | 58.90 |
| DeepseekLLM-67B | 69.70 | 44.17 | 59.31 | 63.59 | 39.15 | 55.53 | 69.04 | 50.25 | 62.43 | 57.86 | 50.11 | 54.98 | 65.23 | 45.96 | 58.22 |
| Llama2-70B | 64.28 | 41.51 | 55.01 | 56.35 | 40.85 | 51.24 | 61.99 | 43.07 | 55.33 | 51.79 | 41.15 | 47.84 | 58.73 | 41.68 | 52.52 |
| Llama3-70B | 75.72 | 53.03 | 66.48 | 71.45 | 51.09 | 64.74 | 74.78 | 63.02 | 70.64 | 63.77 | 58.08 | 61.65 | 71.65 | 56.36 | 66.08 |
| Qwen1.5-72B | 72.71 | 50.69 | 63.75 | 69.28 | 54.12 | 64.28 | 71.97 | 66.73 | 70.13 | 63.96 | 59.62 | 62.35 | 69.63 | 57.75 | 65.31 |
| Qwen1.5-110B | 73.11 | 53.58 | 65.16 | 73.65 | 54.18 | 67.23 | 75.36 | 70.75 | 73.74 | 64.55 | 65.27 | 64.82 | 71.98 | 60.91 | 67.95 |
| Mistral-Large-123B | 79.43 | 59.82 | 71.45 | 74.21 | 63.76 | 70.76 | 77.98 | 68.19 | 74.54 | 66.92 | 66.10 | 66.61 | 74.84 | 64.37 | 71.03 |
| Llama3.1-405B | 77.63 | 58.85 | 69.99 | 76.57 | 57.58 | 70.31 | 78.50 | 69.90 | 75.47 | 68.86 | 64.73 | 67.33 | 75.64 | 62.81 | 70.96 |
| *Closed-source LLM* | | | | | | | | | | | | | | | |
| PaLM-2 | 70.07 | 38.98 | 57.41 | 63.81 | 41.91 | 56.59 | 65.11 | 49.43 | 59.59 | 60.41 | 45.96 | 55.22 | 64.85 | 44.01 | 57.26 |
| Claude-2.1 | 68.39 | 44.54 | 58.68 | 62.09 | 50.24 | 58.18 | 66.58 | 52.81 | 61.74 | 53.93 | 50.55 | 52.67 | 62.97 | 49.42 | 58.04 |
| Claude-3-Opus | 77.53 | 52.25 | 67.24 | 72.53 | 64.12 | 69.76 | 75.08 | 68.69 | 72.83 | 64.36 | 62.80 | 63.78 | 72.57 | 61.75 | 68.63 |
| GPT-3.5 | 71.34 | 39.22 | 58.27 | 60.78 | 42.97 | 54.91 | 65.27 | 52.16 | 60.66 | 54.42 | 39.01 | 48.69 | 63.04 | 43.45 | 55.91 |
| GPT-4 | 78.53 | 59.36 | 70.73 | 75.40 | 59.21 | 70.06 | 77.38 | 67.64 | 73.95 | 67.21 | 64.40 | 66.16 | 74.85 | 62.66 | 70.41 |
| Claude-3.5-Sonnet | 77.16 | 58.07 | 69.39 | 75.13 | 61.76 | 70.72 | 77.92 | 71.46 | 75.65 | 69.55 | 64.73 | 67.76 | 75.13 | 63.97 | 71.07 |
| GPT-4o | 81.51 | 57.80 | 71.86 | 75.60 | 58.61 | 70.00 | 80.57 | 71.76 | 77.47 | 69.35 | 68.68 | 69.10 | 76.95 | 64.15 | 72.29 |

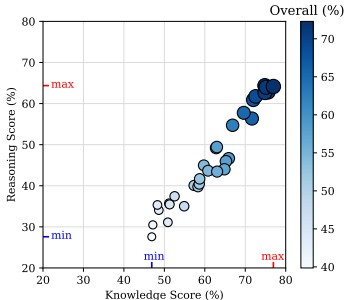
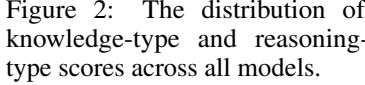

Figure 2: The distribution of knowledge-type and reasoning-type scores across all models.

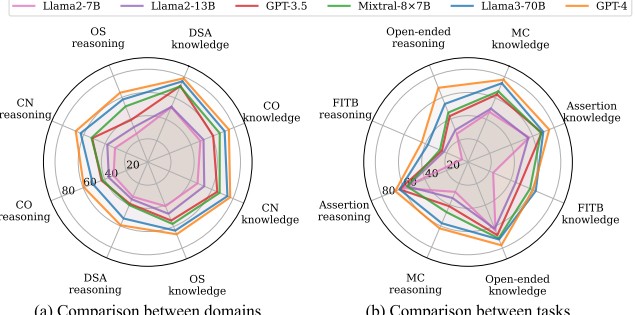

(a) Comparison between domains

(b) Comparison between tasks

Figure 3: Comparison of representative LLMs' scores across different domains and tasks.

in the field of CS while also posing significant challenges to the best-performing existing models. Subsequently, we conduct a comprehensive analysis of the experimental results from various aspects including **Foundation Models**, **Knowledge & Reasoning**, **Domains**, and **Task Formats**.

**Comparison between Foundation Models.** Firstly, the closed-source models Claude3.5/GPT-4o represent the highest standard on CS-Bench, with a proficiency exceeding 70%. Secondly, leading open-source models (Mistral-Large-123B and Llama3.1-405B) have significantly narrowed the gap between open-source and closed-source models. On one hand, both models exhibit a substantial increase in parameters; on the other hand, current closed-source models are tending towards a balance between performance and efficiency, exemplified by GPT-4o-mini and Gemini Flash. Thirdly, newer models demonstrate significant improvements compared to earlier versions. For example,

| Type | Model | Score | Reasoning / Completion Tokens |
|---|---|---|---|
| Knowledge | GPT-4o | 76.95 | - / 8.67 |
| | OpenAI-o1-mini | 77.60 (+0.65) | 251.0 / 269.06 (×31.03) |
| | OpenAI-o1-preview | 83.61 (+6.66) | 518.49 / 545.37 (×62.9) |
| Reasoning | GPT-4o | 64.15 | - / 56.43 |
| | OpenAI-o1-mini | 76.12 (+11.97) | 500.1 / 546.78 (×9.69) |
| | OpenAI-o1-preview | 80.98 (+16.83) | 1106.43 / 1198.78 (×21.24) |
| Overall | GPT-4o | 72.29 | - / 26.08 |
| | OpenAI-o1-mini | 77.06 (+4.77) | 341.8 / 370.29 (×14.2) |
| | OpenAI-o1-preview | 82.65 (+10.36) | 732.79 / 783.54 (×30.04) |

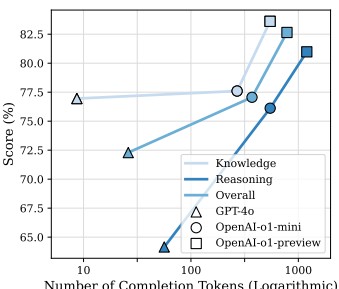

Figure 4: Comparison between OpenAI-o1 and GPT-4o. Reasoning Tokens refer to tokens used for internal reasoning in OpenAI-o1; Completion Tokens refer to all tokens consumed both internally and in external output. "+" and "×" indicate performance gains and token consumption of OpenAI-o1 over GPT-4o.

Figure 5: The relationship between CS-Bench scores and the number of tokens consumed.

among models with scales below 10B, Llama3-8B performs the best, rivaling previous much larger-scale models and even surpassing Llama2-70B, indicating significant potential for compression in model parameters (Delétang et al., 2024). Lastly, while performance variations exist among models of different families at the same scale, models within the same family continue to improve with increasing scale on CS-Bench (see detailed scale analysis in Section 3.4).

**Comparison of Knowledge and Reasoning.** Overall, all models perform worse on reasoning (average 45.60%) compared to knowledge scores (average 60.61%), indicating that reasoning poses a greater challenge to LLMs compared to knowledge. As shown in Figure 2, there is a strong positive correlation between reasoning scores and knowledge scores. However, this correlation is not absolute. For instance, PaLM-2 has a higher knowledge score but a lower reasoning score compared to Claude-2.1, showing PaLM-2's weakness in applying knowledge. Furthermore, more powerful LLMs demonstrate a stronger ability to use knowledge for reasoning compared to weaker LLMs. This is reflected in the much lower reasoning scores of weaker models relative to their knowledge scores. However, as the model's capability increases, the growth in reasoning scores is more pronounced than that of knowledge scores, gradually bridging the gap between knowledge and reasoning abilities.

**Comparison between Domains.** First, regarding knowledge scores in Table 2 and Figure 3 (a), models generally perform best in DSA and worst in OS, which we attribute mainly to differences in the scale of pretraining data and the varying learning capabilities induced by model size. Second, the demand for reasoning ability varies across different domains, as evidenced by the gap between knowledge and reasoning scores. A notable example is GPT-4o, which shows close knowledge and reasoning scores in OS, while exhibiting extreme differences in DSA, with the highest and lowest scores, respectively. We further explore LLMs' performance in fine-coursed subfields in Appendix E.1 and explore the impact of Code and Math abilities on different CS domains in Section 3.5.

**Comparison between Tasks.** As shown in Figure 3 (b) and Table 14, given the varying initial random scores, LLMs generally performs best on Assertion questions (average 63.14% across all models), followed by MC questions (average 55.45%), Open-ended questions (average 49.3%), and performs worst on FITB questions (average 41.58%). However, the variation in task format sensitivity is highly pronounced in weaker models, while stronger models can mitigate the disparities caused by different task formats, exhibiting robustness. For instance, Llama2-7B scores only 26.19% on Open-ended reasoning but 60.61% on Assertion reasoning, whereas GPT-4 scores comparably on both Open-ended reasoning (68.94%) and Assertion reasoning (67.68%).

## 3.3 OPENAI-O1 RESULT

Compared to general models' reasoning processes, OpenAI-o1 models think before they answer, forming a long internal chain of thought through reasoning tokens. As shown in Figure 4, OpenAI-o1 significantly improves reasoning performance. While OpenAI-o1-mini nearly matches GPT-4o on knowledge-type questions, it boosts scores by 11.97 points on reasoning problems. Furthermore, OpenAI-o1-preview achieves substantial progress in both knowledge and reasoning questions. However, this performance gain comes at the expense of a high token consumption. For instance,

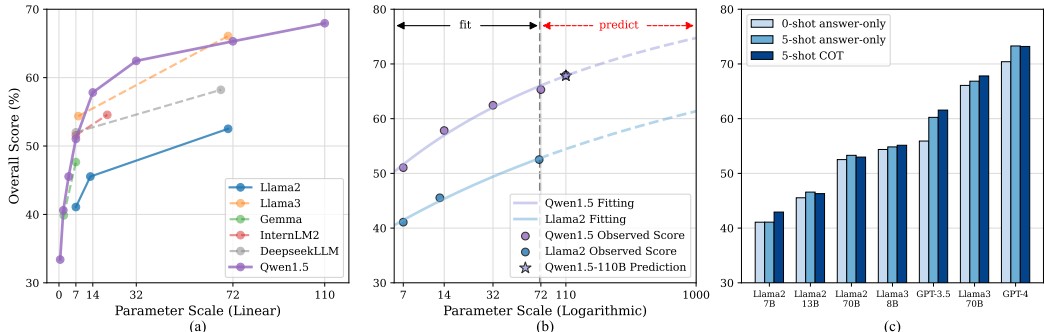

Figure 6: (a) The performance of LLMs at different parameter scales. (b) The scale-score fitting curve of Qwen1.5 and Llama2 series. (c) Comparison of models under 0-shot answer-only, 5-shot answer-only, and 5-shot COT settings.

OpenAI-o1-preview consumes an average of nearly 1.2K tokens on reasoning questions, 21.24 times that of GPT-4o. Figure 5 visualizes the relationship between model performance and completion token consumption. For reasoning questions, performance roughly increases logarithmically with the number of tokens, while the gains from token expenditure are lower for knowledge-type questions. We conduct a case study in Appendix E.7, comparing the responses of GPT-4o and OpenAI-o1.

## 3.4 QUALITATIVE ANALYSIS

**Relationship between Scores and Model Scales.** To investigate how model performance varies with parameter size, we examine several model families and plot the results in Figure 6 (a). Models within the same family consistently improve as parameter size increases, but performance gains diminish with larger models. For instance, the score in Qwen1.5 improves by 16.19% from 0.5B to 7B, by 7.11% from 14B to 72B, and by only 2.66% from 72B to 110B. Figure 6 (b) shows that when the parameter scale grows exponentially, the score increases approximately linearly, indicating a logarithmic scale pattern in the CS field. Given the substantial computational resources required for large-scale models, we aim to establish the relationship between model scales and scores to predict the performance of larger-scale models by fitting smaller-scale model scores. Due to space limitations, the specific design and implementation of the fitting function are provided in Appendix E.2. Overall, we fit the functions of Llama2 and Qwen1.5 series based on models ranging from 7B to 70/72B. We validate the fitting function on Qwen-1.5 110B, where the predicted value (67.83%) closely matches the actual value (67.95%), enabling further predictions for theoretical models, even up to 1000B.

**Comparison between Zero-shot, Few-shot and COT Prompting.** To investigate the impact of few-shot prompts and chain of thought (COT (Wei et al., 2022)) on model performance, we evaluate model's performance under 5-shot answer-only (AO) and 5-shot COT prompts in Figure 6 (c), where the prompt samples are sampled from the validation set and match the domain of the test questions. Given that model-generated results under 0-shot COT often don't adhere to specific formats, making regular matching difficult, we omit 0-shot COT experiments, similar to C-Eval. Additionally, for Open-ended questions, since the answers include detailed explanations, 5-shot COT is the same as 5-shot AO. For all tested models, the 5-shot prompts show improvement compared to 0-shot, with average increases of 1.47% for 5-shot AO and 2.00% for 5-shot COT, respectively. Moreover, the efficacy of few-shot prompts in bringing improvements appears more pronounced in some robust models such as GPT-3.5 and GPT-4, owing to their superior in-context learning capabilities.

**Analysis of Error Types.** To delve into the roots of LLMs' failure on CS-Bench and offer pathways toward improvement, we acquire the solution process of model errors under 5-shot COT, and utilize GPT-4 to categorize each error type in MC questions in Figure 7. It should be emphasized that models may cause joint errors, resulting in more than one error type assigned to a single answer. In general, from Llama2-7B all the way to GPT-4, the total number of errors continues to decrease for both knowledge-type and reasoning-type questions. For knowledge-type questions, both single concept errors and concept confusion show a decreasing trend. Initially, some completely wrong concepts transitioning to partially erroneous ones and subsequently being eliminated, thus exhibiting an initial

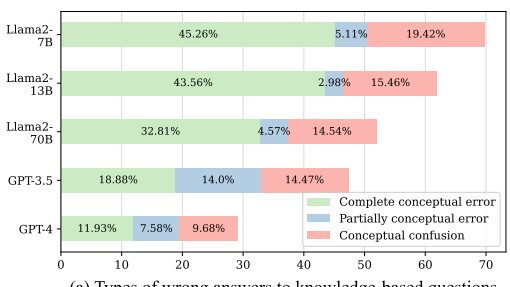 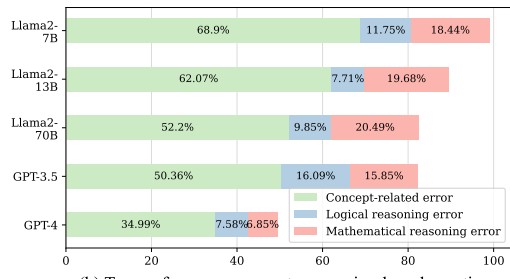

(a) Types of wrong answers to knowledge-based questions

(b) Types of wrong answers to reasoning-based questions

Figure 7: The proportion of different error types varies by models for multiple-choice questions.

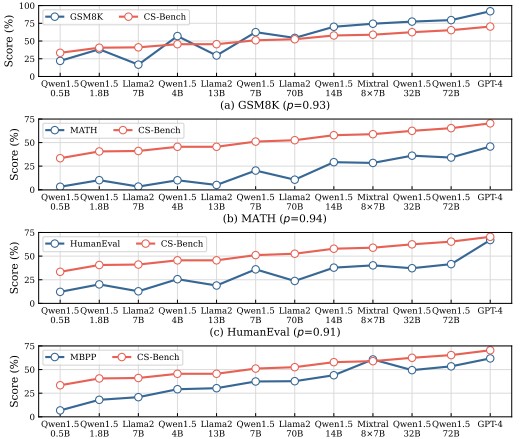

Figure 8: The score changes on CS-Bench as LLM's Math/Code score increases. $p$ denotes Pearson correlation coefficient. We obtain the scores on Math/Code datasets from (Alibaba, 2024).

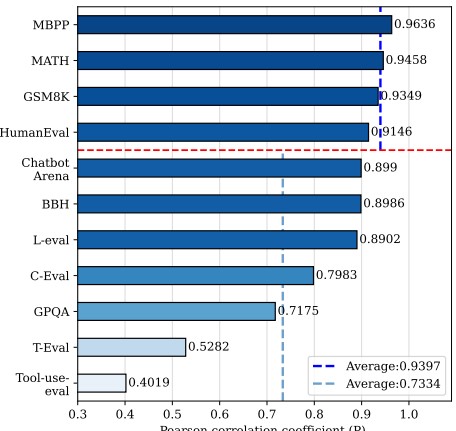

Figure 9: The consistency of model performance between CS-Bench and other benchmarks. Model scores from other benchmarks are sourced from their official papers/websites or the OpenLLM Leaderboard.

rise followed by a decline in partial concept errors. For reasoning-type questions, we observe that a significant portion of errors still fall under the category of knowledge-based mistakes. While stronger models have evidently reduced arithmetic reasoning errors for reasoning inaccuracies, there hasn't been much change in logic reasoning errors specific to the CS field. Our analysis highlights that reinforcing CS knowledge concepts is the most direct and effective approach to improving LLMs' performance in the field of CS. Furthermore, significant improvements in CS reasoning performance are challenging to achieve solely by enhancing general reasoning abilities and mathematical reasoning, necessitating CS-specific reinforcement. More details can be found in E.4.

### 3.5 WHAT'S THE RELATIONSHIP BETWEEN CS, MATH, AND CODE ABILITIES OF LLMS?

To explore the relationship between CS proficiency and the mathematical and coding capabilities of models, we investigate (1) the performance of general LLMs across the fields of Math, Code, and CS, and (2) the performance of LLMs specialized in Code and Math within the field of CS. [4]

**Exploration on General Models.** In Figure 8, we illustrate how the models' performance on CS-Bench varies with increasing scores on the Math datasets (GSM8K (Cobbe et al., 2021), MATH (Hendrycks et al., 2021b)) and Code datasets (HumanEval (Chen et al., 2021), MBPP (Austin et al., 2021)). The overall trend in CS-Bench performance closely aligns with changes in Math and Code scores, as indicated by a Pearson correlation coefficient (Cohen et al., 2009) exceeding 0.9. Besides the general enhancement of diverse competencies that superior models typically bring, we consider this evidence suggests a strong correlation between CS proficiency and skills in Math and Code.

---

[4]We leave the analysis from the model representation perspective in Appendix E.5.

Table 3: The performance of the Math-expert LLMs on CS-Bench (EN). We use blue to emphasize areas where the expert LLMs improve compared to the Chat LLMs.

| Model | Type | DSA | | CO | | CN | | OS | | All | | |
|---|---|---|---|---|---|---|---|---|---|---|---|---|
| | | Klg | Rng | Klg | Rng | Klg | Rng | Klg | Rng | Klg | Rng | Avg |
| InternLm2-7B | Chat | 59.57 | 40.92 | 58.83 | 37.94 | 62.65 | 40.60 | 50.94 | 39.29 | 58.31 | 39.77 | 51.56 |
| InternLM-Math-7B (Ying et al., 2024) | Math | 60.23 | 31.56 | 50.56 | 38.61 | 55.93 | 44.47 | 47.69 | 43.85 | 53.64 | 39.41 | 48.45 |
| DeepseekLLM-7B | Chat | 56.42 | 28.94 | 52.09 | 32.48 | 52.43 | 31.41 | 41.66 | 31.98 | 50.87 | 31.11 | 43.67 |
| DeepSeekMath-Instruct-7B (Shao et al., 2024) | Math | 63.98 | 34.82 | 55.13 | 39.64 | 61.26 | 42.16 | 45.29 | 42.69 | 56.68 | 39.67 | 50.49 |
| Llama2-13B | Chat | 51.74 | 35.00 | 51.81 | 36.18 | 53.03 | 37.99 | 48.12 | 32.36 | 51.31 | 35.46 | 45.54 |
| MAammoTH-13B (Yue et al., 2023) | Math | 50.84 | 28.26 | 46.16 | 34.61 | 51.39 | 30.45 | 34.94 | 32.64 | 46.20 | 31.32 | 40.78 |
| Llama2-70B | Chat | 64.28 | 41.51 | 56.35 | 40.85 | 61.99 | 43.07 | 51.79 | 41.15 | 58.73 | 41.68 | 52.52 |
| WizardMath-70B (Luo et al., 2023a) | Math | 60.17 | 28.67 | 56.41 | 34.91 | 58.52 | 41.51 | 47.01 | 42.53 | 55.77 | 36.67 | 48.82 |

Table 4: The performance of the Code-expert LLMs on CS-Bench (EN).

| Model | Type | DSA | | CO | | CN | | OS | | All | | |
|---|---|---|---|---|---|---|---|---|---|---|---|---|
| | | Klg | Rng | Klg | Rng | Klg | Rng | Klg | Rng | Klg | Rng | Avg |
| Llama2-7B | Chat | 51.51 | 32.61 | 48.89 | 31.82 | 46.72 | 30.75 | 41.04 | 26.26 | 47.15 | 30.48 | 41.08 |
| CodeLlama-7B (Rozière et al., 2024) | Code | 58.90 | 36.15 | 45.46 | 36.24 | 52.87 | 26.23 | 44.33 | 25.33 | 50.36 | 31.09 | 43.34 |
| Dolphcoder-7B (Wang et al., 2024c) | Code | 50.13 | 36.47 | 34.71 | 34.36 | 41.78 | 23.92 | 40.03 | 28.35 | 41.40 | 30.82 | 37.54 |
| WizardCoder-7B (Luo et al., 2023b) | Code | 47.42 | 33.58 | 35.54 | 37.09 | 41.17 | 26.03 | 40.88 | 30.60 | 41.02 | 31.73 | 37.63 |
| Llama2-13B | Chat | 51.74 | 35.00 | 51.81 | 36.18 | 53.03 | 37.99 | 48.12 | 32.36 | 51.31 | 35.46 | 45.54 |
| CodeLlama-13B (Rozière et al., 2024) | Code | 59.87 | 34.17 | 44.96 | 35.82 | 51.56 | 35.83 | 43.28 | 34.56 | 49.84 | 35.08 | 44.47 |
| WizardCoder-13B (Luo et al., 2023b) | Code | 50.80 | 32.98 | 38.69 | 35.27 | 43.42 | 28.34 | 40.88 | 34.29 | 43.27 | 32.59 | 39.38 |

Another piece of evidence is the consistency between CS-Bench and other benchmarks, including non-CS scientific benchmarks (GPQA (Rein et al., 2023)), specialized benchmarks (Tool-use-eval (Alibaba, 2024), T-Eval (Chen et al., 2024), L-eval (An et al., 2023), BBH (Suzgun et al., 2022)), and general benchmarks (Chatbot Arena (Zheng et al., 2023), C-Eval(Huang et al., 2023b)). The results are shown in Figure 9. We observe that CS-Bench and agents, tool usage, and other fields have a low correlation (P<0.8), whereas their correlation with benchmarks involving long texts, multi-step reasoning, and general tasks is moderate (P<0.9). In contrast, the P-values for CS-Bench and benchmarks in mathematics/coding are all above 0.9, reaching as high as 0.96 in MBPP, which strongly supports the assertion that the model's CS capabilities are closely related to its coding and mathematical abilities. Additionally, we find that Chatbot Arena, BBH, and L-eval include subsets related to mathematics and coding, resulting in their correlation with CS-Bench being higher than that of other non-Code/Math benchmarks.

Next, we examine models with inconsistent patterns between CS and Math/Code. In the Math domain, Qwen1.5-7B outperforms Llama2-70B in both GSM8K and MATH, yet in CS-Bench, Llama2-70B surpasses Qwen1.5-7B. In the Code domain, Mixtral-8×7B performs better than Qwen1.5-32B on HumanEval and MBPP, whereas the opposite is observed on CS-Bench. Given the NLP community's sustained focus on the Code and Math domains, some recently released models have been trained on a large amount data in these domains, leading to smaller-scale models outperforming much larger-scale ones (e.g., Qwen1.5-7B surpassing Llama2-70B). However, in the CS domain, due to insufficient attention and training data, even excellent small-scale models struggle to surpass much larger-scale models. This also indicates that CS-Bench has not been overfitted during LLM pretraining, making it a fairer benchmark for measuring model performance differences.

**Exploration on Expert Models.** We present the results of the Math and Code expert LLMs in Tables 3 and 4. Compared to general Chat LLMs, expert LLMs usually sacrifice other abilities to boost proficiency in Math or Code, which is reflected in the lower overall performance of most expert LLMs. Therefore, we are more concerned with identifying the specific aspects of CS where Math and Code models show improvement. Regarding mathematics, InternLm-Math-7B improves InternLm2-7B's performance in CO, CN, and OS reasoning tasks, while DeepseekMath exhibits significant improvements across all domains. According to (Shao et al., 2024), DeepseekMath effectively maintains general knowledge and reasoning ability during specialization. Conversely, MAammoTH and WizardMath perform poorly due to just fine-tuning on limited mathematical datasets, resulting in a significant decline in general knowledge and reasoning. The score changes in LLMs suggest that OS is most closely linked to mathematics, followed by CO, and lastly DSA and CN.

In terms of Code, many Code models show significant improvements in DSA (especially knowledge) and OS (especially reasoning), such as CodeLlama and Dolphcoder. This indicates that the disciplines

of DSA and OS are more closely related to code, thus enhancing knowledge and reasoning abilities in these directions, while CO and CN have lower relevance, leading to a decrease in scores. Finally, we observe that the enhancement brought about by small-scale expert LLMs compared to larger-scale LLMs is more pronounced (see CodeLlama-7B/13B, WizardCoder-7B/13B). We attribute this to the supplementary need for specific knowledge and reasoning capabilities in small-scale LLMs, whereas large-scale LLMs already encompass a greater breadth of knowledge and stronger reasoning abilities, resulting in diminishing gains from further training in specific domains.

## 4 RELATED WORK

**Exploration of LLMs in Computer Science.** Given the powerful capabilities of LLMs, recent research has explored their potential applications across various industries and scientific fields, including finance (Zhao et al., 2024), autonomous driving (Huang et al., 2023a; Zhou et al., 2024), robotics (Yuan et al., 2023a; Xiao et al., 2023; Wang et al., 2024a), medicine (Zhou et al., 2023a; Vaid et al., 2024), and chemistry (Guo et al., 2023; Zhang et al., 2024a). Currently, studies exploring LLMs in the field of computer science fall into two main categories. The first category includes broad evaluation benchmarks covering various fields, such as MMLU (Hendrycks et al., 2021a), CMMLU (Li et al., 2024), C-Eval (Huang et al., 2023b), Xiezhi (Gu et al., 2024), and M3KE (Liu et al., 2023a). However, computer science constitutes only a small fraction of these benchmarks, accounting for less than 5% and lacking detailed CS-specific analysis. The second category focuses solely on exploring specific applications of LLMs within computer science, such as network topology (Donadel et al., 2024), cybersecurity (Ferrag et al., 2024; Murtuza, 2024), and software engineering (Marques et al., 2024; Dipongkor, 2024). Nonetheless, there has been a persistent lack of comprehensive evaluation of LLMs' foundational knowledge and reasoning abilities in computer science. To address this gap, we propose CS-Bench and conduct a thorough evaluation of LLMs, providing guidance for understanding and improving their performance in the CS field.

**Evaluation of LLMs' Capabilities.** Evaluating and understanding the capabilities of LLMs is a major focus within the NLP community. Researchers have extensively explored the capabilities of LLMs including planning (Huang et al., 2024), multilingual processing (Lai et al., 2023; Bang et al., 2023), instruction following (Zhou et al., 2023b; Wang et al., 2023b), cross-domain generalization (Wang et al., 2023a; Song et al., 2023; Wang et al., 2024b) and safety (Mazeika et al., 2024; Diao et al., 2024; Mou et al., 2025). Recently, there has been growing interest in LLMs' abilities in mathematics (Frieder et al., 2023; Collins et al., 2023; Wu et al., 2023; Yuan et al., 2023b; Dong et al., 2024; Liu et al., 2024), code programming (Luo et al., 2023b; Rozière et al., 2024; Wang et al., 2024c; Zhang et al., 2024b; Lin et al., 2024), and logical reasoning (Liu et al., 2023b; Saparov et al., 2023; Xu et al., 2023; Wu et al., 2024). While individual capabilities have been well-studied, research on their integrated application and interrelationships remains sparse. Different from (Dong et al., 2024), which investigates interactions between abilities during the supervised fine-tuning phase, we choose computer science as our research context. Given that computer science inherently integrates coding, mathematics, and reasoning, we utilize CS-Bench in this paper to deeply explore the relationship between LLMs' performance in computer science and their mathematical and coding abilities, aiming to advance cross-capability research and integrated analysis of LLM abilities.

## 5 CONCLUSION

In this work, we introduce CS-Bench, the first benchmark specifically designed to systematically analyze the knowledge and reasoning capabilities of mainstream LLMs in the field of computer science. Our evaluation of over 30 models highlights that even the top-performing GPT-4o/OpenAI-o1 has significant room for improvement in computer science. Further score-scale experiments and error type analyses provide directions for enhancing LLMs in the field. Moreover, our investigation into the relationship between computer science, mathematics, and coding demonstrates their close interconnections and provides valuable insights into LLMs' cross-abilities and applications.

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

## ACKNOWLEDGEMENTS

This work was partially supported by:

- The State Key Laboratory of Massive Personalized Customization System and Technology (No. H&C-MPC-2023-02-07(Q)).
- The State Grid Technology Project (5700-202416236A-1-1-ZN) on "Research on Active Semantic Discovery Technology Based on SG-CIM and Its Application in Power Grid Equipment Supply Chain Optimization".
- China Unicom Software Research Institute under the "Framework Agreement for Seven Model Technology Research and Application Demonstration Projects (Software Development for Government Enterprise Content Generation) from 2024 to 2025" (No. 5500331818).
- The National Natural Science Foundation of China (NSFC Nos. 62076031 and 62076036).
- Zhongguancun Academy.

# Appendix

CONTENTS

# A    LIMITATIONS AND DATASET BIAS

In this paper, we introduce CS-Bench, providing a comprehensive evaluation of LLMs and exploring the relationships between model capabilities. However, there are still some limitations to this paper.

(1) Coverage Limitations: Although CS-Bench has made significant strides in comprehensiveness of CS evaluations compared to existing work, given the breadth of computer science, our evaluations cannot cover the entire scope of computer science knowledge. Furthermore, our assessment content focuses on university-level content, examining LLM's mastery of basic subjects in computer science, rather than specific computer science-related research scenarios.

(2) Evaluation Limitations: In the CS-Bench evaluation experiments, we employ GPT-4 scoring to assess generative tasks such as fill-in-the-blank and open-ended tasks. This might lead to certain evaluation thresholds and costs. However, such issues only constitute about 20% of CS-Bench. Additionally, we provide an evaluation scheme that separates comprehension tasks from CS-Bench, allowing for automatic evaluations without the need for GPT-4. We also explore the effect of different scoring models and find that even slightly inferior models produce score rankings consistent with GPT-4, while significantly reducing costs.

(3) Language Limitations: CS-Bench are primarily focused on Chinese, French, German, and English language environments, ensuring comprehensive and in-depth evaluations in these language environments. However, for other environments, its support and coverage are relatively weak, and further optimization and improvement are needed.

The main biases of CS-Bench can be categorized into two aspects:

(1) Difficulty Level: Overall, the benchmark reflects a university-level difficulty.

(2) Task Focus: It emphasizes knowledge- and reasoning-based questions rather than specific real-world production scenarios in computer science.

The impact of difficulty on LLM evaluation is: lower difficulty tends to narrow the score differences between models, while higher difficulty amplifies these differences. It is worth noting that although CS-Bench maintains an overall university-level difficulty, the diversity of difficulty across questions remains significant, ranging from simple definitions in data structures to challenging computer network application problems. This is reflected in the model score range of 39.86% to 72.29%, demonstrating CS-Bench's effectiveness in distinguishing LLMs' capabilities in computer science and presenting challenges even for the best-performing models.

Regarding task scenarios, a key motivation for CS-Bench is that "some prior work has explored LLMs in specific applications such as cybersecurity and software engineering." However, as stated in Section 4, there has been a lack of comprehensive evaluations of LLMs' fundamental knowledge and reasoning capabilities in computer science. Besides, considering the extensive research already conducted by the LLM community on coding abilities, we view programming as a separate focus area. Therefore, CS-Bench does not include programming questions but instead focuses on the relationship between CS capabilities and coding abilities.

# B    BROADEN IMPACT

**Societal Impact.**    CS-Bench is anticipated to play a significant role in the field of computer science. LLMs, trained and evaluated with the aid of CS-Bench, can enhance the work efficiency of relevant professionals, enabling them to complete computer-related tasks, such as code review, error detection, and algorithm optimization, more quickly and accurately. Although this might result in the disappearance of some repetitive jobs, it could also create new career opportunities. In the realm of education, the CS-Bench dataset can serve as an effective teaching tool, assisting teachers in better explaining complex computer science concepts and techniques, and also enabling students to better understand and master this knowledge through practice. However, we should also be cautious of potential risks associated with CS-Bench, such as exam cheating by students in the CS field, which requires management mechanisms and logging to prevent misuse.

**Ethics Statement.**    We ensure adherence to applicable laws and ethical guidelines during the process of data collection, annotation, and usage, providing adequate compensation to all our crowd

workers. As this benchmark pertains to objective knowledge and reasoning in the field of computer science, the annotation content is not influenced by regional or cultural differences among annotators. Moreover, our dataset does not contain any personally identifiable information or offensive content. The authenticity and accuracy of CS-Bench have been thoroughly verified, providing a reliable basis for evaluating LLMs. CS-Bench is intended solely for academic and research purposes. Any commercial use or other misuse deviating from this purpose is strictly prohibited. We urge all users to respect this provision to maintain the integrity and ethical use of this valuable resource.

## C  MORE DETAILS ON CS-BENCH

In C.1, we introduce the details of data collection and processing for CS-Bench. In C.2, we provide a detailed explanation of the design motivation and statistics for CS-Bench. In C.3, we present the distribution of question and answer lengths for each task in CS-Bench. In C.4, we provide a case example for each type under each dimension of CS-Bench.

### C.1  DETAILS OF DATA COLLECTION AND PROCESSING.

Our data mainly comes from three sources: 1. Public channels providing computer science-related questions, such as professional exams and practice tests[5]. 2. Knowledge-based questions manually extracted and adapted by professionals from various academic-permitted CS-related blog posts [6]. 3. Questions constructed from non-public teaching materials and exam papers authorized by the author's affiliated institution. For resources sourced from the internet, we manually extract the knowledge points and questions. For physical materials, we use Optical Character Recognition (OCR) to obtain the data. We then ensure the accuracy of the collected data through manual cross-checking. The data collection was carried out by a team of five students with bachelor's degrees in computer science. We provided each person with adequate wages, significantly above the local minimum wage standard. Based on whether the questions require in-depth reasoning and computation, we label each question as either knowledge-type or reasoning-type. Additionally, we tag each instance with domain and task type labels. For English data, we used GPT-4 to translate the Chinese instances into English, French, and German, followed by manual verification.

### C.2  DETAILED DESIGN MOTIVATION AND STATISTICS OF CS-BENCH

We elaborate on the design motivation of CS-Bench and statistics under each dimension as follows.

**Evaluation Content.**  To ensure comprehensive coverage of fundamental and critical areas in computer science, we select the four most foundational and prevalent domains within the field of computer science as the core content of the CS-Bench dataset. These four domains are as follows: Data Structure and Algorithm, investigating data organization and algorithmic efficiency; Computer Organization, focusing on hardware composition and foundational system operation; Computer Network, involving the analysis of network communication and data transmission; Operating System, delving into system resource management and process control. As depicted in Figure 10 (a), these four disciplines exhibit a roughly uniform distribution. Furthermore, we subdivide the disciplines into 26 granular chapters, allowing CS-Bench to furnish more nuanced evaluation outcomes for models and provide comprehensive guidance for model refinement. We summarize these chapters in Table 5.

**Task Format.**  To better simulate the diverse forms of problems encountered in the real world, we introduce assertion, fill-in-the-blank, and open-ended questions in addition to multiple-choice questions. Specifically, multiple-choice and assertion questions correspond to understanding tasks in CS, while fill-in-the-blank and open-ended questions correspond to generation tasks in CS. Although assessing generation tasks using GPT-4 incurs certain costs, it is important to emphasize that this component represents only a minority (fill-in-the-blank: 10.67%, open-ended: 7.81%), whereas comprehension tasks relying on rule-based scoring constitute the majority (multiple-choice: 61.22%,

---

[5]e.g., https://github.com/CodePanda66/CSPostgraduate-408, https://github.com/ddy-ddy/cs-408

[6]e.g., https://www.wikipedia.org/, https://www.cnblogs.com/, https://www.csdn.net/, https://zhuanlan.zhihu.com

Table 5: Summary of 26 fine-grained subfields of CS-Bench.

| Chatpter | Main Content | Subject | Question Number |
|---|---|---|---|
| Overview | Concepts and elements of data structure, Temporal and spatial complexity... | DSA | 168 |
| Linear List | Linear tables, Sequential tables and Linked lists... | DSA | 276 |
| Stack, Queue,and Array | Shared stack, Circle queue, Arrays,Special matrices... | DSA | 352 |
| String | Concept and operation of strings, Pattern matching of strings... | DSA | 132 |
| Tree | Binary trees, Traversal of trees ans forests, Huffman tree... | DSA | 428 |
| Graph | Concepts of graphs, Traversals of graphs,Application of graphs... | DSA | 368 |
| Searching | Sequential search, Half-split search, Chunked search, Red-black tree, B-tree and B+ tree, Hash search... | DSA | 316 |
| Sorting | Insert Sorting, Swap Sorting, Selection Sorting, Merge Sorting, Heap Sorting, Merge Sorting, Cardinality Sorting, External Sorting Algorithms... | DSA | 356 |
| Overview | Hardware and performance indicators of computers... | CO | 224 |
| Data Representation and Operation | Number system and encoding, Representation and operation of fixed-point numbers and floating-point numbers... | CO | 436 |
| Storage System | Main Memory, External Memory, Cache Memory, Virtual Memory... | CO | 448 |
| Instruction System | Instruction format, Instruction addressing method, CISC and RISC... | CO | 312 |
| Central Processing Unit | Functions of CPU, Instruction execution process, CPU internal bus and data path, CPU hard wiring design and micro programming, Exception and interrupt mechanisms, Instruction pipelines, and multiprocessor concepts... | CO | 488 |
| Bus | Overview of the bus, Bus arbitration, Bus operation and timing, Bus standards... | CO | 268 |
| Input/Output System | I/O interfaces and methods... | CO | 312 |
| Overview and Architecture | Concepts, compositions, functions of computer networks, Architecture and reference models of computer networks... | CN | 296 |
| Physical Layer | Fundamentals of Communication Theory, Transmission Media and Physical Layer Devices... | CN | 328 |
| Data Link Layer | Data frames, Error control, Flow control and Reliable transmission, Media access control, Local and wide area networks, and data link layer devices... | CN | 632 |
| Network Layer | Overview of network layer functions, Routing algorithms, IPv4 and IPv6, Routing protocols, IP multicast, Mobile IP, Router... | CN | 600 |
| Transport Layer | The services provided by the transport layer, UDP and TCP protocols... | CN | 364 |
| Application Layer | Network application model, Domain name system DNS, FTP protocol, World Wide Web, and HTTP... | CN | 408 |
| Overview | Concepts of operating systems, Development and classification of operating systems, Operational mechanisms and architecture of operating systems... | OS | 332 |
| Processes and Threads | Processes and threads, Scheduling of processors, Synchronization and mutual exclusion of processes, Deadlock issues... | OS | 700 |
| Memory Management | Concept of memory management, Concept of virtual memory management, and methods of virtual memory management... | OS | 432 |
| File Management | File systems, Organization and management of disks... | OS | 332 |
| Input/Output Management | I/O devices and control methods, I/O core subsystem, Buffer management... | OS | 368 |

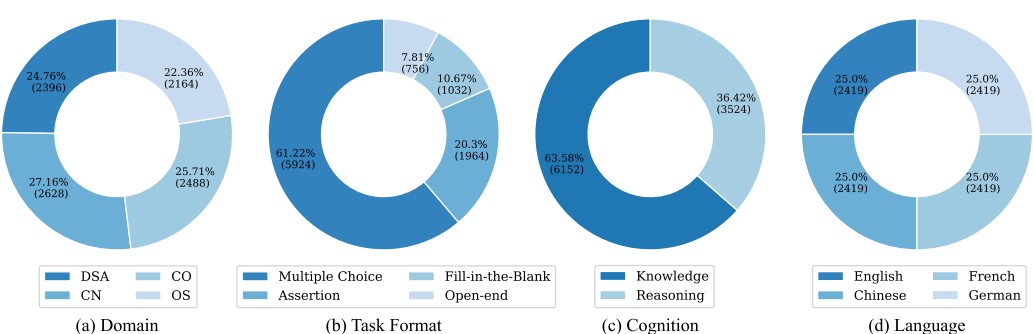

Figure 10: The quantity and proportion of each type in different dimensions on CS-Bench.

assertion: 20.3%). Therefore, if resources are limited, we recommend considering the independent use of understanding tasks from CS-Bench for evaluation purposes.

**Knowledge / Reasoning.** The design goal of CS-Bench is not only to assess the mastery of knowledge in the field of CS but also to evaluate the model's ability to reason using CS knowledge. Therefore, each dataset is labeled with "knowledge" or "reasoning", corresponding to simple questions requiring knowledge recall and challenging questions necessitating knowledge inference, respectively. As shown in Figure 10 (c), knowledge-based questions account for 63.58%, while reasoning-based questions account for 36.42%.

**Language.** To assess the ability of LLMs in addressing CS problems in various linguistic environments, and to adapt CS-Bench for the evaluation of a wider range of LLMs, CS-Bench comprises quadrilingual (English, Chinese, French, and German) data, with each language accounting for 25%.

The other data is obtained through translation by GPT-4, followed by manual verification of processed Chinese data.

## C.3   DISTRIBUTION OF WORD LENGTHS

We compute the distributions of word lengths for questions and answers in CS-Bench (English) across various task formats, as illustrated in Figure 11. For Multiple-Choice questions, the question length includes both the question itself and the four options. Since Multiple-Choice and Assertion questions are comprehension tasks, the answers consist of only one character (A/B/C/D or True/False). For generation tasks, Fill-in-the-blank answers are relatively short, with an average word length of approximately 2, whereas Open-ended questions typically yield longer answers as they entail detailed explanatory processes.

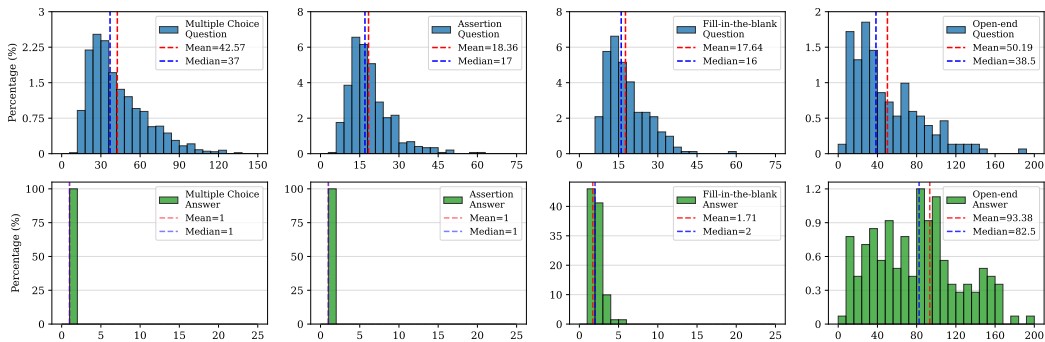

Figure 11: Question and answer lengths of each task format in CS-Bench (English).

## C.4   CS-BENCH EXAMPLES

We present samples of knowledge and reasoning types in Table 6, samples from various domains in Table 7, samples of different task formats in Table 8, and samples from different languages in Table 9.

Table 6: Examples of knowledge-type and reasoning-type.

| Type | Example |
|---|---|
| Knowledge | **Question:**
The three fundamental elements of data structure include ().
A: Logical structure, storage structure, operations on data.
B: Logical structure, algorithm design, program implementation.
C: Data types, data storage, data manipulation.
D: Data Definition, Data Implementation, Data Manipulation.
**Answer:**
A
**Analysis:**
None |
| Reasoning | **Question:**
The time complexity of a certain algorithm is $O(n^2)$, indicating that the algorithm's ().
A: The problem size is $O(n^2)$.
B: Execution time equals $O(n^2)$.
C: The execution time is directly proportional to $O(n^2)$.
D: The problem size is directly proportional to $O(n^2)$.
**Answer:**
C
**Analysis:**
The time complexity is $O(n^2)$, which means the time complexity $T(n)$ satisfies $T(n) \leq c * n^2$ (where $c$ is a proportionality constant), that is, $T(n) = O(n^2)$. The time complexity $T(n)$ is a function of the problem size $n$, and the problem size remains $n$, not $n^2$. |

Table 7: Examples of samples in different domains.

| Domain | Example |
|---|---|
| Data Structure and Algorithm | **Question:**
The correct statement about data structures is ().
A: The logical structure of data is independent of its storage structure.
B: The storage structure of data is independent of its logical structure.
C: The logical structure of data uniquely determines its storage structure.
D: The data structure is determined solely by its logical structure and storage structure.
**Answer:**
A
**Analysis:**
The logical structure of data is approached from the perspective of practical problems, using only abstract expressions and is independent of the various choices of data storage methods. The storage structure of data is the mapping of the logical structure on a computer, and it cannot exist independently of the logical structure. Data structure includes three essential elements, all of which are indispensable. |
| Computer Organization | **Question:**
A complete computer system should include ().
A: Arithmetic Logic Unit (ALU), Memory, Control Unit
B: Peripheral devices and host computer
C: Host and Application
D: The accompanying hardware devices and software systems
**Answer:**
D
**Analysis:**
A is a component of the computer host, while B and C only involve parts of the computer system and are both incomplete. |
| Computer Network | **Question:**
The most basic function of computer networks is ().
A: Data Communication
B: Resource Sharing
C: Distributed Processing
D: Information Synthesis Processing
**Answer:**
A
**Analysis:**
The functions of computer networks include: data communication, resource sharing, distributed processing, integrated information processing, load balancing, enhancing reliability, etc. However, the most fundamental function is data communication, which is also the basis for realizing other functions. |
| Operating System | **Question:**
Among the following options, () is not an issue of concern for the operating system.
A: Manage bare-metal computers
B: Design and provide an interface between user programs and hardware systems
C: Manage computer system resources
D: Compiler for High-Level Programming Languages
**Answer:**
D
**Analysis:**
The operating system manages computer software/hardware resources, expands the bare machine to provide a more powerful extended machine, and acts as an intermediary between the user and the hardware. Clearly, the compiler for high-level programming languages is not a concern of the operating system. The essence of a compiler is a set of program instructions that are stored in the computer. |

Table 8: Examples of different task formats.

| Type | Example |
|------|---------|
| Multiple Choice | **Question:**
Given that the storage space for a circular queue is the array A[21], with front pointing to the position before the head element and rear pointing to the tail element, assuming the current values of front and rear are 8 and 3, respectively, the length of the queue is ().
A: 5
B: 6
C: 16
D: 17
**Answer:**
C
**Analysis:**
The length of the queue is (rear - front + maxsize) % maxsize = (rear - front + 21) % 21 = 16. This situation is the same as when front points to the current element and rear points to the next element after the last element in the queue. |
| Assertion | **Question:**
In a directed graph with n vertices, the degree of each vertex can reach up to 2n.
**Answer:**
False.
**Analysis:**
In a directed graph, the degree of a vertex is equal to the sum of its in-degree and outdegree. In a directed graph with n vertices, any given vertex can have at most one pair of oppositely directed edges connecting it with each of the other n-1 vertices. |
| Fill-in-the-blank | **Question:**
In a sequential list of length n, when deleting the ith ($1 \leq i \leq n$) element, () elements need to be moved forward.
**Answer:**
n-i
**Analysis:**
The elements from a[i+1] to a[n] need to be moved forward by one position, involving the movement of n-(i+1)+1=n-i elements. |
| Open-ended | **Question:**
Given that the 9th level of a complete binary tree has 240 nodes, how many nodes does the entire complete binary tree have? How many leaf nodes are there?
**Answer:**
In a complete binary tree, if the 9th level is full, then the number of nodes = $2^{(9-1)} = 256$. However, currently, there are only 240 nodes on the 9th level, indicating that the 9thlevel is not full and is the last level. Levels 1 to 8 are full, so the total number of nodes = $2^8 + 240 = 495$. Since the 9th level is the last level, all nodes on the 9th level are leaf nodes. Moreover, the parents of the 240 nodes on the 9th level are on the 8th level, with the number of parents being 120, which means there are 120 branch nodes on the 8th level, and the rest are leaf nodes. Therefore, the number of leaf nodes on the 8th level is $2^{(8-1)} - 120 = 8$. Consequently, the total number of leaf nodes = 8 + 240 = 248.
**Analysis:**
None |

Table 9: Examples of different languages.

| Type | Example |
|---|---|
| English | **Question:**
For a linear list with sequential storage, the operation with a time complexity of $O(1)$ should be ().
A: Sort n elements in ascending order.
B: Remove the i-th ($1 \le i \le n$) element.
C: Change the value of the i-th element ($1 \le i \le n$).
D: Insert a new element after the i-th ($1 \le i \le n$) element.
**Answer:**
C
**Analysis:**
The time complexity for sorting n elements is at least O(n) (when initially ordered), and typically $O(n \log_2 n)$ or $O(n^2)$. Options B and D are clearly incorrect. Sequential lists support random access by index. |
| Chinese | **Question:**
对于顺序存储的线性表，其算法时间复杂度为$O(1)$的运算应该是()。
A: 将n个元素从小到大排序
B: 删除第i ($1 \le i \le n$)个元素
C: 改变第i ($1 \le i \le n$)个元素的值
D: 在第i ($1 \le i \le n$)个元素后插入个新元素
**Answer:**
C
**Analysis:**
对n个元素进行排序的时间复杂度最小也要$O(n)$（初始有序时）通常为$O(n \log_2 n)$或$O(n^2)$。B和D显然错误。顺序表支持按序号的随机存取方式。 |
| French | **Question:**
Pour une liste linéaire stockée de manière séquentielle, l'opération dont la complexité temporelle algorithmique est $O(1)$ devrait être ().
A: Trier n éléments dans l'ordre croissant.
B: Supprimer le $i$-ème ($1 \le i \le n$) élément.
C: Modifier la valeur du ième élément ($1 \le i \le n$).
D: Insérez un nouvel élément après le i-ème ($1 \le i \le n$) élément.
**Answer:**
C
**Analysis:**
La complexité temporelle pour trier $n$ éléments est au minimum $O(n)$ (lorsque la liste est initialement ordonnée), généralement $O(n \log_2 n)$ ou $O(n^2)$. Les options B et D sont manifestement incorrectes. Une liste séquentielle supporte l'accès aléatoire par indice. |
| German | **Question:**
Für sequentiell gespeicherte lineare Listen, welche Operation hat eine algorithmische Zeitkomplexität von $O(1)$? ().
A: Sortiere n Elemente der Größe nach von klein nach groß.
B: Lösche das $i$-te ($1 \le i \le n$) Element .
C: Ändere den Wert des i-ten Elements ($1 \le i \le n$).
D: Füge nach dem i-ten ($1 \le i \le n$) Element ein neues Element ein.
**Answer:**
C
**Analysis:**
Die Zeitkomplexität für das Sortieren von $n$ Elementen beträgt mindestens $O(n)$ (wenn sie anfänglich sortiert sind) und ist normalerweise $O(n \log_2 n)$ oder $O(n^2)$. B und D sind offensichtlich falsch. Eine sequentielle Liste unterstützt den zufälligen Zugriff nach Index. |

# D MORE DETAILS ON EXPERIMENT SETUP

In D.1, we present the question templates used to prompt models for each type of task. In D.2, we show the prompts used for GPT-4 to score models' answers to fill-in-the-blank and open-ended questions, and validate the effectiveness of GPT-4's automatic scoring through consistency experiments with human scoring. We also explore the effect of different scoring models. In D.3, we detail the experimental environment used to implement model inference. In D.4, we introduce all the evaluated model families.

## D.1 DETAILS OF TEMPLATE FOR EACH TASK FORMAT

We present the templates for querying LLMs with various question formats in Table 10.

Table 10: Prompt Templates for asking various questions to LLMs.

| Type | Prompt Template |
|------|-----------------|
| Multiple Choice | This is a multiple-choice question. Please read the question carefully and choose the correct answer. Question: <Question> 
 Which one of the following options is correct? Options: 
 (A) <A> 
 (B)  
 (C) <C> 
 (D) <D> 
 Please provide the answer to this question directly (a single letter): |
| Assertion | This is a true/false question. Please determine whether the following statement is true or false. Statement: <Question> 
 Please give the answer directly (true or false): |
| Fill-in-the-blank | You are a professor proficient in computer science. This is a fill-in-the-blank question. Give answers to the following question without explanation or repeating it. 
 Question: <Question> 
 Answer: |
| Open-ended | This is a subjective Question: <Question> 
 Please provide a brief answer to this question: |

## D.2 DETAILS OF GPT-4 SCORING

**GPT-4 Scoring Prompt.** In Table 13, we present the prompts utilized to instruct GPT-4 in scoring the outputs of LLMs in CS generation tasks, encompassing both FITB and Open-ended questions.

**Consistency between GPT-4 Scoring and Manual Scoring.** To assess the effectiveness of GPT-4 scoring in evaluating LLM responses, we conduct a consistency experiment between GPT-4 prediction scores and manual scores. For Fill-in-the-blank and Open-ended types, we randomly sample 100 instances from the GPT-4 scoring samples and employ three human annotators to score these predicted results. These three annotators all hold bachelor's degrees in computer science. Their scoring criteria were consistent with the evaluation standards provided to GPT-4 and can be found in Table 13. In Table 11, we report the consistency scores among human annotators (measured by Cronbach's alpha), as well as the consistency scores between the average human annotation scores and GPT-4 scoring (measured by Pearson correlation coefficient). The excellent consistency between human and GPT-4 scores validates the effectiveness of GPT-4 scoring.

Table 11: Consistency between GPT-4 scoring and human scoring.

| Type | Annotation Count | Consistency | |
|------|------------------|-------------|-------------|
| | | Human-GPT4 | Human-Human |
| Fill-in-the-blank | 100 | 0.808 | 0.9311 |
| Open-ended | 100 | 0.9494 | 0.9751 |

**Effect of Different Scoring Models** To investigate the impact of different models acting as judges, as well as the relationship between the tested model and the judge's capabilities on the robustness of the evaluation process, we first use the Claude family to construct a set of models with varying abilities: Claude3-Haiku < Claude3-Sonnet < Claude3-Opus < Claude3.5-Sonnet (following the official Claude capability ranking (Anthropic, 2024b)). Next, we pair each two models respectively as the judge and the tested model. The results are as shown in Table 12.

Table 12: Performance of tested models under different scoring models.

| Model | Claude3-Jaiku (Judge) | Claude3-Sonnet (Judge) | Claude3-Opus (Judge) | Claude3.5-Sonnet (Judge) | GPT-4 (Judge) |
|---|---|---|---|---|---|
| Claude3-Haiku (Test) | 64.61 | 64.56 | 63.22 | 62.45 | 60.69 |
| Claude3-Sonnet (Test) | 64.33 | 63.75 | 62.13 | 62.05 | 60.17 |
| Claude3-Opus (Test) | 71.74 | 71.29 | 70.46 | 69.97 | 68.63 |
| Claude3.5-Sonnet (Test) | 73.97 | 73.46 | 72.43 | 72.27 | 71.01 |

Firstly, when all Claude models act as judges, their scoring of the tested models aligns consistently with GPT-4. This effectively demonstrates the robustness of the evaluation process, as consistent ranking results are obtained regardless of whether the tested model's performance is inferior or superior to that of the judge model (Claude3-Haiku and Claude3.5-Opus representing the two extremes of judge model capabilities). We believe that as long as a model possesses a certain level of discernment, it can serve as a judge, without needing to outperform all tested models. Even the least capable model, Claude3-Haiku, can yield score rankings consistent with GPT-4 during evaluation, while costing only $0.25 / 1.25 per million tokens for input/output, respectively, significantly reducing evaluation costs. Secondly, we observe that stronger models tend to be more stringent in their scoring. This is reflected in the assessment scores for all tested models, consistently following the order: Claude3-Haiku < Claude3-Sonnet < Claude3-Opus < Claude3.5-Sonnet. We believe this occurs because more capable evaluation models can identify deeper levels of error. Lastly, we find that the models score fairly and do not tend to give themselves disproportionately high scores (as seen when the same model acts as both judge and test model).

### D.3 DETAILS OF INFERENCE IMPLEMENTATION

For all open-source models, we utilize a cluster with 8 NVIDIA A100-80GB GPUs to run the inference, and we use vLLM (Kwon et al., 2023) for inference acceleration, applying the corresponding chat templates and the same hyper-parameters: batch size=1, temperature=0, top-p=1.0, and max_tokens=2048. For all closed-source models with API access, we also adopt the generation scheme with temperature=0, and simply run the inference with CPUs, which typically completes within a day. During the evaluation of GPT-4, we also applied the setting of temperature=0. To avoid error bias, we conducted the experiments 3 times and took the average of the scores. For models supporting web search or tool calls, we disable these features to ensure a fair comparison.

### D.4 DETAILS OF THE MODELS BEING EVALUATED

**Gemma** (Team et al., 2024) is a family of lightweight, open models from Google, built from the same research and technology used to create the Gemini models. They are text-to-text, decoder-only large language models, available in English, with open weights, pre-trained variants, and instruction-tuned variants. The Gemma model excels on academic benchmarks in language understanding, reasoning, and security. Gemma publishes models in two sizes (2 billion and 7 billion parameters) .

**Llama2** (Touvron et al., 2023) is an upgraded version of Llama developed by MetaAI. It utilizes more robust data cleaning and mixing techniques, and up-samples sources closest to factual information, which can enhance knowledge and reduce hallucinations. Additionally, it employs Grouped-Query Attention technology to lessen reliance on memory.

**Llama3** (Meta, 2024) is the latest generation of large language models developed by MetaAI. The training dataset for Llama 3 is seven times larger than that used for Llama 2, with the amount of code included being four times that of Llama 2. Compared to previous versions of the model, it has seen a tremendous enhancement in reasoning, code generation, and instruction following capabilities.

Table 13: Scoring Prompts for Fill-in-the-blank and Open-ended Questions.

| Type | Prompt Template |
|---|---|
| Fill-in-the-blank | You are now a teaching assistant. As a TA, your task is to grade the fill-in-the-blank assignments of computer science students.
You will see the standard answer for each question (these answers are verified and completely correct), and you need to score the students' answers based on this.
If the student's answer conveys the same meaning as the standard answer or other answers (different formats are also considered correct), then award 1 point; if not, then 0 points.
Question: \<question\>
Standard Answer: \<correct_answer\>
Other Answers: \<other_answers\>
Student Response: \<predict_output\>
Score (0 or 1): |
| Open-ended | You are now serving as a teaching assistant. In this role, your task is to grade the subjective homework assignments of computer science students. You will be presented with the standard answers for each question (which are verified and completely correct), and you must use these to score the students' responses. The grading scale ranges from 1 to 10 points, with 10 being the highest and 1 being the lowest. When grading, please take into consideration the accuracy, relevance, completeness, and depth of thought of the answers. Scores should be assigned based on the following *criteria*:

First Tier: 1-3 points
Accuracy: The answer contains several fundamental errors, showing limited understanding. Relevance: The answer has low relevance to the question and standard answer, with most content straying from the requirements. Completeness: The answer omits multiple key points, failing to cover the main aspects of the question.

Second Tier: 4-6 points
Accuracy: There are some errors in the answer, although most of the basic concepts are understood correctly.
Relevance: The answer is generally relevant to the question and standard answer, but some content does not fully conform to the requirements. Completeness: The answer is fairly complete, but lacks some important details or certain key points are not fully elaborated.

Third Tier: 7-8 points
Accuracy: The answer is almost entirely correct, with only very minor errors. Relevance: The answer is highly relevant to the question and standard answer, focused and with almost no deviation from the topic. Completeness: The answer is comprehensive and detailed, covering all key aspects very well.

Fourth Tier: 9-10 points
Accuracy: The answer is free of any errors, demonstrating a deep understanding and precise grasp of the issue. Relevance: The answer is in complete accordance with the requirements, strictly aligned with the question and standard answer, without any deviation. Completeness: The answer is structured rigorously, logically organized, and systematically covers all aspects of the question.

Grading Guide: When assigning a score, please first make a preliminary assessment of accuracy based on the student's answer compared to the standard answer. Then, consider the relevance and completeness to determine the final score. Ensure that each point awarded is based on a fair and justified comprehensive evaluation. |

**Llama3.1-405B** (Dubey et al., 2024) is an auto-regressive language model developed by MetaAI, utilizing an optimized transformer architecture. Trained on over 15 trillion tokens, it demonstrates excellent flexibility, control, and advanced capabilities. Llama3.1-405B is the largest openly available foundation model released by MetaAI.

**Llama3-Chinese** (Wang & Zheng, 2024) is an instruction-tuned language model for Chinese and English users with various abilities such as roleplaying and tool-using built upon the Meta-Llama-3-8B-Instruct model.

**ChatGLM3** (Zeng et al., 2022) is a next-generation conversational pre-trained model jointly released by Zhipu AI and KEG Lab of Tsinghua University. ChatGLM3-6B adopts a newly designed Prompt format, in addition to regular multi-turn dialogue. It also natively supports complex scenarios such as function call, code interpretation.

**Baichuan2** (Yang et al., 2023) is a large-scale multilingual model developed by Baichuan Company. It adopts several advanced techniques in its design and training process, including Rotary Position Embedding, a novel position encoding technique, SwiGLU activation function, and memory efficient attention mechanism. Compared with Baichuan1, its performance has been greatly improved.

**InternLM2** (Cai et al., 2024) is an open-source large-scale language model developed by Shanghai AI Laboratory. This model has good processing ability for ultra long texts and adopts COOL RLHF technology. It solves human preference conflicts through a conditional reward model and performs multiple rounds of online RLHF to improve the model's alignment ability.

**Qwen1.5** (Bai et al., 2023) is a family of language models developed by Alibaba. It has features such as SwiGLU activation, attention QKV bias, group query attention, mixture of sliding window attention and full attention, etc. Qwen 1.5 series models have strong basic capabilities including language understanding.

**Mistral-7B** (Jiang et al., 2023), a 7-billion-parameter language model designed for superior performance and efficiency, which is developed by Mistral AI. Mistral 7B leverages Packet Query Attention (GQA) for faster inference, combined with Sliding Window Attention (SWA) to efficiently process sequences of arbitrary length while reducing inference costs.

**Mixtral-8×7B** (Jiang et al., 2024) is a Sparse Mixture of Experts (SMoE) language model developed by Mistral AI. Its architecture is the same as that of the Mistral 7B, except that each layer consists of 8 feedforward blocks (i.e., experts). Mixtral has demonstrated exceptional abilities in math, code generation, and tasks that require multilingual understanding.

**Mistral-Large-123B** (MistralAI, 2024) is a language model developed by Mistral AI with a 128K context window, supporting dozens of languages including French, German, Spanish, Italian, Russian, Chinese, Japanese, and Korean, as well as over 80 programming languages. It is designed for single-node inference with long-context applications in mind – its size of 123 billion parameters allows it to run at large throughput on a single node.

**DeepSeekLLM** (Bi et al., 2024) is a family of models released by DeepSeek-AI, and its core architecture borrows from the Llama model. This family of models employs Multi-Head Attention (MHA) and Group Query Attention (GQA) techniques, which significantly enhance their performance and efficiency. Furthermore, DeepSeekLLM demonstrates strong bilingual capabilities in both Chinese and English.

**PaLM-2** (Anil et al., 2023) is the higher-performance successor to PaLM released by Google, which differs in terms of dataset mixing. Compared to the first-generation PaLM version, it uses a smaller model but performs more training calculations. It also relies on more diverse pre-training targets.

**Claude2** Claude2.1(Anthropic, 2023) and Claude3 (Anthropic, 2024a) are AI models developed by Anthropic, showcasing advanced language understanding and generation capabilities. Utilizing the constitutional AI framework, Claude models are designed to ensure helpfulness and trustworthiness. Claude-3.5-Sonnet (Anthropic, 2024b) is the first product in the Claude-3.5 model series developed by Anthropic. It demonstrates significant improvements in understanding nuance, humor, and complex instructions, and excels at generating high-quality content with a natural, relatable tone. It possesses

exceptional coding capabilities, allowing it to independently write, edit, and execute code, along with advanced reasoning and troubleshooting skills. Additionally, it is Anthropic's most powerful visual model to date.

**GPT**   GPT-3.5 (OpenAI, 2022), GPT-4 (Achiam et al., 2023) and GPT-4o (OpenAI, 2024a), released by OpenAI, are part of the GPT-series models enhanced by a three-stage reinforcement learning with human feedback (RLHF) algorithm. This algorithm not only improves the models' ability to follow instructions but also significantly reduces the generation of harmful or toxic content. Additionally, GPT-4 supports image inputs and achieves human-level performance on various benchmarks. GPT-4o, the latest model developed by OpenAI, boasts powerful real-time reasoning, language interaction, and multimodal capabilities.

**OpenAI-o1**   (OpenAI, 2024b) is a new large language model developed by OpenAI, trained with reinforcement learning to handle complex reasoning. It can produce a long internal chain of thought before responding to the user, and o1 refines its chain of thought and improves its strategies by learning to recognize and correct mistakes, break down complex steps into simpler ones, and adopt alternative approaches when the current method fails. It currently has two versions: o1-preview, which has strong reasoning capabilities and broad world knowledge, and o1-mini, a lightweight version with faster reasoning speed.

**GLM-4**   (AI, 2024) is a new generation base large model developed by Zhipu AI. It has strong tool calling and multi-modal capabilities, as well as strong mathematical reasoning ability and code generation ability.

**ERNIE**   (Baidu, 2023) ERNIE3.5 and ERNIE4 are large language models developed by Baidu. ERNIE3.5 is capable of processing text data in multiple languages and has a good understanding and representation ability for entities and relationships in text. Ernie 4 has adopted more advanced knowledge graph information and more advanced knowledge integration technology, further improving the performance of the model.

# E    MORE DETAILS ON EXPERIMENT

In E.1, we present detailed performance of the models on CS-Bench (EN), including the leaderboard, task formats, and domains. In E.2, we describe and validate the design of the scale-score fitting function. In E.3, we compare models' performance across different contexts, with a focus on evaluating and analyzing the model's performance on CS-Bench (CN). In E.4, we conduct case studies to better understand the specific details of the models' failures on CS-Bench. In E.5, We further analyze and interpret the relationship phenomena among Code, Math, and CS abilities from internal representations and data characteristics. In E.6, based on our findings, we explore and experiment with several specific approaches to enhance CS capabilities. In E.7, we analyze examples from OpenAI-o1 and compare them with GPT-4o.

## E.1    DETAILS OF MODEL PERFORMANCE

**The Leaderboard on CS-Bench.**    We visualize the results of LLMs on CS-Bench in Figure 12.

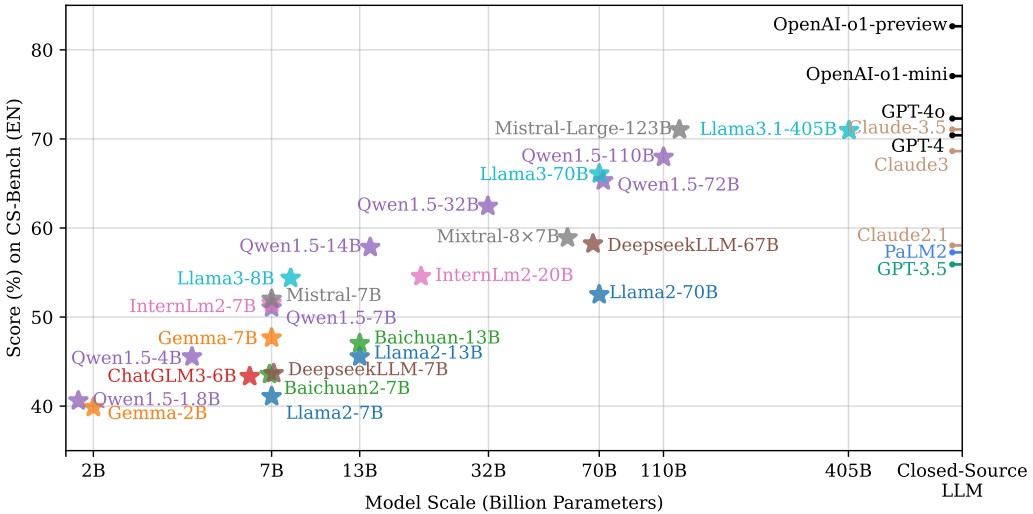

Figure 12: The leaderboard of LLMs on CS-Bench (EN).

**Detailed Performance on Each Task Format.**    We present models' performance on four types of tasks in Table 14 and visualize the results in Figure 13.

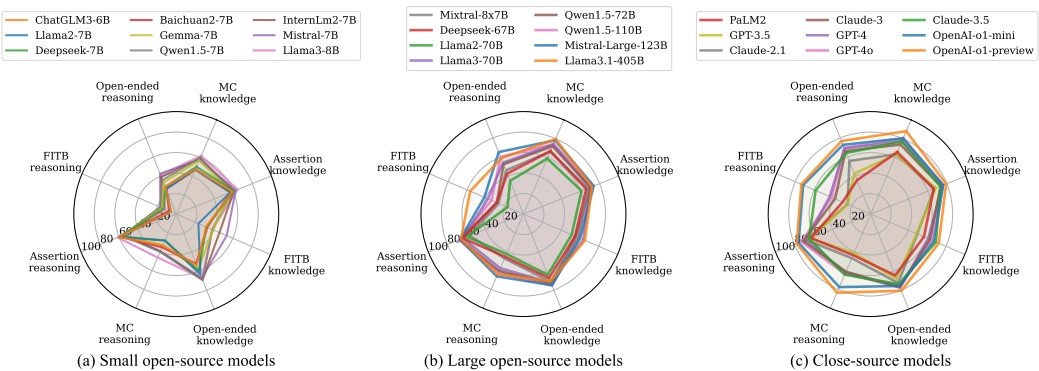

(a) Small open-source models          (b) Large open-source models          (c) Close-source models

Figure 13: Performance of various LLMs for each ability dimension about task formats.

**Detailed Performance on Each Subfield.**    In Figure 14, we visualize the models' knowledge and reasoning performance across the four domains of CS-Bench. Subsequently, we focus on the models' performance in 26 fine-grained subfields. Table 15 presents the results of eight representative models.

Table 14: Zero-shot scores (%) of LLMs across question formats on CS-Bench (EN).

| Model | Multiple-choice | | | Assertion | | | Fill-in-the-blank | | | Open-ended | | | All | | |
|---|---|---|---|---|---|---|---|---|---|---|---|---|---|---|---|
| | Klg | Rng | Avg | Klg | Rng | Avg | Klg | Rng | Avg | Klg | Rng | Avg | Klg | Rng | Avg |
| Random | 25.00 | 25.00 | 25.00 | 50.00 | 50.00 | 50.00 | 0.00 | 0.00 | 0.00 | 10.00 | 10.00 | 10.00 | 27.4 | 24.12 | 26.20 |
| *Open-source LLM (Scale < 10B)* | | | | | | | | | | | | | | | |
| Gemma-2B | 46.87 | 25.85 | 38.74 | 52.58 | 48.48 | 51.64 | 34.12 | 7.55 | 27.68 | 49.60 | 26.02 | 32.96 | 46.89 | 27.59 | 39.86 |
| Qwen1.5-4B | 53.00 | 35.47 | 46.22 | 56.84 | 58.59 | 57.24 | 29.41 | 11.32 | 25.02 | 55.40 | 28.58 | 36.47 | 51.18 | 35.70 | 45.54 |
| ChatGLM3-6B | 47.51 | 33.07 | 41.92 | 58.97 | 60.61 | 59.34 | 31.76 | 5.66 | 25.43 | 53.80 | 28.94 | 36.25 | 48.63 | 34.07 | 43.33 |
| Llama2-7B | 47.00 | 28.06 | 39.67 | 56.84 | 60.61 | 57.70 | 23.53 | 5.66 | 19.20 | 63.80 | 26.19 | 37.25 | 47.15 | 30.48 | 41.08 |
| DeepseekLLM-7B | 50.19 | 28.06 | 41.63 | 60.49 | 58.59 | 60.06 | 31.76 | 13.21 | 27.26 | 59.80 | 28.67 | 37.83 | 50.87 | 31.11 | 43.67 |
| Baichuan2-7B | 47.51 | 35.27 | 42.77 | 57.14 | 59.60 | 57.70 | 32.94 | 7.55 | 26.78 | 52.40 | 26.90 | 34.40 | 48.29 | 35.33 | 43.57 |
| Gemma-7B | 56.70 | 33.07 | 47.56 | 58.05 | 57.58 | 57.94 | 38.82 | 15.09 | 33.06 | 58.20 | 33.36 | 40.67 | 54.90 | 35.02 | 47.66 |
| Qwen1.5-7B | 59.90 | 40.08 | 52.23 | 58.97 | 56.57 | 58.42 | 38.24 | 16.98 | 33.08 | 69.60 | 35.75 | 45.71 | 57.34 | 40.08 | 51.05 |
| InternLm2-7B | 59.26 | 39.48 | 51.61 | 60.49 | 55.56 | 59.36 | 45.88 | 15.09 | 38.41 | 69.00 | 39.03 | 47.84 | 58.31 | 39.77 | 51.56 |
| Mistral-7B | 57.34 | 39.68 | 50.51 | 62.61 | 54.55 | 60.77 | 53.53 | 16.98 | 44.66 | 67.40 | 42.39 | 49.75 | 58.63 | 40.44 | 52.01 |
| Llama3-8B | 61.81 | 46.09 | 55.73 | 64.44 | 61.62 | 63.80 | 38.24 | 11.32 | 31.71 | 67.60 | 41.33 | 49.06 | 59.75 | 44.97 | 54.37 |
| *Open-source LLM (Scale > 10B)* | | | | | | | | | | | | | | | |
| Llama2-13B | 50.06 | 33.87 | 43.79 | 55.93 | 56.57 | 56.08 | 44.71 | 22.64 | 39.36 | 62.00 | 29.65 | 39.16 | 51.31 | 35.46 | 45.54 |
| Baichuan-13B | 53.00 | 37.68 | 47.07 | 58.66 | 53.54 | 57.49 | 35.88 | 16.98 | 31.30 | 59.80 | 31.15 | 39.58 | 52.53 | 37.44 | 47.03 |
| Qwen1.5-14B | 64.62 | 50.70 | 59.23 | 62.61 | 59.60 | 61.92 | 51.76 | 28.30 | 46.07 | 70.60 | 43.45 | 51.44 | 62.79 | 49.18 | 57.83 |
| InternLm2-20B | 62.20 | 43.69 | 55.04 | 61.09 | 62.63 | 61.44 | 51.18 | 24.53 | 44.72 | 67.20 | 36.02 | 45.19 | 60.81 | 43.66 | 54.56 |
| Qwen1.5-32B | 70.63 | 57.92 | 65.71 | 63.53 | 62.63 | 63.32 | 53.53 | 22.64 | 46.04 | 73.20 | 48.76 | 55.95 | 66.87 | 54.72 | 62.45 |
| Mixtral-8×7B | 66.28 | 47.09 | 58.85 | 67.78 | 56.57 | 65.22 | 58.24 | 26.42 | 50.52 | 71.00 | 45.93 | 53.30 | 65.91 | 46.66 | 58.90 |
| DeepseekLLM-67B | 66.92 | 45.29 | 58.55 | 65.96 | 63.64 | 65.43 | 54.71 | 28.30 | 48.30 | 67.20 | 42.57 | 49.81 | 65.23 | 45.96 | 58.22 |
| Llama2-70B | 58.88 | 42.28 | 52.46 | 61.09 | 59.60 | 60.75 | 51.18 | 16.98 | 42.88 | 63.80 | 34.96 | 43.44 | 58.73 | 41.68 | 52.52 |
| Llama3-70B | 73.95 | 57.52 | 67.59 | 69.91 | 63.64 | 68.48 | 63.53 | 37.74 | 57.27 | 72.00 | 53.98 | 59.28 | 71.65 | 56.36 | 66.08 |
| Qwen1.5-72B | 72.03 | 60.32 | 67.50 | 70.52 | 66.67 | 69.64 | 55.29 | 28.30 | 48.74 | 73.00 | 52.30 | 58.39 | 69.63 | 57.75 | 65.31 |
| Qwen1.5-110B | 74.33 | 62.73 | 69.84 | 73.25 | 67.68 | 71.98 | 57.06 | 33.96 | 51.46 | 75.20 | 60.00 | 64.47 | 71.98 | 60.91 | 67.95 |
| Mistral-Large-123B | 78.29 | 66.13 | 73.58 | 73.86 | 66.67 | 72.22 | 60.00 | 41.51 | 55.52 | 74.80 | 65.13 | 67.97 | 74.84 | 64.37 | 71.03 |
| Llama3.1-405B | 79.57 | 63.93 | 73.52 | 72.04 | 65.66 | 70.58 | 65.29 | 56.60 | 63.18 | 70.80 | 58.58 | 62.17 | 75.64 | 62.81 | 70.96 |
| *Closed-source LLM* | | | | | | | | | | | | | | | |
| PaLM-2 | 65.91 | 43.66 | 57.30 | 66.36 | 62.77 | 65.54 | 56.52 | 29.79 | 50.04 | 64.47 | 35.64 | 44.12 | 64.85 | 44.01 | 57.26 |
| Claude-2.1 | 63.47 | 46.89 | 57.05 | 66.87 | 67.68 | 67.06 | 49.41 | 24.53 | 43.38 | 72.40 | 55.84 | 60.71 | 62.97 | 49.42 | 58.04 |
| Claude-3-Opus | 73.82 | 61.32 | 68.98 | 73.56 | 70.71 | 72.91 | 62.94 | 37.74 | 56.83 | 76.73 | 66.11 | 69.23 | 72.57 | 61.75 | 68.63 |
| GPT-3.5 | 63.35 | 41.48 | 54.89 | 68.39 | 63.64 | 67.30 | 48.82 | 24.53 | 42.93 | 68.00 | 42.65 | 50.11 | 63.04 | 43.45 | 55.91 |
| GPT-4 | 77.27 | 62.32 | 71.48 | 75.38 | 67.68 | 73.62 | 61.18 | 43.40 | 56.87 | 77.40 | 68.94 | 71.43 | 74.85 | 62.66 | 70.41 |
| Claude-3.5-Sonnet | 77.14 | 64.33 | 72.18 | 73.25 | 64.65 | 71.28 | 69.41 | 58.49 | 66.76 | 73.60 | 64.16 | 66.94 | 75.13 | 63.97 | 71.07 |
| GPT-4o | 80.08 | 63.73 | 73.75 | 75.68 | 72.73 | 75.01 | 64.71 | 41.51 | 59.08 | 75.20 | 69.47 | 71.16 | 76.95 | 64.15 | 72.29 |
| *Special Reasoning LLM* | | | | | | | | | | | | | | | |
| OpenAI-o1-mini | 80.46 | 77.56 | 79.34 | 76.29 | 75.76 | 76.17 | 67.06 | 71.70 | 68.19 | 75.60 | 72.74 | 73.58 | 77.60 | 76.12 | 77.06 |
| OpenAI-o1-preview | 87.99 | 83.57 | 86.28 | 79.33 | 77.78 | 78.98 | 72.35 | 73.58 | 72.65 | 80.60 | 76.73 | 77.87 | 83.61 | 80.98 | 82.65 |

Figure 14: Performance of various LLMs for each ability dimension about CS domains.

Firstly, we can observe significant variations in scores across different subfields within the same domains for the models. Taking the DSA domain as an example, Llama2-70B scores range from 45.44% to 76.67% across different chapters (average 56.93%), while GPT-3.5 scores range from 55.17% to 80.00% (average 60.67%). Secondly, the performance of different models in the same subfield is generally consistent compared to the average scores. For instance, all models perform above the average scores in the "Overview" and "Stack, Queue, and Array" subfields of DSA but below average in the "Tree" and "Graph" subfields. These detailed scores allow us to understand which content poses greater challenges for the models and provides guidance for improving the models' performance in computer science by strengthening these weaker subfields.

We further observe that although the overall scores of models from the same family increase with scale, not all chapters follow this pattern. As shown in Figure 15, the Llama2 series exhibits a trend

Table 15: Detailed scores of models on fine-grained subfields.

| Content | Llama2-7B | Llama2-13B | Llama2-70B | Mixtral-8×7B | Llama3-8B | Llama3-70B | GPT-3.5 | GPT-4 |
|---|---|---|---|---|---|---|---|---|
| *Data Structure and Algorithm* | | | | | | | | |
| Overview | 56.67 | 51.11 | 59.44 | 68.06 | 73.33 | 68.06 | 71.11 | 74.17 |
| Linear List | 34.48 | 44.83 | 53.45 | 58.62 | 53.45 | 65.52 | 55.17 | 67.24 |
| Stack, Queue, and Array | 49.61 | 50.91 | 57.40 | 57.66 | 58.96 | 71.95 | 61.43 | 76.49 |
| String | 76.67 | 66.67 | 76.67 | 66.67 | 70.00 | 80.00 | 80.00 | 70.00 |
| Tree | 32.78 | 36.33 | 45.78 | 47.89 | 35.67 | 57.11 | 40.33 | 60.56 |
| Graph | 43.80 | 37.47 | 45.44 | 65.70 | 54.56 | 68.23 | 56.96 | 68.61 |
| Searching | 51.29 | 52.00 | 61.14 | 60.57 | 54.86 | 56.71 | 58.14 | 74.86 |
| Sorting | 30.52 | 37.27 | 56.10 | 52.08 | 54.55 | 71.56 | 62.21 | 74.68 |
| Average | 46.98 | 47.07 | 56.93 | 59.66 | 56.92 | 67.39 | 60.67 | 70.83 |
| *Computer Organization* | | | | | | | | |
| Overview | 51.20 | 61.40 | 61.60 | 76.40 | 68.20 | 80.20 | 73.20 | 81.80 |
| Data Representation and Operation | 27.95 | 38.72 | 38.46 | 50.51 | 39.74 | 50.38 | 45.64 | 57.44 |
| Storage System | 41.80 | 46.10 | 58.00 | 61.70 | 53.60 | 68.10 | 56.20 | 68.50 |
| Instruction System | 51.76 | 53.68 | 57.79 | 59.56 | 53.82 | 75.74 | 65.29 | 80.44 |
| Central Processing Unit | 41.93 | 42.66 | 53.67 | 54.50 | 51.65 | 62.75 | 51.74 | 74.86 |
| Bus | 60.70 | 59.12 | 61.40 | 66.32 | 47.37 | 71.75 | 66.49 | 73.33 |
| Input/Output System | 37.58 | 35.48 | 29.19 | 52.42 | 44.03 | 52.42 | 35.48 | 58.23 |
| Average | 44.70 | 48.17 | 51.44 | 60.20 | 51.20 | 65.91 | 56.29 | 70.66 |
| *Computer Network* | | | | | | | | |
| Overview and Architecture | 52.15 | 48.31 | 58.77 | 62.77 | 58.15 | 68.62 | 57.23 | 69.08 |
| Physical Layer | 42.11 | 47.61 | 52.25 | 57.89 | 53.52 | 65.77 | 54.51 | 69.01 |
| Data Link Layer | 32.35 | 41.06 | 42.35 | 57.12 | 50.61 | 59.62 | 60.23 | 63.94 |
| Network Layer | 38.40 | 48.78 | 58.47 | 62.37 | 65.19 | 75.57 | 62.98 | 77.48 |
| Transport Layer | 42.95 | 48.72 | 66.28 | 70.77 | 63.46 | 81.79 | 61.54 | 86.79 |
| Application Layer | 47.61 | 55.00 | 60.34 | 65.91 | 63.30 | 75.34 | 64.55 | 79.89 |
| Average | 42.60 | 48.25 | 56.41 | 62.81 | 59.04 | 71.12 | 60.17 | 74.37 |
| *Operating System* | | | | | | | | |
| Overview | 39.74 | 40.65 | 48.57 | 65.32 | 60.65 | 69.87 | 51.82 | 68.31 |
| Processes and Threads | 34.14 | 42.61 | 43.57 | 55.73 | 50.83 | 63.57 | 47.58 | 66.82 |
| Memory Management | 31.63 | 42.04 | 52.04 | 51.02 | 53.67 | 60.71 | 51.02 | 70.41 |
| File Management | 40.00 | 49.34 | 57.37 | 54.87 | 55.66 | 61.97 | 56.32 | 64.08 |
| Input/Output Management | 34.88 | 36.83 | 41.46 | 50.98 | 47.07 | 51.10 | 38.05 | 59.76 |
| Average | 36.08 | 42.29 | 48.60 | 55.58 | 53.58 | 61.44 | 48.96 | 65.88 |
| Overall | 41.08 | 45.54 | 52.52 | 58.90 | 54.37 | 66.08 | 55.91 | 70.41 |

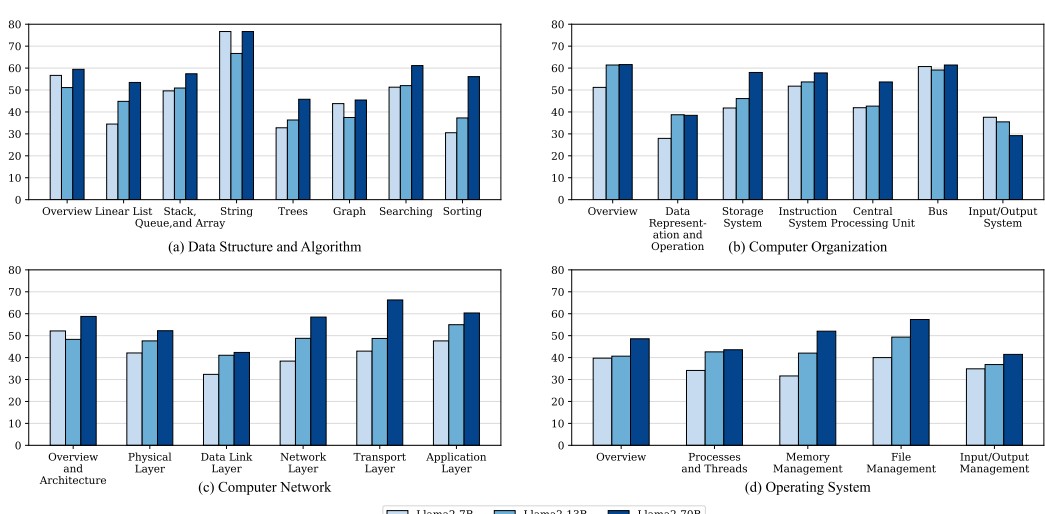

(a) Data Structure and Algorithm

(b) Computer Organization

(c) Computer Network

(d) Operating System

Llama2-7B   Llama2-13B   Llama2-70B

Figure 15: The performance of the Llama2 series models in each subfield.

of scores increasing with scale in most subfields (17 out of 26 subfields); however, there are some exceptions. For instance, Llama2-7B performs exceptionally well in the "string" chapter of DSA, while Llama2-13B excels in the "Data Representation and Operation" chapter of CO, surpassing the performance of Llama2-70B.

### E.2   SCALE-SCORE FITTING FUNCTION FOR CS-BENCH

To enhance CS performance, large-scale models are often utilized; however, these models demand more computational resources for both training and deployment inference. Therefore, it is desirable

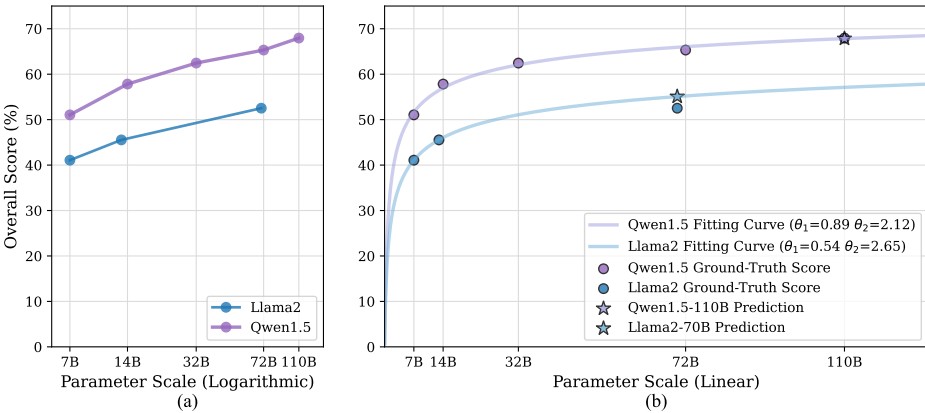

Figure 16: The logarithmic scale-score performance and scale-score fitting curve of Qwen1.5 and Llama2 series.

to establish a relationship between model scale and CS performance, enabling the prediction of theoretically larger models' scores on CS-Bench based on the performance of smaller-scale models. The established fitting function should adhere to the following criteria:

1. The score should monotonically increase with the increase in model scale, approaching 0 as the scale approaches 0, and approaching 1 (100%) as the scale approaches infinity.

2. As illustrated in Figure 16 (a), when the model scale varies exponentially, the score should exhibit an approximately linear trend.

3. Due to variations in performance and change slopes among different model families at the same scale, the fitting function needs to incorporate model-family-specific hyperparameters.

Guided by these criteria, we experiment with various functions and find the following function to satisfy the conditions and work best:

$$\text{Score} = 1 - \frac{1}{\theta_1 log_{10}(\theta_2 \cdot \text{Scale} + 1) + 1} \tag{1}$$

Where $\theta_1$ and $\theta_2$ are hyperparameters specific to the model family. To validate the effectiveness of the function, we estimate hyperparameters based on the minimum mean square error on small-scale models and predict performance scores on larger-scale models. For the Qwen1.5 family, we use models of 7, 14, 32, and 72B to predict the 110B model's performance. For the Llama2 series, we predict the 70B model's performance based on 7B and 13B. As depicted in Figure 16 (b), for Qwen1.5 110B, the predicted score (67.83%) closely matches the true value (67.95%). For Llama2-70B, with only two reference data points, the predicted score (55.08%) deviates from the true value (52.52%) by only 2.56%.

### E.3 MODEL PERFORMANCE ON CS-BENCH ACROSS DIFFERENT LANGUAGES

We first compare the performance of multilingual models in different languages within CSBench, as shown in Figure 17. It is observed that the Llama2 and Llama3 series show a significant decline in performance for languages other than English, such as Llama3-8B. In contrast, GPT-4o maintains a good balance across multiple languages. Next, we select the Chinese model as a case outside of English, conducting in-depth tests on Chinese-oriented models and comparing their performance differences between Chinese and English.

**Performance on CS-Bench (CN).** We assess models that support Chinese on CS-Bench (CN). The foundation models include the LLama3 and GPT-4 series, which are not specifically optimized for Chinese, as well as Chinese-oriented open-source models, including ChatGLM, Baichuan2, InternLm2, Qwen1.5 and llama3-chinese series. We also evaluate Chinese-oriented closed-source models, including GLM-4 and ERNIE-3.5/4. Details of these models are provided in Appendix D.4.

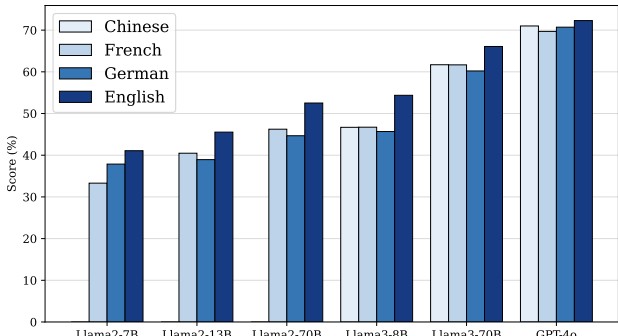

Figure 17: Comparison of models in different languages on CS-Bench. Due to the Llama2 series not supporting Chinese, we ignore their results in CS-Bench (CN).

Table 16: Zero-shot scores (%) of LLMs across domains on CS-Bench (CN), where "Klg" represents knowledge-type, "Rng" represents reasoning-type, and "Avg" represents Average.

| Model | Data Struc & Algo | | | Computer Organization | | | Computer Network | | | Operating System | | | Overall | | |
|---|---|---|---|---|---|---|---|---|---|---|---|---|---|---|---|
| | Klg | Rng | Avg | Klg | Rng | Avg | Klg | Rng | Avg | Klg | Rng | Avg | Klg | Rng | Avg |
| Random | 28.04 | 24.63 | 26.65 | 26.57 | 25.24 | 26.13 | 26.34 | 22.49 | 24.98 | 29.06 | 24.23 | 27.27 | 27.4 | 24.12 | 26.20 |
| *Open-source LLM (Scale < 10B)* | | | | | | | | | | | | | | | |
| ChatGLM3-6B | 41.74 | 32.48 | 37.97 | 44.07 | 34.91 | 41.05 | 49.02 | 32.31 | 43.14 | 43.02 | 32.86 | 35.98 | 44.67 | 33.09 | 40.45 |
| Baichuan2-7B | 42.04 | 31.51 | 37.75 | 44.93 | 37.88 | 42.61 | 50.74 | 31.11 | 43.83 | 42.18 | 34.07 | 39.16 | 45.27 | 33.47 | 40.97 |
| InternLm2-7B | 41.97 | 34.54 | 38.95 | 55.77 | 38.67 | 50.13 | 60.05 | 41.86 | 53.65 | 50.94 | 44.07 | 48.39 | 52.71 | 39.61 | 47.94 |
| Qwen1.5-7B | 49.13 | 37.71 | 44.48 | 60.86 | 44.48 | 55.46 | 60.90 | 45.68 | 55.54 | 58.38 | 48.24 | 54.61 | 57.62 | 43.79 | 52.59 |
| Llama3-8B | 50.47 | 29.68 | 42.01 | 50.81 | 36.30 | 46.03 | 56.09 | 42.21 | 51.21 | 52.01 | 38.85 | 47.12 | 52.46 | 36.61 | 46.69 |
| Llama3-8B-Chinese | 49.20 | 33.72 | 42.90 | 54.99 | 33.09 | 47.77 | 58.77 | 48.59 | 55.19 | 55.58 | 41.10 | 50.20 | 54.84 | 39.17 | 49.13 |
| *Open-source LLM (Scale > 10B)* | | | | | | | | | | | | | | | |
| Baichuan2-13B | 48.83 | 34.68 | 43.07 | 54.18 | 36.00 | 48.18 | 55.11 | 39.85 | 49.74 | 49.19 | 40.27 | 45.88 | 52.10 | 37.63 | 46.83 |
| Qwen1.5-14B | 51.47 | 48.81 | 50.39 | 64.43 | 46.85 | 58.63 | 68.69 | 55.18 | 63.94 | 69.58 | 56.59 | 64.76 | 63.78 | 51.81 | 59.42 |
| InternLm2-20B | 51.97 | 38.03 | 46.30 | 58.36 | 45.76 | 54.20 | 60.60 | 50.50 | 57.05 | 58.70 | 45.66 | 53.86 | 57.59 | 44.85 | 52.95 |
| Qwen1.5-32B | 55.89 | 56.70 | 56.22 | 67.74 | 60.00 | 65.19 | 70.33 | 66.83 | 69.10 | 72.40 | 62.03 | 68.55 | 66.77 | 61.35 | 64.80 |
| Llama3-70B | 53.28 | 55.41 | 54.15 | 67.97 | 49.58 | 61.91 | 71.07 | 61.81 | 67.81 | 65.29 | 57.36 | 62.35 | 64.86 | 56.18 | 61.70 |
| Qwen1.5-72B | 58.16 | 52.02 | 55.66 | 70.28 | 52.91 | 64.55 | 75.25 | 66.23 | 72.08 | 74.12 | 63.19 | 70.06 | 69.73 | 58.52 | 65.64 |
| *Closed-source LLM* | | | | | | | | | | | | | | | |
| GPT-3 | 54.15 | 39.63 | 48.24 | 60.86 | 43.27 | 55.06 | 64.29 | 48.89 | 58.87 | 56.36 | 39.84 | 50.22 | 59.27 | 42.96 | 53.33 |
| GPT-4 | 60.03 | 60.28 | 60.13 | 77.60 | 60.24 | 71.88 | 73.50 | 72.86 | 73.27 | 71.46 | 65.60 | 69.29 | 71.06 | 64.80 | 68.78 |
| GPT-4o | 61.67 | 66.45 | 63.62 | 78.86 | 55.32 | 71.10 | 78.61 | 74.17 | 77.05 | 72.66 | 69.94 | 71.67 | 73.46 | 66.69 | 71.00 |
| GLM-4 | 58.12 | 58.37 | 58.22 | 74.03 | 59.49 | 69.24 | 71.65 | 70.21 | 71.14 | 73.31 | 67.14 | 71.06 | 69.55 | 63.75 | 67.44 |
| ERNIE-3.5 | 58.16 | 55.62 | 57.13 | 74.56 | 58.73 | 69.34 | 74.68 | 65.16 | 71.33 | 72.13 | 63.37 | 68.94 | 70.28 | 60.63 | 66.77 |
| ERNIE-4 | 57.92 | 62.33 | 59.72 | 78.24 | 64.18 | 73.60 | 76.27 | 69.74 | 73.97 | 75.84 | 69.54 | 73.54 | 72.49 | 66.36 | 70.26 |

Table 17: Zero-shot scores (%) of LLMs across task formats on CS-Bench (CN).

| Model | Multiple-choice | | | Assertion | | | Fill-in-the-blank | | | Open-ended | | | Overall | | |
|---|---|---|---|---|---|---|---|---|---|---|---|---|---|---|---|
| | Klg | Rng | Avg | Klg | Rng | Avg | Klg | Rng | Avg | Klg | Rng | Avg | Klg | Rng | Avg |
| Random | 25.00 | 25.00 | 25.00 | 50.00 | 50.00 | 50.00 | 0.00 | 0.00 | 0.00 | 10.00 | 10.00 | 10.00 | 27.4 | 24.12 | 26.20 |
| *Open-source LLM (Scale < 10B)* | | | | | | | | | | | | | | | |
| ChatGLM3-6B | 45.21 | 34.07 | 40.90 | 54.41 | 48.48 | 53.05 | 23.53 | 11.32 | 20.57 | 43.80 | 25.22 | 30.68 | 44.67 | 33.09 | 40.45 |
| Baichuan2-7B | 44.96 | 32.26 | 40.05 | 53.80 | 56.57 | 54.43 | 29.41 | 13.21 | 25.48 | 47.20 | 27.52 | 33.31 | 45.27 | 33.47 | 40.97 |
| InternLm2-7B | 51.09 | 40.08 | 46.83 | 59.88 | 55.56 | 58.89 | 44.12 | 18.87 | 38.00 | 60.80 | 33.27 | 41.37 | 52.71 | 39.61 | 47.94 |
| Qwen1.5-7B | 59.64 | 48.50 | 55.33 | 60.79 | 50.51 | 58.44 | 42.35 | 15.09 | 35.74 | 58.20 | 30.35 | 38.54 | 57.62 | 43.79 | 52.59 |
| Llama3-8B | 53.26 | 35.67 | 46.45 | 56.23 | 59.60 | 57.00 | 42.35 | 16.98 | 36.20 | 49.60 | 29.47 | 35.39 | 52.46 | 36.61 | 46.69 |
| Llama3-8B-Chinese | 55.43 | 40.08 | 49.49 | 59.57 | 56.57 | 58.88 | 42.94 | 16.98 | 36.64 | 55.60 | 30.62 | 37.97 | 54.84 | 39.17 | 49.13 |
| *Open-source LLM (Scale > 10B)* | | | | | | | | | | | | | | | |
| Baichuan2-13B | 52.11 | 39.48 | 47.22 | 59.57 | 51.52 | 57.73 | 40.00 | 16.98 | 34.42 | 43.40 | 27.08 | 31.88 | 52.10 | 37.63 | 46.83 |
| Qwen1.5-14B | 67.82 | 57.72 | 63.91 | 65.05 | 56.57 | 63.11 | 43.53 | 24.53 | 38.92 | 63.80 | 34.96 | 43.44 | 63.78 | 51.81 | 59.42 |
| InternLm2-20B | 58.49 | 46.89 | 54.00 | 59.57 | 54.55 | 58.42 | 47.06 | 26.42 | 42.05 | 67.00 | 35.40 | 44.69 | 57.59 | 44.85 | 52.95 |
| Qwen1.5-32B | 71.26 | 68.74 | 70.28 | 64.74 | 63.64 | 64.49 | 51.76 | 28.30 | 46.07 | 63.40 | 42.04 | 48.32 | 66.77 | 61.35 | 64.80 |
| Llama3-70B | 66.03 | 60.32 | 63.82 | 66.57 | 65.66 | 66.36 | 58.24 | 33.96 | 52.35 | 59.00 | 40.71 | 46.09 | 64.86 | 56.18 | 61.70 |
| Qwen1.5-72B | 72.41 | 67.74 | 70.60 | 72.34 | 55.56 | 68.51 | 54.71 | 28.30 | 48.30 | 63.80 | 34.96 | 43.44 | 69.73 | 58.52 | 65.64 |
| *Closed-source LLM* | | | | | | | | | | | | | | | |
| GPT-3 | 57.98 | 42.48 | 51.98 | 65.05 | 61.62 | 64.27 | 54.71 | 24.53 | 47.39 | 56.60 | 36.81 | 42.63 | 59.27 | 42.96 | 53.33 |
| GPT-4 | 73.31 | 67.13 | 70.92 | 72.04 | 67.68 | 71.04 | 62.35 | 60.38 | 61.87 | 60.40 | 54.16 | 56.00 | 71.06 | 64.80 | 68.78 |
| GPT-4o | 75.92 | 69.33 | 73.37 | 73.86 | 68.69 | 72.68 | 62.94 | 50.94 | 60.03 | 70.20 | 62.92 | 65.06 | 73.46 | 66.69 | 71.00 |
| GLM-4 | 73.68 | 69.76 | 72.16 | 68.09 | 57.58 | 65.69 | 55.03 | 47.17 | 53.12 | 68.00 | 52.92 | 57.36 | 69.55 | 63.75 | 67.44 |
| ERNIE-3.5 | 72.24 | 63.71 | 68.94 | 69.30 | 61.62 | 67.55 | 63.91 | 50.94 | 60.76 | 70.40 | 51.95 | 57.38 | 70.28 | 60.63 | 66.77 |
| ERNIE-4 | 73.55 | 70.35 | 72.31 | 72.34 | 56.57 | 68.74 | 70.00 | 67.92 | 69.50 | 68.40 | 58.32 | 61.28 | 72.49 | 66.36 | 70.26 |

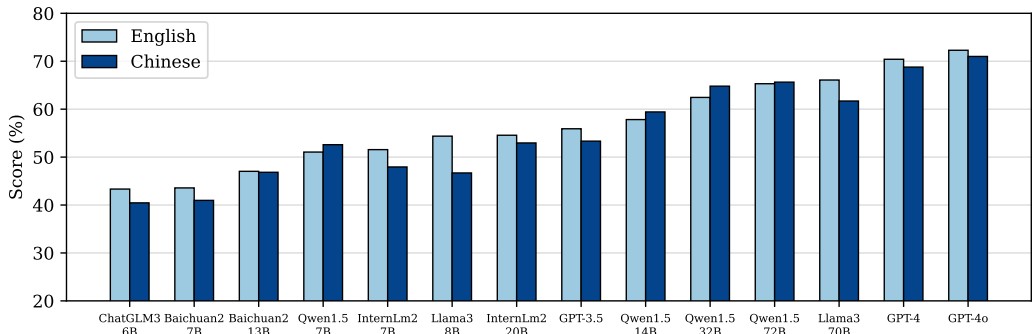

Figure 18: Comparison of models in English and Chinese on CS-Bench.

As shown in Table 16 and Table 17, the scores of these models on CS-Bench(CN) range from 40.45% to 70.26%. Despite not being specifically optimized for Chinese, GPT-4o still achieves the best performance. Among the Chinese-oriented models, ERNIE-4 outperforms GPT-4, achieving performance close to GPT-4o. Additionally, ERNIE-3.5 and GLM-4 score similarly, slightly lower than GPT-4's performance in Chinese. Notably, Llama3-8B-chinese surpasses Llama3-8B by 2.44%, highlighting the importance of adapting models to specific languages. We further compare the performance of the models on CS-Bench(EN) and CS-Bench(CN) in Figure 18. Compared to English, the GPT and Llama3 series, which are not optimized for Chinese, perform worse on Chinese context. For instance, Llama3-8B experiences a decrease of 7.68% on Chinese, and Llama3-70B drops by 4.38%. Although some Chinese-oriented models also show slight decreases in performance in the Chinese context, such as InterLm2-20B, the decline is much less significant than that of the Llama3 series. Moreover, the Qwen1.5 series even demonstrates improved performance on Chinese tasks. Finally, we observe that larger models within the same family are less affected by different languages, as reflected in Baichuan2-7/13B, Internlm2-7/20B, and Llama3-8/70B.

## E.4 CASE STUDY OF ERROR TYPES

We first introduce the error types of knowledge-type questions and reasoning-type questions in Table 18 and Table 19. To facilitate a better understanding of each error type, we provide examples of each error type made by GPT-3.5 in knowledge-based and reasoning-based questions in Table20 and 21, respectively. Additionally, Table 22 presents several examples that contain multiple error types.

Table 18: Error types for knowledge-type questions.

| Error Type | Explanation |
| --- | --- |
| Complete conceptual error | For a certain piece of knowledge or concept, the model is completely unaware of it or has misunderstood all of it. |
| Partial conceptual error | For a certain piece of knowledge or concept, the model has only grasped part of it or has misunderstood part of its content. |
| Conceptual confusion | For multiple pieces of knowledge or concepts, the model has incorrectly matched their names with their attributes. |

Table 19: Error types for reasoning-type questions.

| Error Type | Explanation |
| --- | --- |
| Concept-related error | The model misunderstands certain pieces of knowledge, causing incorrect answers, This includes the three types of knowledge-based errors mentioned above. |
| Logical reasoning error | The model made logical reasoning errors when answering computer science-related questions. |
| Mathematical reasoning error | During the response process, errors in numerical calculations, symbolic calculations, and other computational processes caused the model to produce incorrect answers. |

Table 20: Example of knowledge-type question error types. We use red to mark errors in the output and blue to indicate the reasons for the errors.

| Error Type | Example |
|---|---|
| Complete conceptual error | **Question:**
When two sequential stacks share the same one-dimensional array space, how are their stack bottoms and stack tops set?"
A: Two stacks with the same bottom end, extending in opposite directions from the top.
B: Two stacks with tops at the same end and bottoms extending in opposite directions.
C: The bottom and top of the stack are at the same end.
D: Two stacks with bottoms at opposite ends and tops extending towards the middle.
**Answer:**
**D**
**Wrong Output:**
When two sequential stacks share the same one-dimensional array space, their stack bottoms and stack tops are set as follows: Two stacks with tops at the same end and bottoms extending in opposite directions. This means that the top of one stack and the bottom of the other stack are at the same end of the array, while their bottoms extend in opposite directions. This allows for efficient utilization of the array space for both stacks. The correct answer is (B). Answer: (B) Two stacks with tops at the same end and bottoms extending in opposite directions.
**Error Details:**
In GPT-3.5's response, it was mentioned that the top of two stacks are at the same end, while the bottom of the stack extends in opposite directions. This reflects that GPT-3.5 is completely not familiar with the implementation of shared stacks. |
| Partial conceptual error | **Question:**
What are the main functions of the network layer?
A: Data Transparent Transmission
B: Power Management
C: Signal Amplification
D: Data Compression
**Answer:**
**A**
**Wrong Output:**
The main functions of the network layer are data transparent transmission and data compression. The network layer is responsible for routing and forwarding data packets between different networks, ensuring that data is delivered to the correct destination. It does not handle power management or signal amplification. The correct answers are A and D.
**Error Details:**
GPT-3.5 correctly answered that transparent data transmission is a function of the network layer, but at the same time incorrectly pointed out that data compression is the main function of the network layer, witch indicating an incorrect understanding of certain aspects of network layer functionality. |
| Conceptual confusion | **Question:**
What are the basic steps of simple selection sort?
A: Select the element with the smallest key from the elements to be sorted and swap it with the first element. B: Insert elements into the sorted sequence in order.
C: Compare adjacent elements in sequence and swap them.
D: Use the divide and conquer method to sort elements.
**Answer:**
**D**
**Wrong Output:**
The basic steps of simple selection sort are: (A) Select the element with the smallest key from the elements to be sorted and swap it with the first element. (C) Compare adjacent elements in sequence and swap them. The correct answer is A and C.
**Error Details:**
The answers output by GPT-3.5 are A and C, and it is claimed that the basic steps of simple selection sorting are to select the smallest element and exchange it with the first element, and to compare adjacent elements and exchange them. This is obviously incorrect because comparing adjacent elements and swapping them is a Bubble Sort This indicates that GPT confuses bubble sorting and selection sorting. |

Table 21: Example of reasoning-type question error types.

| Error Type | Example |
|---|---|
| Concept-related error | **Question:**
In the topological ordering of the directed graph G, if vertex Vi precedes vertex Vj, then the following situation cannot occur: ()
A: G contains an arc <Vi, Vj>.
B: There is a path from Vi to Vj in G.
C: There is no arc <Vi, Vj> in G.
D: There is a path from Vj to Vi in G.
**Answer: D**
**Wrong Output:**
If vertex Vi precedes vertex Vj in the topological ordering of the directed graph G, then it means that there is a directed path from Vi to Vj in G. Therefore, the si- tuation where there is no arc <Vi, Vj> in G cannot occur. The correct answer is (C) There is no arc <Vi, Vj> in G.
**Error Details:**
The sequence obtained by topological sorting has i before j. GPT-3.5 chose C in the answer, but according to the topological sorting rule, placing i before j does not necessarily indicate that two nodes can be directly connected. GPT-3.5 does not under- stand the meaning of topological sorting. |
| Logical reasoning error | **Question:**
In the array representation of a linear list with n elements, the operations with a time complexity of O(1) are () I. Accessing the ith ($1 \leq i \leq n$) node and finding the immediate predecessor of the ith ($2 \leq i \leq n$) node. II. Inserting a new node after the last node. III. Deleting the first node. IV. Sequentially outputting the values of these n elements.
A:I
B: II, III
C: I, II
D: I, II, III
**Answer: C**
**Wrong Output:**
The operations with a time complexity of O(1) in the array representation of a linear list with n elements are: I. Accessing the ith ($1 \leq i \leq n$) node and finding the immediate predecessor of the ith($2 \leq i \leq n$) node. II. Inserting a new node after the last node. III. Deleting the first node. Therefore, the correct answer is (D) I, II, III. Answer: D
**Error Details:**
For item II in the question, after deleting the first element of the array, the next n-1 elements need to be moved forward, resulting in a time complexity of o (n). However, GPT-3.5 considers it to be o(1), which is a GPT-3.5 inference error. |
| Mathematical reasoning error | **Question:**
In a computer with an 8-bit adder, the machine numbers of signed integers X and y are represented in two's complement, with [x] complement = F5H and [y] complement = 7EH. If the calculation X - y is performed in this adder,the carry-in information for the lower bit and the overflow flag OF after the operation are respectively ().
A: 1,1 B: 1,0 C: 0,1 D: 0,0
**Answer: C**
**Wrong Output:**
First, let's find the two's complement of y. The two's complement of a number is obtained by inverting all the bits and adding 1. [y] complement = 7EH Inverting all the bits: 10000001. Adding 1: 10000010.Now, we can perform the subtraction X - y using the 8-bit adder. [X] complement = F5H [y] complement = 10000010. Adding X and the two's complement of y: 11110101 + 10000010 = 101101111. The result is 10110111, which is the two's complement representation of the result of X - y. To determine the carry-in information for the lower bit, we need to look at the carry-out from the lower bit. In this case, the carry-out is 1, so the carry-in for the lower bit is also 1. To determine the overflow flag (OF), we need to compare the carry-in and carry-out from the sign bit. In this case, the carry-in and carry-out from the sign bit are both 0, so the overflow flag is 0.
**Error Details:**
The complement of x and (- y) is calculated incorrectly. |

Table 22: Examples containing multiple types of errors.

| Error Type | Example |
|---|---|
| Logical reasoning error & Mathematical reasoning error | **Question:**
Given that the system uses a 32-bit physical address and a 48-bit virtual address, with a page size of 4KB and a page table entry size of 8B. Assuming the system employs pure paging, it will use a ()-level page table, and the page offset is () bits.
A: 3, 12
B: 3, 14
C: 3, 12
D: 4, 14
**Answer:**
C
**Wrong Output:**
To determine the number of levels in the page table, we need to calculate the number of bits required to represent the virtual address space. The virtual address space is 48 bits, and the page size is 4KB, which is equivalent to $2^{12}$ bytes. Therefore, the num- ber of pages in the virtual address space is $2^{48}$ / $2^{12}$ = $2^{36}$ pages. Since each page table entry is 8 bytes, we can calculate the number of entries in each level of the page table. At the first level, there are $2^{36}$ entries, which can be represented by 36 bits. At the second level, each entry points to a page table at the next level, so there are $2^{36}$ entries. At the third level, each entry points to a page table at the next level, so there are $2^{36}$ entries. Therefore, we need 3 levels of page tables. The page offset is the number of bits required to represent the offset within a page. Since the page size is 4KB, which is equivalent to $2^{12}$ bytes, the page offset is 12 bits. Therefore, the correct answer is (A) 3, 12.
**Error Details:**
The number of page table items that can be accommodated on each page is 4KB/8B = 512=$2^9$, while GPT-3.5 did not mention calculations and directly stated that the first level page table has $2^{36}$ items; According to the GPT-3.5 calculation, if there are $2^{36}$ items in the first level page table, there should be a 36/36=first level page table instead of a third level page table, resulting in a mathematical inference error. From the process of calculating the number of page table levels in GPT-3.5, it can be seen that its logical reasoning path is not correct. |
| Complete conceptual error & Conceptual confusion | **Question:**
The following network connecting devices that operate at the data link layer are ()
A: Repeaters and Hubs
B: Hubs and Bridges
C: Network bridges and local area network switches
D: Hubs and Local Area Network (LAN) Switches
**Answer:**
C
**Wrong Output:**
The correct answer is (D) Hubs and Local Area Network (LAN) Switches. Repeaters operate at the physical layer, while hubs and bridges operate at the data link layer. Local Area Network (LAN) switches also operate at the data link layer. Therefore, the correct combination of network connecting devices that operate at the data link layer is hubs and LAN switches.
**Error Details:**
GPT-3.5 is unaware that the bridge is a network link device operating at the data link layer, indicating a lack of knowledge about the bridge. Meanwhile, GPT-3.5 believes that the Hub is a device at the data link layer, indicating that it confuses physical layer devices with data link layer devices. |

## E.5 ANALYSIS FROM MODEL REPRESENTATION

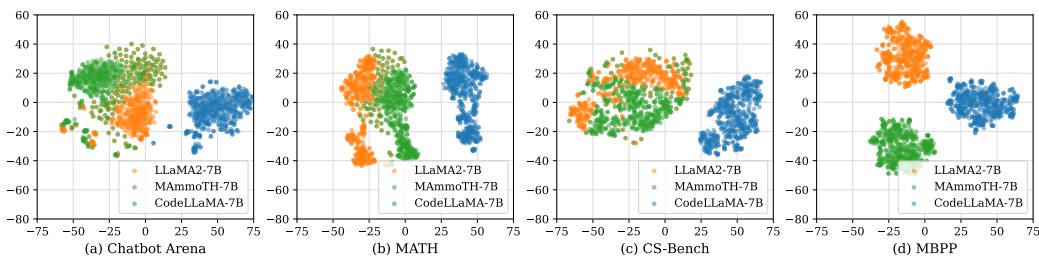

Figure 19: Representation Visualization of General and Expert Models.

To analyze the reasons behind capability correlations, we first explore the relationships between different abilities of general models at the representation level. We extract the last hidden layer of Llama3-8B-instruct and obtain representations for different domains through average pooling. We then calculate the average cosine distance between CS-Bench reasoning-type representations and other data representations, where smaller distances indicate greater similarity.

As shown in Table 23, We find that the similarity of representations between different abilities/domains generally aligns with the observed capability correlations, especially in the areas of code and math. This explains the correlation between CS skills and math and coding abilities at the representation level. Additionally, due to BBH containing various reasoning data, the distance between CS-Bench and BBH is relatively small, which is consistent with the analysis in Section 3.5.

Table 23: Representation Distance between CS-Bench and Different Datasets.

| Dataset | Domain | Distance |
|---|---|---|
| MBPP | code | 0.4645 |
| MATH | math | 0.5395 |
| BBH | reasoning | 0.5494 |
| T-Eval | tool | 0.6310 |
| Chatbot Arena (first turn) | chat | 0.5555 |
| GPQA | science | 0.5775 |
| L-eval | long context | 0.6409 |

Next, we analyze from the perspective of training data characteristics. We intuitively find that some questions in CS-Bench can be understood as mathematical problems in a computer science context, while the main role of code is to supplement the model's knowledge of data structures and logical reasoning abilities. This data-level correlation explains the expert model experimental results in Table 3 and 4, specifically why math and code expert models help improve CS capabilities in certain areas, even when their general abilities are weakened. For example: Q1: `If the data part is 3800B and the maximum fragment size is 1420B, what is the total length in bytes of the third fragment?` and Q2: `The height h of a binary tree with 1025 nodes is ().`

Finally, we analyze the relationship between general models and math & code expert models in computer science representations. We select the chat model Llama2-7B-Chat, math expert model MAmmoTH, and code expert model CodeLlama, all based on Llama2-7B-base, to observe their average cosine distances for the same data points in CS-Bench.

Table 24: Representation Distance of Different Models on CS-Bench.

| Model pair | Distance |
|---|---|
| between Chat and Math model | 0.4645 |
| between Chat and Code model | 0.5395 |
| between Math and Code model | 0.5494 |

Table 24 show that expert models' representations shift to some extent compared to the Chat model, especially the Code model, which can be attributed to Code's unique data patterns. Further representation visualization in Figure 19 reveals that in the general dataset Chatbot-arena, the features of the three models do not significantly separate. However, in MATH and CS-Bench datasets, the Code model forms a distinct feature cluster, while Math and Chat models' representations remain similar, as we state before: many questions in CS can be viewed as mathematical problems in a CS context. Lastly, in the MBPP dataset, the representations of the three models clearly differentiate and form their own clusters.

### E.6 EXPLORATION FOR IMPROVING CS PERFORMANCE

Although the specific methods to enhance LLM performance in CS are not the primary focus of this paper, we are eager to explore potential solutions based on our findings. These include capability transfer, inference frameworks, integrating external knowledge sources, incorporating symbolic reasoning systems, and targeted fine-tuning.

**Capability Transfer.** The capability analysis experiments in Section 3.5 demonstrate that math and code expert models can improve CS performance in certain areas. Therefore, in scenarios where CS data is scarce, training on math or code tasks can be leveraged to transfer and enhance CS capabilities.

**Inference Frameworks.** The experiments in Section 3.3 show that compared to GPT-4o, OpenAI-o1 significantly improves the reasoning scores on CS-Bench. This suggests that combining LLMs with reasoning frameworks can be expected to enhance reasoning capabilities. However, this approach comes with the trade-off of increased token consumption during inference.

**Integrating External Knowledge Sources.** The error type analysis in Section 3.4 identifies knowledge gaps as the main reason for LLM failures in the CS field. To address this, we implemented a RAG framework that uses Wikipedia as an external knowledge source.

Table 25: Changes in model scores after incorporating RAG.

| Model | Knowledge | Reasoning | Overall |
|---|---|---|---|
| GPT-3.5-turbo | 63.04 | 43.45 | 55.91 |
| GPT-3.5-turbo (+RAG) | 64.92(+1.88) | 48.68(+5.23) | 59(+3.09) |
| Llama3-8B | 59.75 | 44.97 | 54.37 |
| Llama3-8B(+RAG) | 66.11(+6.36) | 50.64(+5.67) | 60.47(+6.1) |

edge source. The process involves extracting key terms from the question, retrieving relevant information from Wikipedia, and summarizing it to provide supplemental knowledge, which is then integrated into the context for answering the question. The results are shown in Table 25. Both knowledge and reasoning scores of the model have improved significantly, aligning with the findings in error analysis: the primary errors in knowledge and reasoning questions are knowledge-related issues. Performance is expected to be further enhanced when combined with a more advanced RAG framework or more suitable knowledge sources.

**Incorporating Symbolic Reasoning Systems.** To enable joint reasoning between language and symbolic systems, we developed a Python API as a symbolic reasoning tool. This allows the LLM to generate Python code when it identifies a need for symbolic reasoning.

Table 26: Changes in model scores after incorporating Python Tool.

| Model | Knowledge | Reasoning | Overall |
|---|---|---|---|
| gpt-4o | 76.95 | 64.15 | 72.29 |
| gpt-4o(+Python Tool) | 77.47(+0.5) | 71.86(+7.71) | 75.42(+3.13) |

bolic reasoning. The generated code is executed using the Python tool, and the results are reintegrated into the LLM's context for further output generation. The results are shown in Table 26. Since symbolic reasoning focuses on improving reasoning tasks, we observe that there is little improvement in the model's knowledge capabilities, but there is a significant enhancement in its reasoning performance.

**Fine-tuning.** The error analysis also highlighted the need for specialized fine-tuning for CS. Due to the scarcity of fine-tuning data in the CS domain, we randomly sampled 10% of the English data from CS-Bench as a mini-test set.

Table 27: Changes in model scores after CS fine-tuning.

| Model | Knowledge | Reasoning | Overall |
|---|---|---|---|
| Llama3-8b-Instruct | 66.89 | 43.73 | 58.07 |
| Llama3-8b-Instruct (+SFT) | 71.19(+4.3) | 49.88(+6.15) | 63.07(+5) |

test set. The remaining data was converted into SFT training data and used to fine-tune Llama3-8B-Instruct. As shown in Table 27, despite the challenges posed by the difficulty of CS tasks and the limited size of the training set, the model's knowledge and reasoning capabilities improved after fine-tuning. We expect that fine-tuning with a larger amount of domain-specific data could further enhance the LLM's performance in CS.

Through these discussions, we aim to provide insights and directions for the future development of LLMs in the field of computer science.

### E.7 CASE STUDY OF OPENAI-O1

We present the responses of GPT-4o, OpenAI-o1-mini, and OpenAI-o1-preview to the same questions in Table 28, 29, and 30, and analyze them in detail as follows.

**Example 1.** In this example, the models GPT-4o, OpenAI-o1-mini, and OpenAI-o1-preview each scored 0.2, 0.3, and 0.2 respectively, but none of them provided the correct answer. GPT-4o had a correct understanding of the IEEE 754 standard concept but incorrectly identified the e fraction (last 23 bits) as 1000 0000 0000 0000 0000 0000 (24 bits), which led to the wrong result. OpenAI-o1-mini also had a correct understanding of the IEEE 754 standard concept but made an error in identifying the e fraction (last 23 bits), resulting in an incorrect outcome. Similarly, OpenAI-o1-preview made a mistake in the interpretation of the fractional part, leading to an incorrect result. All three models had a correct understanding of the IEEE 754 standard; however, they made errors in identifying the last 23 bits of the fraction, which led to incorrect results.

**Example 2.** In this example, the models GPT-4o, OpenAI-o1-mini, and OpenAI-o1-preview each scored 0.9, 1.0, and 0.3 respectively. GPT-4o and OpenAI-o1-mini answered the question correctly, while OpenAI-o1-preview did not provide the correct answer. OpenAI-o1-preview showed a detailed calculation process, it contained several critical errors that led to an incorrect result. The model used an incorrect throughput formula. The maximum throughput of the pure ALOHA protocol is 18.4% of the channel capacity, but the model directly used the full bandwidth of 56 kb/s for calculations. This resulted in an overestimation of the maximum number of stations.

**Example 3.** In this example, the models GPT-4o, OpenAI-o1-mini, and OpenAI-o1-preview each scored 0.3, 0.3, and 0.7 respectively, but none of them provided the correct answer. GPT-4o and OpenAI-o1-mini used incorrect methods to calculate the total number of nodes. The models incorrectly calculated the number of nodes from level 0 to level 8 as $2^9 - 1 = 511$. The $9^{th}$ level should have $2^8 - 1 = 255$ nodes because the levels are counted starting from 0, which led to incorrect conclusions. OpenAI-o1-preview correctly calculated the total number of nodes but lacked an analysis of the leaf nodes at level 8 when calculating the number of leaf nodes. Although the answer was not fully complete, it arrived at the correct result.

Table 28: An example where GPT-4o, OpenAI-o1-mini , and OpenAI-o1-preview all failed.

| Type | Content |
|------|---------|
| Question | **Question:**
Explain which type of number
(normalized number, denormalized number, infinity, 0) the following IEEE 754 single-precision value represents. When it represents a specific value, please provide
that value: 0b1111 1111 1000 0000 0000 0000 0000 0000.
**Reference Answer:**
Since the exponent field of the number is all ones and the mantissa field is zero, with the sign bit being one, it represents negative infinity. |
| GPT-4o | **Output:**
The given IEEE 754 single-precision value is:
0b1111 1111 1000 0000 0000 0000 0000 0000
To determine what this represents, let's break it down:
1. The first bit is the sign bit. 2. The next 8 bits are the exponent. 3. The remaining 23 bits are the fraction (or mantissa).
For the given value:
- The sign bit (first bit) is 1, indicating a negative number.
- The exponent (next 8 bits) is 1111 1111, which is 255 in decimal.
- The fraction (last 23 bits) is 1000 0000 0000 0000 0000 0000.
According to the IEEE 754 standard: - An exponent of 255 (all ones) with any non-zero fraction indicates a NaN (Not a Number). - An exponent of 255 with a zero fraction would indicate infinity, determined by the sign bit.
In this case, the fraction is non-zero (it starts with 1000). Therefore, this specific combination represents a NaN (Not a Number).
So, the IEEE 754 single-precision value 0b1111 1111 1000 0000 0000 0000 0000 0000 represents a NaN (Not a Number).
**Score:**
0.2 |
| OpenAI-o1-mini | **Output:**
The IEEE 754 single-precision value 0b11111111 10000000 00000000 00000000 represents a NaN (Not a Number).
Explanation:
- Sign Bit: 1 (negative)
- Exponent: 11111111 (255), which is the maximum value.
- Fraction (Mantissa): 10000000000000000000000 (not all zeros)
In IEEE 754: - When the exponent is 255 and the fraction is not all zeros, the value is classified as NaN. - If the exponent were 255 and the fraction were all zeros, it would represent infinity.
Since the fraction here is not all zeros, the value is NaN, indicating an undefined or unrepresentable value.
**Score:**
0.3 |
| OpenAI-o1-preview | **Output:**
The given IEEE 754 single-precision binary value is: 0b11111111110000000000000000000000
Breaking it down into the IEEE 754 format:
- Sign bit (S): The first bit is '1', indicating the sign (though for NaNs, sign bits are generally ignored).
- Exponent bits (E): The next 8 bits are '11111111'.
- Fraction/Mantissa bits (F): The remaining 23 bits are '10000000000000000000000'.
Analysis:
1. Exponent Analysis: - Exponent bits are '11111111' ($E = 255$), which is the maximum possible value for the exponent in IEEE 754. - When the exponent is all ones ($E = 255$), it indicates either infinity or NaN.
2. Fraction Analysis: - Fraction bits are not all zeros ($F \neq 0$). Specifically, the most significant bit of the fraction is '1', and the rest are zeros. - When $E = 255$ and $F \neq 0$, the value represents a Not a Number (NaN).
3. Type of NaN: - Quiet NaN vs. Signaling NaN: - If the most significant bit of the fraction (also known as the quiet bit) is '1', it's a Quiet NaN (qNaN). - If this bit is '0' and there's any non-zero bit in the fraction, it's a Signaling NaN (sNaN). - In this case, the most significant bit of the fraction is '1', so it's a Quiet NaN.
Conclusion:
The given IEEE 754 single-precision value represents a Quiet NaN (Not a Number). This special value is used to represent undefined or unrepresentable numerical results, such as the result of $0/0$ or the square root of a negative number.
Answer: It represents a quiet NaN (Not a Number) in IEEE 754 single-precision format.
**Score:**
0.2 |

Table 29: An example where GPT-4o and OpenAI-o1-mini succeeded, OpenAI-o1-preview failed.

| Type | Content |
|---|---|
| Question | **Question:**
A group of N stations share a 56kb/s pure ALOHA channel, with each station outputting a 1000-bit frame on average every 100s, even if the previous frame has not finished transmitting. What is the maximum value of N?
**Reference Answer:**
For the pure ALOHA protocol, the channel utilization is 0.184, so the available bandwidth is 0.184x56Kb/s. The bandwidth required by each station is 1000/100-10b/s. Therefore, the maximum value of N is 1030. |
| GPT-4o | **Output:**
To determine the maximum number of stations $N$ that can share a 56 kb/s pure ALOHA channel efficiently, we need to consider the throughput of the pure ALOHA protocol. The maximum throughput for pure ALOHA is 18.4%, or $\frac{1}{2e}$.
First, let's calculate the frame generation rate for one station:
- One station generates a 1000-bit frame every 100 seconds. - Therefore, the frame rate per station ($G$) is $\frac{1000 \text{ bits}}{100 \text{ s}} = 10$ bps.
For the total maximum channel capacity:
- The channel capacity is 56,000 bps. - With the maximum throughput efficiency of 18.4%, the effective throughput is $56,000 \times 0.184 = 10,304$ bps.
Now, to find the maximum number of stations $N$:
- Each station contributes 10 bps to the load. - The maximum number of stations can be calculated by dividing the effective throughput by the load per station: $N = \frac{10,304 \text{ bps}}{10 \text{ bps per station}} = 1,030.4$.
Since $N$ must be an integer, the maximum value of $N$ is 1,030.
Thus, the maximum value of $N$ is 1,030 stations.
**Score:**
0.9 |
| OpenAI-o1-mini | **Output:**
To determine the maximum number of stations $N$ that can share a 56 kb/s pure ALOHA channel, follow these steps:
1. Calculate Frame Transmission Time ($T$): $T = \frac{1000 \text{ bits}}{56,000 \text{ bits/s}} \approx 0.01786$ seconds (17.86 ms)
2. Determine the Offered Load ($G$): Each station sends a frame every 100 seconds, so the load per station per frame time is $G_i = \frac{T}{100} = \frac{1}{5600}$. Total offered load for $N$ stations: $G = N \times G_i = \frac{N}{5600}$
3. Apply Pure ALOHA Throughput: The maximum throughput for pure ALOHA is $S_{\max} = \frac{1}{2e} \approx 0.184$.
4. Set Offered Load to Maximum Throughput: $\frac{N}{5600} \leq \frac{1}{2e}$. Solving for $N$: $N \leq \frac{5600}{2e} \approx \frac{5600}{5.436} \approx 1030$
Conclusion: The maximum number of stations $N$ that can effectively share the 56 kb/s pure ALOHA channel is approximately 1030.
**Score:**
1.0 |
| OpenAI-o1-preview | **Output:**
To determine the maximum number of stations $N$ that can share the 56 kb/s pure ALOHA channel without exceeding the maximum acceptable load, we follow these steps:
1. Calculate the Frame Transmission Time ($T_{\text{frame}}$)
The time it takes to transmit one frame is: $T_{\text{frame}} = \frac{\text{Frame Size}}{\text{Channel Bandwidth}} = \frac{1000 \text{ bits}}{56,000 \text{ bits/sec}} = \frac{1}{56}$ sec
2. Determine the Offered Load per Station ($G_{\text{station}}$)
Each station generates frames at an average rate of: $\lambda_{\text{station}} = \frac{1 \text{ frame}}{100 \text{ sec}}$
The offered load per station in frames per frame time is: $G_{\text{station}} = \lambda_{\text{station}} \times T_{\text{frame}} = \left(\frac{1}{100}\right) \times \left(\frac{1}{56}\right) = \frac{1}{5,600}$
3. Calculate the Total Offered Load ($G$)
For $N$ stations, the total offered load is: $G = N \times G_{\text{station}} = N \times \left(\frac{1}{5,600}\right)$
4. Apply the Pure ALOHA Throughput Formula
In pure ALOHA, the maximum throughput occurs when the offered load $G = 0.5$ frames per frame time: $G_{\max} = 0.5$
Setting the total offered load equal to the maximum: $N \times \left(\frac{1}{5,600}\right) = 0.5$
5. Solve for $N$
$N = 0.5 \times 5,600 = 2,800$
Conclusion:
The maximum number of stations $N$ that can share the channel without exceeding the optimal load for pure ALOHA is 2,800 stations.
Answer: 2800
**Score:**
0.3 |

Table 30: An example where GPT-4o and OpenAI-o1-mini failed, OpenAI-o1-preview succeeded.

| Type | Content |
|---|---|
| Question | **Question:**
Given that the 9th level of a complete binary tree has 240 nodes, how many nodes does the entire complete binary tree have? How many leaf nodes are there?
**Reference Answer:**
In a complete binary tree, if the 9th level is full, then the number of nodes = $2^{9-1} = 256$. However, currently, there are only 240 nodes on the 9th level, indicating that the 9th level is not full and is the last level. Levels 1 to 8 are full, so the total number of nodes = $2^8 - 1 + 240 = 495$. Since the 9th level is the last level, all nodes on the 9th level are leaf nodes. Moreover, the parents of the 240 nodes on the 9th level are on the 8th level, with the number of parents being 120, which means there are 120 branch nodes on the 8th level, and the rest are leaf nodes. Therefore, the number of leaf nodes on the 8th level is $2^{8-1} - 120 = 8$. Consequently, the total number of leaf nodes = $8 + 240 = 248$. |
| GPT-4o | **Output:**
In a complete binary tree, each level $k$ has $2^k$ nodes. The 9th level has 240 nodes. Given this, we need to determine the total number of nodes in the tree and the number of leaf nodes.
First, let's confirm that the given information fits the structure of a complete binary tree. The 9th level should have $2^9 = 512$ nodes. Since 240 nodes do not fit this pattern, it implies the tree is incomplete at that level, hence not truly complete. Still, if this were a theoretical exercise, we might infer the tree is complete up until level 8 and partially filled at the 9th level.
1. Calculating the Total Number of Nodes: - Levels 0 through 8 are fully filled, with $2^0 + 2^1 + \cdots + 2^8$ nodes. - The sum of nodes from level 0 to 8 is $2^9 - 1 = 511$. - Adding 240 nodes from the 9th level: $511 + 240 = 751$. So, the total number of nodes in the tree is 751.
2. Calculating the Number of Leaf Nodes: - In a complete binary tree, the leaf nodes are typically the nodes at the last level or partially filled level. - Here, the 9th level only has 240 nodes. These are leaf nodes. So, there are 240 leaf nodes.
Therefore, the entire complete binary tree has 751 nodes and 240 leaf nodes.
**Score:**
0.3 |
| OpenAI-o1-mini | **Output:**
In a complete binary tree where the 9th level has 240 nodes:
- Total Nodes: The first 8 levels are fully filled, containing $2^0 + 2^1 + \cdots + 2^8 = 511$ nodes. Adding the 240 nodes at the 9th level gives a total of $511 + 240 = 751$ nodes.
- Leaf Nodes: All 240 nodes at the 9th level are leaf nodes.
Summary: - Total Nodes: 751 - Leaf Nodes: 240
**Score:**
0.3 |
| OpenAI-o1-preview | **Output:**
To determine the total number of nodes in the complete binary tree and the number of leaf nodes, we can follow these steps:
1. Calculate the total nodes from levels 1 to 8:
In a complete binary tree, levels 1 to 8 are fully filled. The number of nodes at each level doubles starting from 1 at the root.
- Total nodes from level 1 to level 8: $\sum_{i=0}^{7} 2^i = 2^8 - 1 = 256 - 1 = 255$ nodes
2. Add the nodes at level 9:
It's given that there are 240 nodes at level 9.
- Total nodes in the tree: 255 (nodes from levels 1-8) + 240 (nodes at level 9) = 495 nodes
3. Determine the number of leaf nodes:
In a complete binary tree, all nodes at the last level are leaf nodes. Therefore, the number of leaf nodes is equal to the number of nodes at level 9.
- Number of leaf nodes: 240
Answer:
The complete binary tree has 495 nodes in total, and there are 240 leaf nodes.
**Score:**
0.7 |

