# OpenReview forum: "CS-Bench: A Comprehensive Benchmark for Large Language Models towards Computer Science Mastery"
_ICLR.cc/2025/Conference — ICLR 2025 Poster_

### Official Review · Reviewer_Rfxy · 2024-10-21

**Soundness:** 4
**Presentation:** 3
**Contribution:** 3
**Rating:** 8
**Confidence:** 4

**Summary:**

This paper introduces CS-Bench, the most comprehensive and detailed benchmark for testing computer science knowledge and reasoning that exists to date. It contains over 10,000 items over 26 subfields of computer science, uses 4 different response types (multi choice, assertion, fill-in-the-blank, and open-ended responding), and is presented in 4 languages (English, Chinese, German, French). The authors study over 30 LLMs, including both open- and closed-source models. They provide extensive analysis of performance across the benchmark and its subdomains, and conduct post-hoc studies of how performance scales with model size and the kinds of errors that LLMs make on CS-Bench.

**Strengths:**

In the already saturated LLM benchmark market, this paper has a number of stand-out strengths. First, it is highly comprehensive. Unlike other benchmarks which seek generality at the cost of precision and depth, CS-Bench focuses on a single domain and contains thousands of in-depth testing items from it. This is an improvement on existing computer science benchmarks like MMLU. Second, it is multilingual. This facilitates, among other things, the generalisability of deeper CS knowledge and reasoning skills when changing the surface level features of the language the system is operating in. This will also allow users to measure the degree of contamination, since (ex hypothesi) a considerable amount of LLM training data will be from English-language sources. Third, the authors include multiple response types, ranging from multiple choice to open ended responding. This facilitates researchers using CS-Bench to probe CS knowledge and reasoning skills with more nuance. Fourth, the results on logarithmic scaling laws is intriguing. Finally, the extensive error analysis is commendable and often not seen in benchmark papers such as this. It demonstrates again that CS Bench can be used to produce a much richer picture of the knowledge and reasoning skills of LLMs in this domain than can be gleaned by simply aggregating task performance.

**Weaknesses:**

There are a few weaknesses with this paper. First, CS-Bench is static and openly available online. This means that it is possible that LLM developers can train directly on the benchmark, leading to contamination. While we as a community hope that developers wouldn't be so mal-intentioned, it is an increasingly common phenomenon. Dynamic benchmarks (e.g., DynaBench) in which testing items are produced on the fly are able to sidestep such problems.

Second, true tests of computer science skills, particularly for software engineering and development, come from situating models in a coding environment, with access to a terminal. The authors might consider comparing their approach to this more embodied approach to evaluation.

Finally, I am sceptical of the use of GPT-4 for translating the benchmark from Chinese to other languages. While this is certainly scalable, I fear that the translations will be poor, particularly for the most complex questions. The authors' note that some manual checking was done, but I would like to see a more robust measure of translation quality, perhaps through asking a sample of independent native speakers of English, French, and German to check a random sample of questions and answers for soundness.

**Questions:**

1. What do the authors suggest to avoid the issue of contamination?
2. Have the authors considered using answer ordering permutation, number switching, orquestion negations to reduce the risk of contamination? By number switching, I mean dynamically changing the numbers in questions so that the answers are different but the fundamental relationships continue to hold. By question negations, I mean asking the opposite question by including a negative particle, such that the question has a different answer.
3. What is the utility of CS-Bench when compared to approaches in which LLMs are situated in an environment with access to a terminal and tasked to solve problems?
4. Can the authors provide further evidence that the interlingual translations using GPT-4 are sound and accurate?

---

> ### Author Response · Authors · 2024-11-21
> **Response (1/2)**
>
> Thank you for your helpful comments and recognition of our CS-Bench's support for multiple languages and various types, in-depth evaluations, intriguing scaling laws, and extensive error analysis. We will address your concerns point by point below:
>
> ---
>
> ### Weakness 1 & Question 1 & Question 2: CS-Bench is static and openly available online. This may lead to the benchmark being trained on, resulting in data contamination. What do the authors suggest to avoid the issue of contamination, and have the authors considered using some methods for dynamic benchmarking?
>
> It is important to protect the test set from contamination. Your suggestions are very enlightening to us.
>
> In line with your suggestions, on one hand, we prevent the most basic form of test set cheating by randomizing the position of the correct option. On the other hand, considering the diversity and complexity of CS-Bench instances, we find that not all questions are suitable for dynamic reconstruction. Therefore, we carefully select a subset of data suitable for dynamic generation and maintain it as a private data subset. Through this approach, we can dynamically generate test instances to detect whether a model has been privately trained on this bench, thereby mitigating data contamination.
>
> Specifically, we dynamically generate new test instances based on three strategies:
>
> **1. Random Dynamic Numbers**
>
> For calculation-based questions, such as: "How many different permutations of elements are there when {n} distinct elements are pushed onto a stack and then popped off?", we manually define calculation rules like "ans = Catalan number C(n, 2n)/(n+1)". A system sets a predefined range for random numbers, such as n ∈ (5, 10), and dynamically generates a valid number n, which is then substituted into the formula to compute the correct answer. This ensures that the generated question and its answer are both consistent and reasonable.
>
> **2. Statement Rewriting**
>
> For open-ended questions, we use GPT-4 to rewrite the questions multiple times while preserving their original meaning. These rewritten versions are manually validated to form a dynamic question pool. For instance:
>
> - Original question:"In DMA mode, what mechanism is typically triggered to issue an interrupt request after data transfer is complete?"
> - Rewritten version:"What mechanism usually initiates an interrupt request following the completion of data transfer in DMA mode?"
>
> During usage, the system randomly selects one of the rewritten versions as the dynamic question.
>
> **3. Random Semantic Negation**
>
> For Assertions questions, GPT-4 generates semantically negated versions of the original question and reverses the answer, with manual validation. For example:
>
> - Original question:“Mailbox communication is a direct method of communication.”(Answer: False)
> - Negated version:“Mailbox communication is an indirect method of communication.”(Answer: True)
>
> ---
>
> ### Weakness 2 & Question 3: What is the utility of CS-Bench when compared to approaches in which LLMs are situated in an environment with access to a terminal and tasked to solve problems?
>
> As mentioned in the "Related Work" section (lines 509-515), a primary motivation for CS-Bench is that while there have been some works exploring the application of LLMs in specific scenarios such as software engineering and terminal environments, a comprehensive assessment of LLMs' more fundamental knowledge and reasoning abilities in computer science has been consistently lacking. Therefore, CS-Bench focuses on comprehensively evaluating LLMs' knowledge and reasoning abilities in computer science, rather than being limited to specific application scenarios.
>
> We believe this is both a strength and a bias of CS-Bench. Combining CS-Bench with studies targeting specific application scenarios can provide a richer perspective. We elaborated on this point in detail in Appendix A: Limitations and Dataset Bias.

---

> > ### Author Response · Authors · 2024-11-21
> > **Response (2/2)**
> >
> > ### Weakness 3 & Question 4: The quality of using GPT-4 to translate the benchmark from Chinese to other languages may be relatively poor, although some manual checking was done. Can the authors provide further evidence that the interlingual translations using GPT-4 are sound and accurate?
> >
> > Our primary method of checking the translation quality of GPT-4 is back-translation, which involves independently translating the translated text back into Chinese and comparing it to the original to identify any discrepancies.
> >
> > To further validate the quality of multilingual data processed by GPT-4 and human verification, we randomly select 30 pairs of instances with identical content in different languages. For each language, we hire a data annotator who is either a native speaker or hold an advanced language certificate, such as the Common European Framework of Reference for Languages C1 (CEFR C1). They independently evaluated the translations based on the following scoring system:
> >
> > - 1 point: Poor translation quality, difficult to convey the original meaning, significantly affecting the ability to answer questions;
> > - 2 points: The translation conveys the general meaning of the original text, minimally affecting the ability to answer questions, but lacks fluency and accuracy;
> > - 3 points: Generally correct translation with minor word choice issues, accurately reflects the original meaning without affecting the ability to answer questions;
> > - 4 points: Correct translation with good fluency;
> > - 5 points: Very accurate translation with precise word choice.
> >
> > The statistical results are as follows:
> >
> > | Language | 1 point | 2 points | 3 points | 4 points | 5 points |
> > | --- | --- | --- | --- | --- | --- |
> > | English | 0 | 0 | 0 | 4 (13.3%) | 26 (86.7%) |
> > | German | 0 | 0 | 1 (3.33%) | 7 (23.33%) | 22 (73.3%) |
> > | French | 0 | 0 | 0 | 8 (26.7%) | 22 (73.3%) |
> >
> > Based on the statistical results, the vast majority of the data achieve correct and fluent translations (scoring at least 4 points). All data score at least 3 points (potentially lacking fluency but not affecting the ability to answer questions). Therefore, we consider this level of translation quality to be acceptable and believe it has minimal impact on the evaluation of LLMs. Besides, we will continuously re-examine our data to avoid rare instances of low-quality translations.
> >
> > We hope this addresses your concerns.

---

> > > ### Comment · Reviewer_Rfxy · 2024-11-25
> > >
> > > Thank you for your response to my comments and for the extensive work you have done to address my concerns. Your strategies to handle data contamination are exactly what I would have done, and are, arguably, the most one can feasibly do. At some point, we must simply trust that developers will not intentionally train their models on evaluation benchmarks.
> > >
> > > I agree that CS-Bench is a complement to the more agentic evaluation methods that are on the market already, although I worry that ultimately, when it comes to deployment decisions, we will care more about systems that can interact with a terminal while employing their CS knowledge more than systems that just show case their knowledge in question-answer settings. In a similar way, companies hiring software engineers usually interview candidates in an interactive coding environment rather than testing knowledge with multiple-choice, since it is more indicative of how the candidate will actually perform on the job. Nevertheless, CS-Bench is a stepping stone and complementary evaluation tool to these other methods.
> > >
> > > Your validation of the translation quality is impressive and the results are satisfying. I am happy that the translations are good enough quality for an evaluation scenario, but I would advise caution when interpreting any results indicating lower performance in English, French, or German. These may be due to inconsistent, disfluent, or inaccurate translations that ought to be remedied.

---

> > > > ### Author Response · Authors · 2024-11-25
> > > > **Thank you for Reviewer's reply**
> > > >
> > > > We are pleased to see that our response has alleviated your concerns. Your suggestions have all been very helpful to us. Thank you again for your thorough review and suggestions regarding our work.

---

### Official Review · Reviewer_g1yj · 2024-11-01

**Soundness:** 3
**Presentation:** 3
**Contribution:** 3
**Rating:** 6
**Confidence:** 4

**Summary:**

This paper introduces CS-Bench, a comprehensive, multilingual benchmark designed to evaluate the computer science knowledge and reasoning abilities of Large Language Models (LLMs). Comprising approximately 10,000 meticulously curated samples across 26 subfields within four key CS areas (Data Structures and Algorithms, Computer Organization, Computer Networks, and Operating Systems), CS-Bench supports evaluation in English, Chinese, French, and German, and includes diverse task formats such as multiple-choice, assertion, fill-in-the-blank, and open-ended questions. The authors conduct an extensive evaluation of over 30 mainstream LLMs on CS-Bench, analyzing their performance relative to model scale and investigating the reasons for model failures. They further explore the relationship between LLM capabilities in computer science, mathematics, and coding, observing strong correlations and evaluating the performance of specialized expert LLMs on the benchmark. The findings suggest a logarithmic growth relationship between model scale and CS performance, highlight the challenge of CS-specific reasoning for LLMs, and reveal the potential for expert LLMs to improve performance in certain CS subfields. CS-Bench aims to serve as a valuable resource for assessing and improving the diverse reasoning capabilities of LLMs in the context of computer science.

**Strengths:**

Focus on Reasoning: The inclusion of both knowledge-based and reasoning-based questions is a good step towards a more comprehensive evaluation of LLMs' understanding of CS concepts.

Extensive Evaluation: Testing numerous LLMs, both open-source and closed-source, across various scales provides a good overview of current capabilities and highlights areas where models struggle. The scale-score analysis is also a useful contribution.

Comprehensive Benchmark: The breadth of CS topics covered, along with the multilingual aspect and diverse task formats (multiple-choice, assertion, fill-in-the-blank, open-ended), makes this a valuable contribution. Creating such a benchmark is a substantial undertaking.

Error Analysis: The analysis of error types provides some insights into the reasons for model failures and points towards potential directions for improvement.

**Weaknesses:**

Writing and Presentation Could Be Improved: The writing is occasionally dense and could be more concise and clear. Some figures, especially the radar charts, are difficult to interpret and don't effectively communicate the key findings. A more visual and intuitive presentation of the results would enhance readability and impact.

Limited Insight into Cross-Capability Relationships: While the paper shows correlations between CS, math, and coding abilities, it doesn't delve deeply into why these correlations exist. More in-depth analysis is needed to understand the underlying factors driving these relationships. Do specific architectural choices or training data characteristics explain the observed correlations? The analysis of expert LLMs is a step in the right direction, but it lacks depth. How do the internal representations of CS concepts differ between general and expert LLMs?

Lack of Actionable Insights for Improvement: The error analysis, while pointing towards knowledge gaps and reasoning challenges, doesn't offer concrete suggestions for how to address these limitations. Beyond general recommendations like "knowledge supplementation" and "CS-specific reasoning," what specific techniques or strategies could be employed to improve LLM performance on CS tasks? Exploring methods like curriculum learning, incorporating external knowledge sources, or fine-tuning on specific CS datasets could provide more actionable insights.

Limited Discussion of Benchmark Biases: All benchmarks have biases. What are the potential biases in CS-Bench, and how might these biases influence the results? For example, the reliance on university-level content might not fully reflect the diversity of CS tasks in real-world applications. Addressing this limitation by including tasks from different levels of expertise or from specific industry applications would strengthen the benchmark's generalizability.

**Questions:**

Reducing Reliance on GPT-4 for Evaluation: The dependence on GPT-4 for scoring generative tasks raises validity and reproducibility concerns. While you explored alternative scoring models and human evaluation, these were limited. Could you elaborate on the feasibility of using alternative automatic metrics like BLEU, ROUGE, or METEOR, perhaps in conjunction with domain-specific constraints or regular expressions? Could a smaller, more accessible model be fine-tuned specifically for evaluating CS-related generative tasks? This would improve reproducibility and lower the barrier for other researchers to use CS-Bench. Also, provide more details on the human evaluation: the number of annotators, their expertise level, and the specific guidelines they followed. Calculating inter-annotator agreement (e.g., using Cohen's kappa) would strengthen the validation of both human and GPT-4 scoring.

Deepening the Analysis of Cross-Capability Relationships: The observed correlations between CS, math, and coding performance are interesting, but the analysis lacks depth. Could you investigate why these correlations exist? Are certain model architectures better suited for integrating these skills? Does the training data composition play a significant role? Analyzing the internal representations of CS concepts in general vs. expert LLMs could provide valuable insights. Consider using techniques like probing tasks or representational similarity analysis to understand how different models learn and represent CS knowledge. This would move beyond simple correlations to provide a more mechanistic understanding of cross-capability transfer.

Generating Actionable Insights for Improvement: The error analysis identifies knowledge gaps and reasoning challenges, but it doesn't offer concrete solutions. How can we leverage these insights to improve LLM performance on CS tasks? Could you explore targeted fine-tuning on CS-specific datasets, curriculum learning strategies, or methods for incorporating external knowledge sources (e.g., knowledge graphs, symbolic reasoning systems)? Providing concrete examples of how these techniques could be applied would significantly enhance the paper's practical value.

Addressing Benchmark Biases: What are the potential biases introduced by the composition of CS-Bench (e.g., university-level focus, source selection)? How might these biases affect the evaluation of different LLMs? Consider expanding the benchmark to include tasks from different levels of expertise (e.g., introductory CS, advanced topics, real-world applications) and diversifying the sources to mitigate potential biases. Discuss the limitations of CS-Bench in representing the full spectrum of CS knowledge and skills. A critical self-reflection on the benchmark's limitations would strengthen the paper.

Improving Visualization and Clarity: Some figures, particularly the radar charts, are difficult to interpret. Consider using alternative visualizations (e.g., bar charts, scatter plots) that more clearly communicate the key findings. Streamline the writing to improve conciseness and clarity. Focus on conveying the key messages effectively and avoid unnecessary jargon. Ensure that the figures are clearly labeled and explained in the captions. A more polished presentation would enhance the paper's impact and accessibility.

---

> ### Author Response · Authors · 2024-11-21
> **Response (1/4)**
>
> Thank you for your valuable comments and recognition of our work in proposing a comprehensive CS benchmark and conducting an extensive evaluation and analysis. We will address your concerns point by point below:
>
> ---
> ### Weakness 1 & Question 5: Some writing and charts are occasionally dense, especially the radar charts, which lack simplicity and clarity.
>
> Thank you for your suggestions on our writing and visualization. We recognize that while Radar Chart 3 in the main text aimed to compare different models' capabilities across various aspects, it indeed lacked clarity. In our new manuscript, we've replaced it with a bar chart, keeping the original radar chart only in the appendix. We'll also continue to refine our writing to enhance overall quality.
>
> ---
> ### Weakness 2 & Question 2: While the paper shows correlations between CS, math, and coding abilities, deeper analysis is needed to understand the underlying factors driving these relationships, such as data characteristics and internal representations.
>
> To analyze the reasons behind capability correlations, we first explore the relationships between different abilities of general models at the representation level. We extract the last hidden layer of Llama3-8B-instruct and obtain representations for different domains through average pooling. We then calculate the average cosine distance between CS-Bench reasoning-type representations and other data representations, where smaller distances indicate greater similarity. The results are as follows:
>
> | Dataset | Domain | Distance |
> | --- | --- | --- |
> | MBPP | code | 0.4645 |
> | MATH | math | 0.5395 |
> | BBH | reasoning | 0.5494 |
> | T-Eval | tool | 0.6310 |
> | Chatbot Arena (first turn) | chat | 0.5555 |
> | GPQA | science | 0.5775 |
> | L-eval | long context | 0.6409 |
>
> We find that the similarity of representations between different abilities/domains generally aligns with the observed capability correlations in Figure 9 , especially in the areas of code and math. This explains the correlation between CS skills and math and coding abilities at the representation level. Additionally, due to BBH containing various reasoning data, the distance between CS-Bench and BBH is relatively small, which is consistent with the analysis in lines 465-467 of the manuscript.
>
> Next, we analyze from the perspective of training data characteristics. We intuitively find that some questions in CS-Bench can be understood as mathematical problems in a computer science context. For example:
>
> ```
> Q1: If the data part is 3800B and the maximum fragment size is 1420B, what is the total length in bytes of the third fragment?
> Q2: The height h of a binary tree with 1025 nodes is ().
> ```
>
> , while the main role of code is to supplement the model's knowledge of data structures and logical reasoning abilities. This data-level correlation explains the expert model experimental results in Tables 3 and 4, specifically why math and code expert models help improve CS capabilities in certain areas, even when their general abilities are weakened.
>
> Finally, we analyze the relationship between general models and  math & code expert models in computer science representations. We select the chat model Llama2-7b-Chat, math expert model MAmmoTH, and code expert model CodeLlama, all based on Llama2-7b-base, to observe their average cosine distances for the same data points in CS-Bench.
>
> |  | between Chat and Math model | between Chat and Code model | between Math and Code model |
> | --- | --- | --- | --- |
> | distance | 0.2381 | 0.5432 | 0.6456 |
>
> Results show that expert models' representations shift to some extent compared to the chat model, especially the code model, which can be attributed to Code's unique data patterns. Further representation visualization in Figure 19 of manuscript reveals that in the general dataset chatbot-arena, the features of the three models do not significantly separate. However, in math and csbench datasets, the code model forms a distinct feature cluster, while math and chat models' representations remain similar, as we stated: some questions in CS can be viewed as mathematical problems in a CS context. Lastly, in the MBPP dataset, the representations of the three models clearly differentiate and form their own clusters.

---

> > ### Author Response · Authors · 2024-11-21
> > **Response (2/4)**
> >
> > ### Weakness 3 & Question 3：While error analysis identified knowledge gaps and reasoning challenges, it did not offer concrete strategies to mitigate these limitations or improve LLM performance on CS tasks.
> >
> > The core objective of this work is to introduce CS-Bench for a comprehensive evaluation of LLM performance in CS, analyze their shortcomings and challenges, and provide valuable insights for improving LLMs. Although the specific methods to enhance LLM performance in CS are not the primary focus of this paper, we are eager to explore potential solutions based on our findings. These include capability transfer, reasoning frameworks, integrating external knowledge sources, incorporating symbolic reasoning systems, and targeted fine-tuning.
> >
> > **1. Capability Transfer:**
> >
> > The capability analysis experiments in Section 3.5 demonstrate that math and code expert models can improve CS performance in certain areas. Therefore, in scenarios where CS data is scarce, training on math or code tasks can be leveraged to transfer and enhance CS capabilities.
> >
> > **2. Reasoning Frameworks:**
> >
> > The experiments in Section 3.3 show that compared to GPT-4o, OpenAI-o1 significantly improves the reasoning scores on CS-Bench. This suggests that combining LLMs with reasoning frameworks can be expected to enhance reasoning capabilities. However, this approach comes with the trade-off of increased token consumption during inference.
> >
> > **3. Integrating External Knowledge Sources:**
> >
> > The error type analysis in Section 3.4 identifies knowledge gaps as the main reason for LLM failures in the CS domain. To address this, we implemented a RAG framework that uses Wikipedia as an external knowledge source. The process involves extracting key terms from the question, retrieving relevant information from Wikipedia, and summarizing it to provide supplemental knowledge, which is then integrated into the context for answering the question. The results are as follows:
> >
> > | model | knowledge | reasoning | overall |
> > | --- | --- | --- | --- |
> > | GPT-3.5-turbo | 63.04 |  43.45 | 55.91 |
> > | **GPT-3.5-turbo（+RAG)** | **64.92(+1.88)** | **48.68(+5.23)** | **59(+3.09)** |
> > | Llama3-8b-Instruct | 59.75 | 44.97 | 54.37 |
> > | **Llama3-8b-Instruct(+RAG)** | **66.11(+6.36)** | **50.64(+5.67)** | **60.47(+6.1)** |
> >
> > Both knowledge and reasoning scores of the model have improved significantly, aligning with the findings in lines 411-418 of the paper: the primary errors in knowledge and reasoning questions are knowledge-related issues. Performance is expected to be further enhanced when combined with a more advanced RAG framework or more suitable knowledge sources.
> >
> > **4. Incorporating Symbolic Reasoning Systems:**
> >
> > To enable joint reasoning between language and symbolic systems, we developed a Python API as a symbolic reasoning tool. This allows the LLM to generate Python code when it identifies a need for symbolic reasoning. The generated code is executed using the Python tool, and the results are reintegrated into the LLM's context for further output generation. The results are as follows:
> >
> > |  | knowledge | reasoning | overall |
> > | --- | --- | --- | --- |
> > | gpt-4o-0513 | 76.95 | 64.15 | 72.29 |
> > | **gpt-4o-0513(+python Tool)** | **77.47(+0.5)** | **71.86(+7.71)** | **75.42(+3.13)** |
> >
> > Since symbolic reasoning focuses on improving reasoning tasks, we observe that there is little improvement in the model's knowledge capabilities, but there is a significant enhancement in its reasoning performance.
> >
> > **5. Fine-tuning**:
> >
> > The error analysis also highlighted the need for specialized fine-tuning for CS (lines 418-420). Due to the scarcity of fine-tuning data in the CS domain, we randomly sampled 10% of the English data from CS-Bench as a mini test set. The remaining data was converted into SFT training data and used to fine-tune Llama3-8b-instruct. The results on the miniTest set are as follows:
> >
> > | miniTest | knowledge | reasoning | overall |
> > | --- | --- | --- | --- |
> > | Llama3-8b-Instruct | 66.89 | 43.73 | 58.07 |
> > | **Llama3-8b-Instruct（+SFT)** | **71.19(+4.3)** | **49.88(+6.15)** | **63.07(+5)** |
> >
> > Despite the challenges posed by the difficulty of CS tasks and the limited size of the training set, the model's knowledge and reasoning capabilities improved after fine-tuning. We expect that fine-tuning with a larger amount of domain-specific data could further enhance the LLM's performance in CS.
> >
> > Through these discussions, we aim to provide insights and directions for the future development of LLMs in the field of computer science.

---

> > > ### Author Response · Authors · 2024-11-21
> > > **Response (3/4)**
> > >
> > > ### Weakness 4 & Question 4: The discussion of CS-Bench biases is limited. What are the potential biases in CS-Bench, and how might these biases influence the results?
> > >
> > > Our discussion on the biases of CS-Bench is primarily addressed in Appendix A Limitations (lines 918-927). As you noted, all benchmarks inherently have biases, and the main biases of CS-Bench can be categorized into two aspects:
> > >
> > > 1. **Difficulty Level:** Overall, the benchmark reflects a university-level difficulty.
> > > 2. **Task Focus:** It emphasizes knowledge- and reasoning-based questions rather than specific real-world production scenarios in computer science.
> > >
> > > The impact of difficulty on LLM evaluation is: lower difficulty tends to narrow the score differences between models, while higher difficulty amplifies these differences. It is worth noting that although CS-Bench maintains an overall university-level difficulty, the diversity of difficulty across questions remains significant, ranging from simple definitions in data structures to challenging computer network application problems. This is reflected in the model score range of 39.86% to 72.29%, demonstrating CS-Bench's effectiveness in distinguishing LLMs' capabilities in computer science and presenting challenges even for the best-performing models.
> > >
> > > Regarding task scenarios, a key motivation for CS-Bench is that “some prior work has explored LLMs in specific applications such as cybersecurity and software engineering.” However, as stated in the manuscript's related work section (lines 509-515), there has been a lack of comprehensive evaluations of LLMs' fundamental knowledge and reasoning capabilities in computer science.
> > >
> > > We have incorporated this discussion into the revised version of Appendix A Limitations and Dataset Bias (lines 993-1008).

---

> > > > ### Author Response · Authors · 2024-11-21
> > > > **Response (4/4)**
> > > >
> > > > ### Question 1: Consider you elaborate on the feasibility of using alternative automatic metrics or fine-tuning smaller models for evaluating CS-related generative tasks? Also, provide more details on the human evaluation.
> > > >
> > > > As shown in Section 3.1, CS-Bench uses a mixed scoring approach that combines regular expressions and LLM (GPT-4) to score comprehension tasks and generation tasks separately. To validate the reasonableness of LLM-based scoring, we explored the consistency between LLM scoring and human evaluation, as well as the impact of different scoring models, in Appendix D2.
> > > >
> > > > Next, we first explore the feasibility of rule-based traditional metrics and specific regex-based methods as alternatives to LLM scoring.
> > > >
> > > > **1. The specific details of human evaluation are as follows:**
> > > >
> > > > For the fill-in-the-blank (FITB) and Open-ended questions, we randomly selected 100 output instances and had three human annotators independently score the model outputs. These three annotators all hold bachelor's degrees in computer science. Their scoring criteria were consistent with the evaluation standards provided to GPT-4 and can be found in Table 13 (lines 1405-1451).
> > > >
> > > > In Table 11, we have reported the Cronbach’s alpha consistency among human annotators and further report Cohen's kappa scores below. Since Cohen's kappa is applicable only to discrete data, we mapped the continuous subjective question scores to intervals based on the scoring criteria in Table 13. The results are as follows:
> > > >
> > > > | Human- Human Consistency | Cohen's kappa | Cronbach’s Alpha |
> > > > | --- | --- | --- |
> > > > | FITB | 0.9186 | 0.9311 |
> > > > | Open-ended | 0.8959 |  0.9751 |
> > > >
> > > > **2. Rule-based traditional metrics & specific regex-based metrics:**
> > > >
> > > > On one hand, we use traditional metrics such as BLEU, ROUGE, and METEOR to score FITB and open-ended questions. On the other hand, we apply domain-specific regex-based methods. We prompt GPT-4 to extract key keywords from an answer and generate synonymous phrases, followed by manual verification, to create a list of keywords for regex matching for each question.
> > > >
> > > > We validate the metrics' validity through Pearson correlation with human evaluations. To align with human scoring rules, we set threshold t to map these continuous metrics to discrete scores. We perform a grid search for the optimal t for each metric and present the results as follows:
> > > >
> > > > | Metric | BLEU | ROUGE-1 | ROUGE-2 | ROUGE-L | METEOR | CS-Special Regex |
> > > > | --- | --- | --- | --- | --- | --- | --- |
> > > > | Consistency | 0.0977 | 0.5949 | 0.2667 | 0.4848 | 0.5678 | 0.5982 |
> > > >
> > > > The experimental results show that even after searching for the optimal threshold t, the Pearson correlation coefficients of these metrics do not exceed 0.60, indicating poor consistency with human annotations. This suggests that for generation and open-ended tasks, regex/rule-based evaluation metrics struggle to achieve reliable discrimination. This is because regex methods can only perform surface-level matching and score calculation, while LLM and human evaluation standards can involve deeper semantic and multi-dimensional understanding.
> > > >
> > > > **3. Expert Model-Based Scoring:**
> > > >
> > > > To reduce the cost and reproducibility issues associated with using GPT-4 for evaluation, we further explore the feasibility of fine-tuning a smaller expert model as an alternative to LLM scoring.
> > > >
> > > > Specifically, we construct a training set using the questions and answers from CS-Bench, outputs from several models listed in Table 2, and GPT-4 scores. We then test the Pearson correlation between the scores of the new models and human evaluations. The results are as follows:
> > > >
> > > > | Consistency with human | Fill-in-the-blank | Open-ended |
> > > > | --- | --- | --- |
> > > > | GPT-4 | 0.8080 | 0.9494 |
> > > > | Llama3.2-1B-Instruct (+SFT) | 0.7278 | 0.7409 |
> > > > | Llama3.2-3B-Instruct (+SFT) | 0.7407 | 0.8544 |
> > > > | Llama3.1-8B-Instruct (+SFT) | 0.7735 | 0.8663 |
> > > >
> > > > We observe a significant improvement in consistency between expert open-source model scores and human evaluations compared to rule-based methods. Additionally, the accuracy of the scores is positively correlated with the model's parameter size. However, expert models still perform weaker than GPT-4, especially in evaluating open-ended questions, where GPT's generalization ability is more prominent. Further analysis reveals that, due to GPT-4 distillation learning, the expert model closely mimics the scoring consistency of GPT-4:
> > > >
> > > > | Consistency with GPT-4 | Fill-in-the-blank | Open-ended |
> > > > | --- | --- | --- |
> > > > | Llama3.2-1B-Instruct (+SFT) | 0.8459 | 0.7163 |
> > > > | Llama3.2-3B-Instruct (+SFT) | 0.9215 | 0.9117 |
> > > > | Llama3.1-8B-Instruct (+SFT) | 0.9615 | 0.9279 |
> > > >
> > > > To achieve higher consistency, we will next incorporate some human labels into the training data and apply filtering to the GPT-4 labels to improve both consistency and usability. This will serve as an open-source, stable alternative to GPT-4 scoring.

---

> ### Author Response · Authors · 2024-11-26
> **A Kind Reminder for Reading the Response**
>
> Dear Reviewer g1yj,
>
> Thank you for your insightful suggestions. We have done our best to address your concerns. Since the rebuttal period is closing very soon, could you please check the response to see whether it mitigates your concerns? We would greatly appreciate that!
>
> Thank you for your time and consideration, the authors.

---

> > ### Comment · Reviewer_g1yj · 2024-11-27
> > **Concerns addressed**
> >
> > Awfully sorry for the silence, thank you addressing the concerns, currently going over the manuscript afresh and I will add my thoughts in case but so far so good. Cheers!

---

> > > ### Author Response · Authors · 2024-11-28
> > > **Response for Reviewer g1yj (Follow up)**
> > >
> > > Thank you for your reply.
> > >
> > > We would like to know if our response and revised manuscript have well addressed your concerns.
> > > If so, we would greatly appreciate it if you could reconsider the score.
> > > If there are any remaining concerns or suggestions, we are more than willing to discuss further and make corresponding improvements.
> > >
> > > With kind regards,
> > >
> > > The Authors

---

### Official Review · Reviewer_7dpp · 2024-11-01

**Soundness:** 3
**Presentation:** 3
**Contribution:** 1
**Rating:** 5
**Confidence:** 4

**Summary:**

The paper introduces a benchmark of 10k test samples that pertain to Computer Science specific knowledge.

**Strengths:**

1. Commendable that 70% of the benchmark is from private source not already on the intrernet.

2. The paper has a somewhat thorough review of several LLMs on where they stand in the benchmark.

**Weaknesses:**

1. It is not clear what the benchmark aims to achieve or improve. LLMs are already good at MCQ, open-ended, FITB and T/F questions. There is no coding in the benchmark. The authors need to discuss more clearly why this is pivotal and important to the field.

2. Scoring on the benchmark appears to be ad-hoc with regular expression matching, instead of measuring with existing techniques.

3. The benchmark lacks a fingerprint to prevent training. THis would be required to protect the integrity of the benchmark.

4. The paper indicates that this may help evaluate or improve reasoning though it is unclear how?

**Questions:**

Q1. What would the key contributions of the benchmark be to the field at large, as it stands today?

Q2. Can the authors evaluate the value of the data through one of several Value-of-Data approaches including perplexity of training on the benchmark, FT-ing on the benchmark or other approaches?

---

> ### Author Response · Authors · 2024-11-21
> **Response (1/4)**
>
> Thank you for your valuable comments and recognition of our work in data quality and thorough review. Next we will now address your concerns one by one.
>
> ---
>
> ### **Weakness 1:**  It is not clear what the benchmark aims to achieve or improve. LLMs are already good at MCQ, open-ended, FITB and T/F questions. There is no coding in the benchmark. The authors need to discuss more clearly why this is pivotal and important to the field.
>
> Thank you for your question. We will address it in four parts: the motivation and importance of CS-Bench, goals of CS-Bench, LLMs are already good at multiple questions, and why programming questions were not included.
>
> **Motivation and Importance of CS-Bench:**
>
> CS-Bench is motivated by two key factors: "understanding LLMs' performance in the CS field" and "CS can effectively support cross-capability analysis".
>
> First, CS is a critical and rapidly evolving field. Given that LLMs have shown promising results in other scientific domains like chemistry and biology, as well as in specific CS scenarios such as code completion and cybersecurity, we believe applying LLMs effectively in CS can drive innovation, solve complex problems, and improve efficiency (lines 35-43, 501-515). To achieve this, it is essential to deeply understand the strengths and limitations of LLMs in CS. This understanding can guide their application in appropriate scenarios and inform further improvement and optimization of LLMs (lines 45-50). Empirical studies on LLM performance in CS are among the most effective ways to achieve this goal.
>
> Secondly, considering the intersection of CS with coding, mathematics, and reasoning abilities, we have reason to believe that computer science can effectively support cross-capability analysis and research, addressing the community's shortcomings in this area (lines 50-80).
>
> **Goals of CS-Bench:**
>
> Building on the motivation above, CS-Bench focuses on two key objectives, as outlined in RQ1 and RQ2 in the introduction:
>
> 1. To empirically explore the performance of LLMs in CS, identifying their challenges and potential improvement directions.
> 2. To investigate the relationships between LLM capabilities in CS, mathematics, and coding, supporting cross-capability analysis.
>
> **LLMs are already good at multiple questions:**
>
> Although LLMs perform well in tasks like knowledge-based questions and common-sense reasoning, we cannot directly infer that they are equally proficient in the field of CS. One of the goals of our work is to empirically analyze this question (lines 45-50). Within the CS field, CS-Bench includes various question types to simulate different scenarios in CS. The experimental results in Table 2 confirms that even GPT-4 scores only 72.29 overall on CS-Bench, with a reasoning score of just 64.15. For weaker models like Llama3-8B and Llama2-70B, the scores are close to 50, falling far short of being "good."
>
> **Exclusion of Programming Questions:**
>
> Considering the extensive research already conducted by the LLM community on coding abilities, we view programming as a separate focus area. Therefore, CS-Bench does not include programming questions but instead focuses on the relationship between CS capabilities and coding abilities.
>
> ---
>
> ### Weakness 2: Scoring on the benchmark appears to be ad-hoc with regular expression matching, instead of measuring with existing techniques.
>
> CS-Bench employs a hybrid scoring approach combining regular expressions and LLM evaluation. As described in Section 3.1 (lines 183-195), regular expressions are used for comprehension tasks (MC and Assertions), while GPT-4 scores generative tasks (FITB and Open-ended questions) based on reference answers.
>
> The motivation for this hybrid approach is as follows: for comprehension tasks, regular expressions reduce the evaluation complexity and enhance result stability. However, for generative tasks, the diversity of valid answers makes regular matching inadequate for accurate evaluation. Regular expressions only assess text at the surface level, while LLM-based evaluation, grounded in semantic understanding and multidimensional criteria, aligns better with human judgment. Appendix D2 validates GPT-4's scoring reliability through consistency with human ratings (Table 11) and examines the impact of using different scoring models (Table 12).
>
> We further provide the Pearson correlation between regular expression-based or traditional scoring metrics and human ratings for generative tasks. The results are as follows:
>
> | Metric | BLEU | ROUGE-1 | ROUGE-2 | ROUGE-L | METEOR | CS-Special Regex |
> | --- | --- | --- | --- | --- | --- | --- |
> | Consitency | 0.0977 | 0.5949 | 0.2667 | 0.4848 | 0.5678 | 0.5982 |
>
> The results show that the Pearson correlation coefficients for these metrics do not exceed 0.60, indicating poor consistency with human annotations. This further supports the rationale for adopting a hybrid evaluation approach combining regular expressions and LLM scoring.

---

> > ### Author Response · Authors · 2024-11-21
> > **Response (2/4)**
> >
> > ### Weakness 3: The benchmark lacks a fingerprint to prevent training.
> >
> > Protecting data from leakage is crucial. Incorporating the suggestions from Reviewer Rfxy, we take two key measures: first, we randomize the position of correct answers to prevent basic test set overfitting. Second, **we maintain a dynamic private subset of the test data.** By dynamically generating test items, we can detect whether a model has been specifically trained on the CS-Bench dataset, mitigating such risks.
> >
> > Specifically, we dynamically generate new test instances based on three strategies:
> >
> > **1. Random Dynamic Numbers**
> >
> > For calculation-based questions, such as: "How many different permutations of elements are there when {n} distinct elements are pushed onto a stack and then popped off?", we manually define calculation rules like "ans = Catalan number C(n, 2n)/(n+1)". A system sets a predefined range for random numbers, such as n ∈ (5, 10), and dynamically generates a valid number n, which is then substituted into the formula to compute the correct answer. This ensures that the generated question and its answer are both consistent and reasonable.
> >
> > **2. Statement Rewriting**
> >
> > For open-ended questions, we use GPT-4 to rewrite the questions multiple times while preserving their original meaning. These rewritten versions are manually validated to form a dynamic question pool. For instance:
> >
> > - Original question:"In DMA mode, what mechanism is typically triggered to issue an interrupt request after data transfer is complete?"
> > - Rewritten version:"What mechanism usually initiates an interrupt request following the completion of data transfer in DMA mode?"
> >
> > During usage, the system randomly selects one of the rewritten versions as the dynamic question.
> >
> > **3. Random Semantic Negation**
> >
> > For Assertions questions, GPT-4 generates semantically negated versions of the original question and reverses the answer, with manual validation. For example:
> >
> > - Original question:“Mailbox communication is a direct method of communication.”(Answer: False)
> > - Negated version:“Mailbox communication is an indirect method of communication.”(Answer: True)
> >
> > During evaluation, a randomly selected version and its corresponding answer are used.
> >
> > Using this dynamic private subset, we can effectively detect whether a model has been trained on CS-Bench, ensuring the integrity and security of the benchmark.

---

> > > ### Author Response · Authors · 2024-11-21
> > > **Response (3/4)**
> > >
> > > ### Weakness 4: The paper indicates that this may help evaluate or improve reasoning though it is unclear how?
> > >
> > > CS-Bench includes a diverse set of CS-specific reasoning problems to evaluate a model's ability to apply computer science knowledge for reasoning. The results in Table 3 show that models perform poorly on CS reasoning tasks, with Llama3-8B scoring 44.97 and GPT-4 achieving 64.15.
> > >
> > > In the error analysis in Section 3.4, we conducted a detailed examination of the errors made by models on CS-Bench, including knowledge-related mistakes and CS-specific reasoning errors. These findings highlight weaknesses in the reasoning processes of the models and provide concrete directions for improvement.
> > >
> > > Additionally, the capability analysis experiments in Section 3.5 demonstrate that math and code expert models can improve CS performance in certain areas. This suggests that in scenarios where CS data is scarce, training on math or code tasks can transfer and enhance CS performance. By strengthening a model's abilities in specific domains, such as mathematics or programming, we can indirectly improve its reasoning capabilities in the CS domain.
> > >
> > > Building on the error analysis that highlights knowledge gaps and reasoning challenges, we further explore specific solutions:
> > >
> > > **1. Integrating External Knowledge Sources:**
> > >
> > > Error Type Analysis reveals that knowledge deficiencies are the primary reason for LLM failures in CS tasks. To address this, we implement a Retrieval-Augmented Generation (RAG) framework. Wikipedia is used as an external knowledge source: key terms from the questions are first extracted, and relevant content is retrieved and summarized. This supplemental knowledge is then incorporated into the question's context for answering. The results are as follows:
> > >
> > > | model | knowledge | reasoning | overall |
> > > | --- | --- | --- | --- |
> > > | GPT-3.5-turbo | 63.04 |  43.45 | 55.91 |
> > > | **GPT-3.5-turbo（+RAG)** | **64.92(+1.88)** | **48.68(+5.23)** | **59(+3.09)** |
> > > | Llama3-8b-Instruct | 59.75 | 44.97 | 54.37 |
> > > | **Llama3-8b-Instruct(+RAG)** | **66.11(+6.36)** | **50.64(+5.67)** | **60.47(+6.1)** |
> > >
> > > Both knowledge and reasoning type scores improve significantly, aligning with the findings in lines 411-418: the primary errors in knowledge and reasoning questions are rooted in knowledge deficiencies. When using more advanced RAG frameworks or more suitable knowledge sources, performance is expected to be further enhanced.
> > >
> > > **2. Incorporating Symbolic Reasoning Systems:**
> > >
> > > To enable joint reasoning with language and symbolic systems, we implement a Python API as a symbolic reasoning tool. The framework allows the LLM to generate Python code when it determines that symbolic reasoning is required. The generated code is executed using the Python tool, and the results are reintegrated into the LLM's context for further response generation. The results are as follows:
> > >
> > > |  | knowledge | reasoning | overall |
> > > | --- | --- | --- | --- |
> > > | gpt-4o-0513 | 76.95 | 64.15 | 72.29 |
> > > | **gpt-4o-0513(+python Tool)** | **77.47(+0.5)** | **71.86(+7.71)** | **75.42(+3.13)** |
> > >
> > > Since symbolic reasoning focuses on addressing reasoning-specific problems, we observe that LLM performance in knowledge-based tasks showed little improvement, while there was a significant improvement in reasoning tasks.
> > >
> > > **3. Targeted Fine-Tuning:**
> > >
> > > Error analysis also highlights the necessity of optimizing LLMs specifically for CS tasks (lines 418-420). Due to the scarcity of fine-tuning data in the CS field, we randomly sample 10% of the English data from CS-Bench as a mini test set, with the remaining data converted into SFT training data. This dataset is used to fine-tune the Llama3-8b-Instruct. The results on the mini test set are as follows:
> > >
> > > | miniTest | knowledge | reasoning | overall |
> > > | --- | --- | --- | --- |
> > > | Llama3-8b-Instruct | 66.89 | 43.73 | 58.07 |
> > > | **Llama3-8b-Instruct（+SFT)** | **71.19(+4.3)** | **49.88(+6.15)** | **63.07(+5)** |
> > >
> > > Despite the challenges posed by the complexity of CS tasks and the limited size of the training dataset, the SFT process lead to noticeable improvements in both knowledge and reasoning performance. We anticipate that fine-tuning with a larger volume of domain-specific data could further enhance LLMs' capabilities in CS.
> > >
> > > Through the comprehensive analysis and discussions presented in CS-Bench, we aim to provide insights and directions for advancing LLMs in the field of computer science.

---

> > > > ### Author Response · Authors · 2024-11-21
> > > > **Response (4/4)**
> > > >
> > > > ### Question 1: What would the key contributions of the benchmark be to the field at large, as it stands today?
> > > >
> > > > As shown in lines 113-123 of the manuscript, we first introduced CS-Bench, the first benchmark for evaluating LLMs in the CS field. Then, using CS-Bench, we conducted a comprehensive assessment of mainstream LLMs and further quantitatively analyzed the relationship between model performance  and model size, reasons for existing LLMs' failures, and directions for improvement. Finally, we performed exploratory experiments on the cross-capabilities of LLMs using CS-Bench.
> > > >
> > > > In summary, on one hand, we believe this benchmark can inspire the community to gain a deeper understanding of LLMs' strengths and limitations in the CS field, providing a tool for measuring and analyzing the further development of LLMs in CS. On the other hand, given the correlation between CS, Code, and Math abilities, CS-Bench can also serve as a tool for cross-capability research and analysis.
> > > >
> > > > ---
> > > >
> > > > ### Question 2: Can the authors evaluate the value of the data through one of several Value-of-Data approaches including perplexity of training on the benchmark, FT-ing on the benchmark or other approaches?
> > > >
> > > > We emphasize that CS-Bench serves as a benchmark for measuring LLM performance in CS field. From the perspective of the test set, its value lies in the following aspects:
> > > >
> > > > 1. **Authentic Data Collection** (validated by the perplexity of LLMs on CS-Bench, as shown in Table 1).
> > > > 2. **Appropriate Difficulty Levels** (scores of various LLMs range from 39.86% to 72.29%).
> > > > 3. **Diversity** (distinguishing knowledge and reasoning, featuring multiple task formats and sub-domains).
> > > > 4. **Multilingual Coverage** (English, Chinese, German, and French).
> > > >
> > > > To further quantify the value of the dataset, we conduct fine-tuning experiments using CS-Bench.
> > > >
> > > > Specifically, we randomly sample 10% of the English data from CS-Bench as a mini-test set, with the remaining data converted into SFT training data. For SFT, the input corresponds to the original question, while the output contains the answer content (for multiple-choice questions, the correct option's content is used as the output).
> > > >
> > > > After training, we evaluate the perplexity of the model on the mini test set, focusing on the output part given the input. The results are as follows:
> > > >
> > > > |  | PPL |
> > > > | --- | --- |
> > > > | Llama3-8B-Instruct | 9.321 |
> > > > |  **Llama3-8B-Instruct(+SFT)** | **6.562** |
> > > >
> > > > We observe that training the model on CS-specific data significantly reduced its perplexity on the test set. Further evaluation of the model's performance on the mini-test set shows the following score changes:
> > > >
> > > > | miniTest | knowledge | reasoning | overall |
> > > > | --- | --- | --- | --- |
> > > > | Llama3-8b-Instruct | 66.89 | 43.73 | 58.07 |
> > > > | **Llama3-8b-Instruct(+SFT)** | **71.19(+4.3)** | **49.88(+6.15)** | **63.07(+5)** |
> > > >
> > > > After SFT, Llama3-8b-Instruct show significant improvements in both knowledge and reasoning tasks. This further demonstrates that the reduction in perplexity is not due to format fitting but rather to the model's acquisition of CS knowledge and reasoning abilities, highlighting the value of the CS-Bench.

---

> ### Author Response · Authors · 2024-12-02
> **A Kind Reminder for Reading the Response**
>
> Dear Reviewer 7dpp,
>
> Thank you for your insightful suggestions. We have done our best to address your concerns. Since the rebuttal period is closing very soon, could you please check the response to see whether it mitigates your concerns? We would greatly appreciate that!
>
> Thank you for your time and consideration, the authors.

---

### Official Review · Reviewer_xmLP · 2024-11-08

**Soundness:** 3
**Presentation:** 3
**Contribution:** 4
**Rating:** 8
**Confidence:** 3

**Summary:**

Develops CS Bench, a benchmark dataset of CS expertise that tests the ability of LLMs in computer science, as well as the relationship between CS math and code programming ability.  The benchmark measures 4 languages and 26 sufields with 9676 samples. To test this they evaluated using 30 LLMs.

The datasets were constructed using professional exams and practice tests widely available on the internet; extraction from blog articles of questions and answers; teaching materials and exams from the authors institutions.  For scoring, the authors use autoraters (GPT4) for genration tasks and  for comprehension tasks use questions where they can score with a regex

**Strengths:**

The strength of the paper is the dataset . This type of dataset is extremely valuable and I especially appreciate the sourcing of exam questions from universities and the knowledge transfer from blogs across specialized fields with such a large corpus of questions.  The final version of the paper should explain exactly where the questions came from, including giving the class and level at the university so that the difficult level is clear.

I'd also call out the thoroughness of the analysis. The different correlations noted, eg math and CS ability and so forth, are interesting as is the relationship between performance and tokens across scales. None of these are unexpected but they are nice things to see.

**Weaknesses:**

The first sentence "Computer Science (CS) stands as a testament to the intricacies of human intelligence, profoundly advancing the development of artificial intelligence and modern society"  is a bit much and some of the corresponding language needs to be toned down. Could we please not philosphize about the importance of specific fields (especially those were we are practitioners) and instead focus on the task that the paper accomplishes -- building a novel eval set and using it.

Other than this, the generative questions that require autorating to score are less convincing than the regex. The authors are right to emphasize that people should focus on this dataset if they can, but it does make me wonder whether the generative questions could be downgraded even further.

**Questions:**

What does your dataset not include for testing compettency in CS?
What level does this test? The level is notably absent from the paper -- but staging this relative to expertise (high school, undergrad, grad, etc) is extremely useful.
What is the breakdown of problems in the benchmark to expertise?

---

> ### Author Response · Authors · 2024-11-21
> **Response**
>
> Thank you for your helpful comments and recognition of our work in providing a valuable dataset and thorough analysis. Additionally, based on your suggestions, we will accurately explain the source of the issues in the final version.  We will address your concerns point by point below:
>
> ---
>
> ### Weakness 1: Some sentences need to be toned down, avoiding over-philosophizing discussions.
>
> We greatly appreciate your suggestions and have made initial revisions to the relevant sentences in the manuscript. We will continue to refine other writing expressions to further improve the overall quality.
>
> ---
>
> ### Weakness 2: The generative questions that require autorating to score are less convincing than the regex-based ones, leading to the consideration of whether the generative questions could be downgraded even further.
>
> We understand your concerns.
>
> First, generative tasks comprise only a small portion of CS-Bench (below 20%). These tasks don't significantly impact the overall score but greatly enrich task types and case studies. To demonstrate the effectiveness of LLM scoring, Table 11 in Appendix D2 showed high consistency between GPT-4 and human scoring, while Table 12 also explored the impact of different scoring models.
>
> Combining suggestions from other reviewers, we further explore scoring generative tasks using rule-based metrics or Regex and calculate their Pearson correlation with human scores. The results are as follows:
>
> | Metric | BLEU | ROUGE-1 | ROUGE-2 | ROUGE-L | METEOR | Regex |
> | --- | --- | --- | --- | --- | --- | --- |
> | Consistency | 0.0977 | 0.5949 | 0.2667 | 0.4848 | 0.5678 | 0.5982 |
>
> The results show that all the Pearson correlation coefficients of these metrics do not exceed 0.60, indicating poor consistency with human annotations. We believe this is because regex methods can only perform surface-level matching and score calculation, while LLM and human evaluation standards can involve deeper semantic and multi-dimensional understanding. We hope the above experiments and analysis can alleviate your concerns.
>
> ---
>
> ### Question 1: What does your dataset not include for testing competency in CS and what level does this test?
>
> We address your question in two points below:
>
> **Aspects not included in CS-Bench:**
>
> On one hand, CS-Bench focuses on comprehensively evaluating LLMs' fundamental knowledge and reasoning abilities in computer science, thus excluding assessments of LLMs in specific scenarios. This is because a key motivation for CS-Bench is that while some studies have explored LLMs' specific applications in scenarios like cybersecurity and software engineering, a comprehensive assessment of LLMs' more basic knowledge and reasoning capabilities in computer science has been lacking (lines 509-515).
>
> On the other hand, considering the extensive research already conducted by the LLM community on coding abilities, we view programming as a separate focus area. Therefore, CS-Bench does not include programming questions but instead focuses on the relationship between CS capabilities and coding abilities.
>
> **The test level of CS-Bench:**
>
> In Appendix A Limitations, we described the overall testing level of CS-Bench as university-level difficulty. We believe this is an appropriate difficulty to differentiate between LLMs, as reflected in the score range from 39.86% to 72.29%. Moreover, this level of difficulty also poses a challenge to the best-performing existing models (GPT-4o only achieves 64.15 points in reasoning-type questions).
>
> The level is primarily influenced by our data sources and the tendencies of designated annotators during data collection. While CS-Bench maintains an overall university-level difficulty, the questions exhibit a diverse range of complexity, spanning from simple data structure definitions to complex computer network application problems.
>
> We have expanded Appendix A from "Limitations" to "Limitations and Dataset Bias" in the new manuscript, providing further supplementation and modifications.

---

### Author Response · Authors · 2024-11-22
**General Response**

Dear Reviewers:

We sincerely thank you for your thorough review of our manuscript and the valuable comments and suggestions you provided. We appreciate the reviewers‘ praise of our work on the following aspects: **valuable dataset** (xmLP, 7dpp, g1yj, Rfxy), **comprehensive and in-depth analysis** (xmLP, 7dpp, g1yj, Rfxy), **extensive error analysis** (g1yj, Rfxy), **multilingual support** (Rfxy), and **logarithmic scaling laws** (g1yj, Rfxy).

To improve the quality of our paper, we make the following modifications and additions in response to the reviewers‘ comments (marked in red):

1. **Add Appendix E5 "Analysis from Model Representation"** to analyze the relationships between code, mathematics, and computer science capabilities discussed in Section 3.5 from the perspective of internal representations and data characteristics.
2. **Add Appendix E6 "Exploration for Improving CS Performance"** to explore and experiment with several specific approaches to enhance CS capabilities, including capability transfer, knowledge supplementation, integration of symbolic reasoning, and targeted fine-tuning, based on error analysis and findings.
3. **Improve Appendix A "Limitations and Dataset Bias"** with a focus on discussing the difficulty levels, evaluation scope, and potential biases present in CS-Bench.
4. **Improve Appendix D.2 "GPT-4 Scoring Details"** by adding more information about the manual evaluation process.
5. **Optimize some language expressions and visualizations**, mainly in the introduction section and Figure 3.

Additionally, in line with the reviewers‘ suggestions, **we develop a dynamic data subset for non-public maintenance**. This subset dynamically generates test instances through random number generation, random negation, and sentence rewriting to detect whether models have undergone private training on this benchmark, effectively reducing data contamination.

We believe these modifications and additions will better address your concerns and improve the quality and readability of the paper. We look forward to your further feedback.

---

### Author Response · Authors · 2024-11-27
**Looking forward to feedback on the Responses.**

Dear Reviewers:

Hello! We have updated the responses and manuscript to your constructive and insightful comments, and we would like to kindly ask you to take a look at our responses and reevaluate our work based on our clarifications. Please let us know whether our response addresses your concerns or whether there is any further detail we can provide to help address them. We appreciate your time and consideration!

The authors.

---

> ### Comment · Reviewer_g1yj · 2024-11-27
>
> Sure thing. Taking a look!

---

### Meta-Review · Area_Chair_T1z6 · 2024-12-19

**Metareview:**

The paper "CS-Bench: A Comprehensive Benchmark for Large Language Models towards Computer Science Mastery" introduces a novel and extensive benchmark, CS-Bench, designed to evaluate the performance of Large Language Models (LLMs) in the domain of computer science. CS-Bench comprises approximately 10,000 meticulously curated test samples across 26 subfields within four key areas of computer science, supporting multilingual evaluation in English, Chinese, French, and German. The benchmark encompasses various task formats, including multiple-choice, assertion, fill-in-the-blank, and open-ended questions. The authors evaluate over 30 mainstream LLMs using CS-Bench, revealing insights into the relationship between model scale and performance, identifying reasons for model failures, and exploring the correlation between LLM capabilities in computer science, mathematics, and coding. The paper also discusses potential improvements, such as knowledge supplementation and CS-specific reasoning, and introduces measures to mitigate data contamination risks.

#### Contributions
1. **CS-Bench Benchmark**:
   - A comprehensive, multilingual benchmark for evaluating LLMs in computer science, covering diverse subfields and task formats. CS-Bench is the first of its kind and addresses a significant gap in the LLM evaluation landscape.

2. **Error Analysis and Improvement Directions**:
   - Detailed analysis of error types, identifying knowledge gaps and reasoning challenges as key areas for improvement. The paper explores potential solutions, including knowledge supplementation, symbolic reasoning integration, and targeted fine-tuning.

3. **Cross-Capability Analysis**:
   - Exploration of the correlation between CS, mathematics, and coding capabilities, demonstrating that specialized models in mathematics and coding can enhance performance in specific CS subfields.

4. **Mitigation of Data Contamination**:
   - Introduction of dynamic data generation strategies to maintain a private subset of the benchmark, reducing the risk of data contamination.




#### Weaknesses
1. **Clarity and Presentation**:
   - Some reviewers noted that the writing and visualization could be improved, particularly the radar charts, which were found to be dense and difficult to interpret (g1yj). The authors have addressed this by replacing the radar charts with bar charts in the revised manuscript.
   - The paper's narrative occasionally includes over-philosophical statements, which could be toned down for a more focused presentation on the benchmark's contributions (xmLP).

2. **Depth of Analysis**:
   - While the paper demonstrates correlations between CS, mathematics, and coding capabilities, it lacks a deeper analysis of the underlying factors driving these relationships. Reviewers suggested exploring architectural or training data characteristics to explain these correlations (g1yj).
   - The error analysis, while comprehensive, does not offer concrete strategies for improving LLM performance beyond general recommendations (g1yj). The authors have since provided exploratory experiments on knowledge supplementation, symbolic reasoning, and fine-tuning, but further specificity could enhance the paper's practical value.

3. **Benchmark Biases**:
   - The paper acknowledges potential biases, such as the focus on university-level content, but does not fully explore how these biases might influence results or limit the benchmark's generalizability (g1yj, 7dpp). The authors have expanded the discussion in the revised appendix, but a more critical reflection on the benchmark's limitations could strengthen the paper.

4. **Evaluation Methodology**:
   - The reliance on GPT-4 for scoring generative tasks raises concerns about validity and reproducibility (g1yj). The authors have explored alternative metrics and fine-tuned smaller models for scoring, but further validation of these alternatives could improve confidence in the benchmark's evaluation framework.
   - The use of GPT-4 for multilingual translations was questioned, with concerns about translation quality (Rfxy). The authors have provided evidence of translation quality through back-translation and human evaluation, but caution is advised when interpreting results across languages.

5. **Data Contamination**:
   - The static nature of CS-Bench makes it vulnerable to data contamination, as LLM developers could train directly on the benchmark (Rfxy). The authors have implemented dynamic generation strategies for a private subset of the data, but this may not fully mitigate contamination risks for the public dataset.

**Additional Comments On Reviewer Discussion:**

1. **Clarity and Presentation (xmLP, g1yj):**
   - **Concern**: The manuscript's writing and visualizations were occasionally dense, with radar charts criticized for lacking clarity.
   - **Response**: The authors replaced radar charts with more intuitive bar charts and refined the writing to improve clarity and conciseness.
   - **Impact**: Reviewers acknowledged the improvements, with g1yj noting that the revised manuscript addressed presentation concerns effectively.

2. **Depth of Analysis (g1yj):**
   - **Concern**: The analysis of cross-capability relationships and error types lacked depth, with a need for more actionable insights.
   - **Response**: The authors conducted additional experiments to explore capability correlations at the representation level, analyzed training data characteristics, and proposed concrete strategies for improvement, including knowledge supplementation, symbolic reasoning integration, and fine-tuning.
   - **Impact**: Reviewer g1yj appreciated the additional experiments, particularly the exploration of capability transfer and reasoning frameworks, but suggested further exploration of architectural factors.

3. **Benchmark Biases (g1yj, 7dpp):**
   - **Concern**: The potential biases in CS-Bench, such as its focus on university-level content, were not thoroughly discussed.
   - **Response**: The authors expanded the discussion in Appendix A, acknowledging the benchmark's focus on university-level difficulty and task-centric evaluation, and elaborated on the implications for LLM assessment.
   - **Impact**: Reviewer g1yj noted the improved discussion, but emphasized the need for broader task coverage to enhance generalizability.

4. **Evaluation Methodology (g1yj):**
   - **Concern**: The reliance on GPT-4 for scoring generative tasks raised validity concerns, with suggestions for alternative metrics or models.
   - **Response**: The authors explored traditional metrics (e.g., BLEU, ROUGE) and fine-tuned smaller models for scoring, demonstrating their feasibility and limitations compared to GPT-4.
   - **Impact**: Reviewer g1yj found the exploration of alternative metrics valuable but highlighted the need for further validation of expert model-based scoring.

5. **Data Contamination (Rfxy):**
   - **Concern**: The static nature of CS-Bench posed risks of data contamination, with suggestions for dynamic generation methods.
   - **Response**: The authors implemented dynamic data generation strategies, including random number switching, statement rewriting, and semantic negation, to maintain a private subset of the benchmark.
   - **Impact**: Reviewer Rfxy commended the strategies as a feasible solution, acknowledging the inherent trust required in the community to avoid intentional contamination.

6. **Translation Quality (Rfxy):**
   - **Concern**: The use of GPT-4 for multilingual translations was questioned, with concerns about quality and accuracy.
   - **Response**: The authors validated translation quality through back-translation and human evaluation by native speakers, demonstrating high accuracy across languages.
   - **Impact**: Reviewer Rfxy found the validation impressive but advised caution when interpreting results across languages due to potential translation inconsistencies.


**Final Decision Weighting:**
The paper presents a significant and impactful contribution to the LLM evaluation landscape with the introduction of CS-Bench. The authors' responsiveness to reviewer feedback, including additional experiments and clarifications, demonstrates a strong commitment to improving the manuscript. While some concerns about depth of analysis, benchmark biases, and evaluation methodology persist, the authors have made substantial efforts to address these issues. The overall positive feedback from most reviewers and the paper's novel contributions position it well for acceptance, potentially with minor revisions to further address remaining concerns about benchmark biases and evaluation methodology.

---

### Decision · Program_Chairs · 2025-01-22

Accept (Poster)